# Myc-dependent dedifferentiation of Gata6⁺ epidermal cells resembles reversal of terminal differentiation

Miguel Bernabé-Rubio[1], Shahnawaz Ali [1], Priyanka G. Bhosale[1], Georgina Goss[1], Seyedeh Atefeh Mobasseri[1], Rafael Tapia-Rojo[2,3], Tong Zhu [2,3], Toru Hiratsuka[1,4], Matteo Battilocchi[1], Inês M. Tomás[1], Clarisse Ganier[1], Sergi Garcia-Manyes[2,3] & Fiona M. Watt [1,5] ✉

Dedifferentiation is the process by which terminally differentiated cells acquire the properties of stem cells. During mouse skin wound healing, the differentiated Gata6-lineage positive cells of the sebaceous duct are able to dedifferentiate. Here we have integrated lineage tracing and single-cell mRNA sequencing to uncover the underlying mechanism. Gata6-lineage positive and negative epidermal stem cells in wounds are transcriptionally indistinguishable. Furthermore, in contrast to reprogramming of induced pluripotent stem cells, the same genes are expressed in the epidermal dedifferentiation and differentiation trajectories, indicating that dedifferentiation does not involve adoption of a new cell state. We demonstrate that dedifferentiation is not only induced by wounding, but also by retinoic acid treatment or mechanical expansion of the epidermis. In all three cases, dedifferentiation is dependent on the master transcription factor c-Myc. Mechanotransduction and actin-cytoskeleton remodelling are key features of dedifferentiation. Our study elucidates the molecular basis of epidermal dedifferentiation, which may be generally applicable to adult tissues.

Most adult tissue stem cells exhibit striking plasticity, switching from one stem cell type to another or generating additional differentiated lineages in response to injury[1–4]. Despite recent advances in single-cell technologies[5–7], the molecular mechanisms responsible for such plasticity are not fully understood.

The process through which terminally differentiated cells revert to a less differentiated state is known as dedifferentiation. Dedifferentiation is crucial for tissue regeneration in lower vertebrates[8]. Although regeneration in mammals is more restricted, reversion of differentiated cells to stem cells after injury is reported in airway epithelium[9], intestinal crypts[10] and neuronal progenitors[11].

It was previously believed that in multilayered epithelia, such as the epidermis, differentiated cells cannot revert to the stem cell state[12,13]. However, during wound healing, terminally differentiated Gata6⁺ cells dedifferentiate, acquiring the ability to self-renew and differentiate into a broad range of epidermal lineages that contribute to long-term maintenance of the interfollicular epidermis (IFE)[14].

In this Article, we have combined lineage tracing and single-cell RNA sequencing (scRNA-seq) of Gata6⁺ cells to capture cell trajectories during dedifferentiation. We show that dedifferentiation occurs via reversal of the normal differentiation pathway and that wounding, retinoic acid (RA) treatment and mechanical expansion stimulate

[1]Centre for Gene Therapy and Regenerative Medicine, King's College London, London, UK. [2]Department of Physics, Randall Centre for Cell and Molecular Biophysics, Centre for the Physical Science of Life and London Centre for Nanotechnology, King's College London, London, UK. [3]Single Molecule Mechanobiology Laboratory, The Francis Crick Institute, London, UK. [4]Department of Oncogenesis and Growth Regulation, Research Center, Osaka International Cancer Institute, Chuoku, Japan. [5]Directors' Unit, EMBL Heidelberg, Heidelberg, Germany. ✉e-mail: fiona.watt@kcl.ac.uk

dedifferentiation. We identify Myc as a driver of dedifferentiation, consistent with its role in epigenetic cell reprogramming to pluripotency.

## Results

### Single cell transcriptomics of Gata6lin⁺ and lin⁻ cells

We performed scRNA-seq on genetically labelled Gata6 lineage (Gata6lin⁺) cells in wounded and unwounded skin (Fig. 1a). We crossed Gata6EGFPCreERT2 with Rosa26-fl/STOP/fl-tdTomato mice, and treated adult epidermis topically with a low concentration of 4-hydroxy-tamoxifen to selectively label differentiated Gata6-expressing cells[14]. One week later we made a 6-mm-diameter circular full-thickness wound in the back skin. Then, 6, 9 and 11 days post-injury, skin from the wound site was collected using a 4-mm-diameter circular biopsy punch to avoid contamination by non-wounded skin (Extended Data Fig. 1a). As a control, a 4-mm-diameter punch biopsy was collected from back skin distant from the wound in the same mice (day 0).

In unwounded back skin Gata6lin⁺ cells were confined to the upper hair follicle (HF) and sebaceous duct (SD). Lin⁺ cells were suprabasal; lin⁺ basal cells were undetectable (Fig. 1b and Extended Data Fig. 1b,c). On day 6 after wounding a few Gata6lin⁺ cells had exited the HF and were present in the suprabasal layers of the newly formed IFE (Extended Data Fig. 1b,c). At day 9, some Gata6lin⁺ cells in HFs expressed high levels of Itga6 and were in contact with the basement membrane (Fig. 1b and Extended Data Fig. 1b,c). In the re-epithelialized IFE, lin⁺ cells were present in the basal layer and had founded columns of differentiating, suprabasal cells (Fig. 1b).

To compare Gata6lin⁺ and lin⁻ epidermal cells, we flow sorted tdTomato⁺ and tdTomato⁻ cells from wounded and control skin (Fig. 1a and Extended Data Fig. 1d). By sorting Itga6$^{low/mid}$ and Itga6$^{high}$ populations we could distinguish basal (Itga6$^{high}$) from suprabasal (Itg6$^{low/mid}$) cells (Extended Data Fig. 1d). Control Gata6lin⁺ cells almost exclusively expressed low/medium levels of Itga6. In contrast, lin⁻ cells expressing high levels of Itga6 were detected in control skin (Extended Data Fig. 1d). As expected, a substantial fraction of Gata6lin⁺ cells in wounds expressed high levels of Itga6, consistent with re-acquisition of basal cell characteristics (Fig. 1a and Extended Data Fig. 1d–f).

We generated scRNA-seq libraries from 684 cells, comprising Gata6lin⁺ and lin⁻ cells from wounded and unwounded epidermis (Extended Data Fig. 1g), and analyzed the data using the R package Seurat. Projections of lin⁻ and lin⁺ cells onto a t-distributed stochastic neighbour embedding (tSNE) map containing epidermal cells from unwounded telogen skin[15] confirmed that in unwounded skin lin⁻ cells mapped primarily to the basal IFE layer (IFE-B) while Gata6lin⁺ cells mapped to the upper HF compartments (Fig. 1c). Using the second level of clustering from the Joost dataset[15], lin⁺ cells strongly correlated with upper hair follicle population V (uHF-V) (Extended Data Fig. 1h), consistent with the lineage tracing evidence that Gata6lin⁺ cells are suprabasal.

Two-dimensional space uniform manifold approximation and projection for dimension reduction (UMAP) analysis of Gata6lin⁺ and lin⁻ epidermal cells from all three wound timepoints identified six clusters, all containing lin⁺ and lin⁻ cells (Fig. 1d). Thus, Gata6lin⁺ and lin⁻ cells have a common transcriptional identity in healing wounds,

consistent with the reported lineage infidelity of wound epidermal cells[5]. The percentage of lin⁺ cells in each cluster was as follows: 30% (cluster 0), 25% (cluster 1), 48% (cluster 2), 81% (cluster 3), 34 % (cluster 4) and 44% (cluster 5).

Cluster 3, with the highest proportion of lin⁺ cells, comprised differentiating uHF (Krt79⁺ Cst6⁺) and IFE (Calm4⁺ Krt23⁺ Krt1⁺ Krt10⁺) cells (Extended Data Fig. 1i). This is consistent with Gata6lin⁺ cells that exit the HF being initially suprabasal[14] (Extended Data Fig. 1b,c). Messenger RNA (mRNA) in situ hybridization confirmed that Gata6lin⁺ cells expressed Cst6 (ref. 16) and an additional uHF marker, Defb6 (ref. 15), in upper HFs and suprabasal cells of wounded IFE (Fig. 1e). Cluster 0 comprised cells expressing sebaceous gland (SG) markers (Mgst1 and Scd1), cluster 4 expressed outer bulge (OB) markers (Postn and Cxcl14) and cluster 5 expressed IFE markers (Serpinb2 and Ifngr1). We could not assign cluster 1 or 2 to a defined epidermal compartment: three of the most highly expressed genes in cluster 1 were histone family members (H2ac23, H2ac24 and Cenpa), while cluster 2 had no significantly upregulated genes.

### Dedifferentiation trajectory of Gata6 lineage cells

UMAP segmentation of Gata6lin⁺ cells revealed three clusters in the dedifferentiation trajectory. Day 9 and 11 wound cells mainly segregated in cluster 1. Control and day 6 wound cells segregated in cluster 2. Cluster 3 comprised control cells and cells from day 9 and day 11 wounds that had remained in their original HF niche (Fig. 2a).

We used Monocle software to perform pseudotime analysis (Fig. 2a). The trajectory of Gata6lin⁺ cells from upper HF-like to IFE-like was confirmed by constructing a heat map of markers from unwounded telogen skin[15] (Fig. 2b). Cell identity annotation showed that uHF-II was enriched in cluster 3 and cluster 2, and IFE-B in cluster 1 (Fig. 2b). Feature plots of known markers confirmed the identity of the predicted epidermal compartments (Extended Data Fig. 2a). The uHF-II gene signature includes Krt79 and Krt17 (Extended Data Fig. 2a), consistent with the suprabasal signature of control Gatalin⁺ cells (Extended Data Fig. 1h). IFE-B markers include Krt14 and Metallothionein-2 (Mt2); OB markers include Cd34 and Postn (Extended Data Fig. 2a). The enrichment of OB cells in the dedifferentiation trajectory is consistent with reports that Gata6lin⁺ cells are present in the lower bulge following injury[14]. Wound cells predicted as IFE, uHF, OB and SG were transcriptionally close to each other, in line with cells transiently losing their identity during wound healing[5].

Further confirmation that Gata6lin⁺ cells transitioned from cluster 3 to 2 to 1 was obtained using CellRank, which estimates cell state dynamics, detecting initial, intermediate and terminal populations (Fig. 2c). Cells in cluster 3 had a suprabasal signature, whereas cluster 1 showed a basal signature (Fig. 2d). Upregulation of the wound healing markers Krt6a and Krt6b was found in wound cells in all clusters (Fig. 2e). Gata6 expression was mainly observed in clusters 3 and 2 (Fig. 2e), consistent with Gata6lin⁺ cells losing Gata6 expression as they reach the IFE[14].

Comparative gene expression studies confirmed the pseudotime analysis (Extended Data Fig. 2b,c). Downregulation of Defb6 and Cst6 (ref.16), expressed by differentiating cells in the upper HF[15], occurred in

---

**Fig. 1 | Gata6lin⁺ cells and lin⁻ cells in wounds are indistinguishable on the basis of their transcriptomes. a**, Schematic of the experimental design (top) and representative flow cytometric plots (bottom) showing Itga6 levels in Gata6lin⁺ cells at the indicated timepoints after wounding. Light-blue lines represent gating strategy used for quantification in Extended Data Fig. 1e. **b**, Representative section of day 9 wounded skin showing tdTomato Gata6lin⁺ cells stained for Itga6 (green). Nuclei are visualized with DAPI staining (blue). Boxed regions are shown at higher magnification. White arrows indicate lin⁺ cells attached to the basement membrane. Scale bars, 200 μm (overview) and 40 μm (magnifications). n = 4 independent experiments. **c**, Left: schematic of location of epidermal populations in undamaged skin. Right: transcriptomes

of control Gata6lin⁺ and lin⁻ epidermal cells were projected onto tSNE space re-analyzed from Joost et al. (2016)[15]. Note that Gata6lin⁺ cells mapped primarily to the uHF compartment. **d**, Gata6lin⁺ and lin⁻ cells isolated from wounds were visualized on a UMAP plot. Note that Gata6lin⁺ and lin⁻ cells were present in all six clusters. **e**, Detection of Cst6 and Defb6 by mRNA in situ hybridization in skin sections showing tdTomato Gata6lin⁺ cells 9 days after wounding. HFs proximal to wounds and healing IFE are shown. Boxed regions are shown at higher magnification. White arrows indicate lin⁺ cells expressing Cst6 (top) or Defb6 (bottom). Scale bars, 20 μm. Representative images from n = 3 independent experiments.

clusters 1 and 2 (Fig. 2f) and at day 11 (Extended Data Fig. 2c). The basal marker *Itga6* was upregulated in cluster 1 and at days 9 and 11 (Fig. 2f and Extended Data Fig. 2c), consistent with our fluorescence-activated cell sorting (FACS) analysis (Fig. 1a). According to our single-cell data, 95.7% of day 0 cells showed low Itga6 expression (Fig. 2f), within the

typically accepted range (95–99%) of purity for single-cell sorting. Thus, day 0 cells expressing high levels of Itga6 could be contaminants from the lin⁻ population with an IFE-B signature (Fig. 1c), as only one cell co-expressed Itga6 and Gata6. We could not distinguish the contribution of these cells to the dedifferentiation signature.

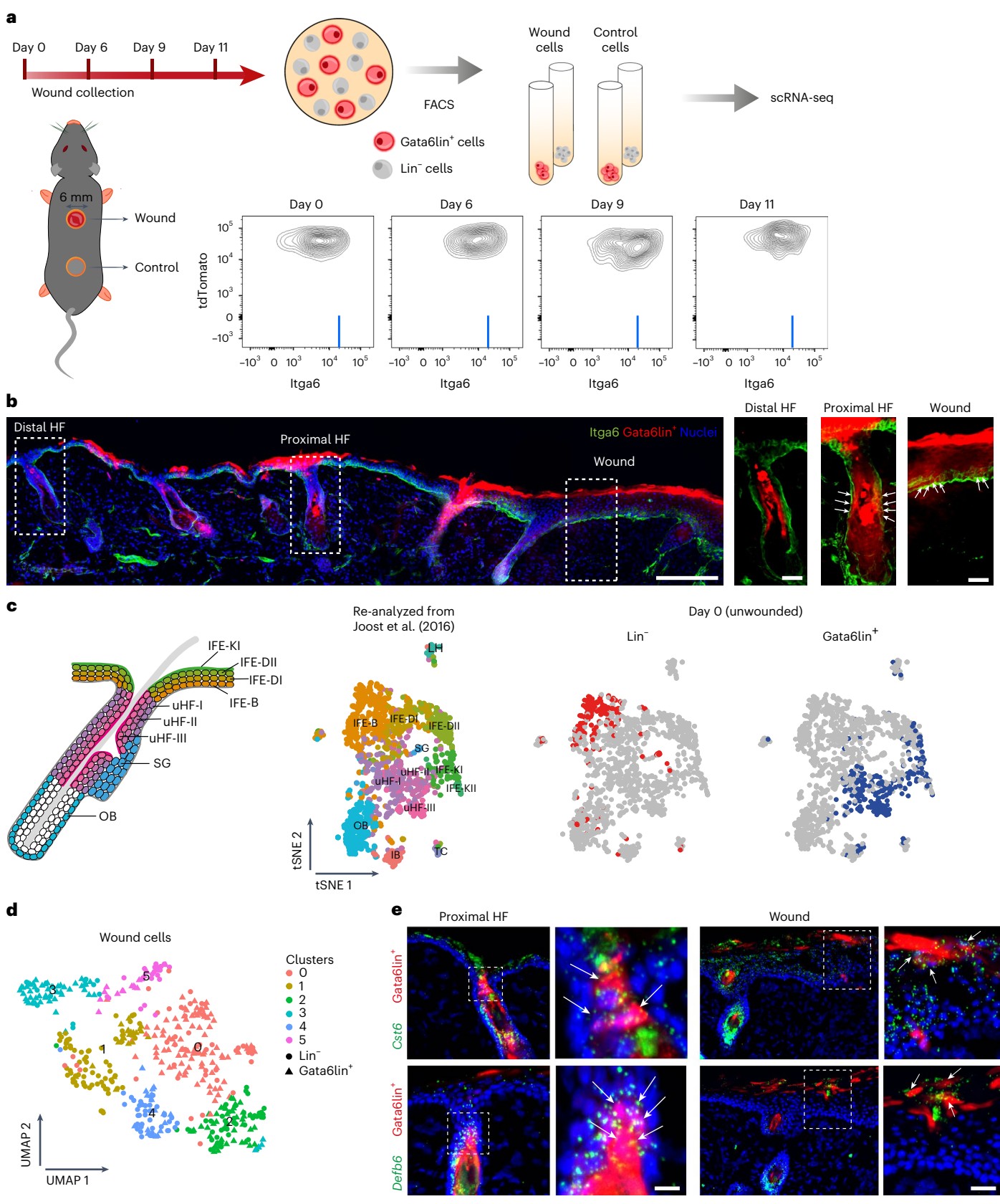

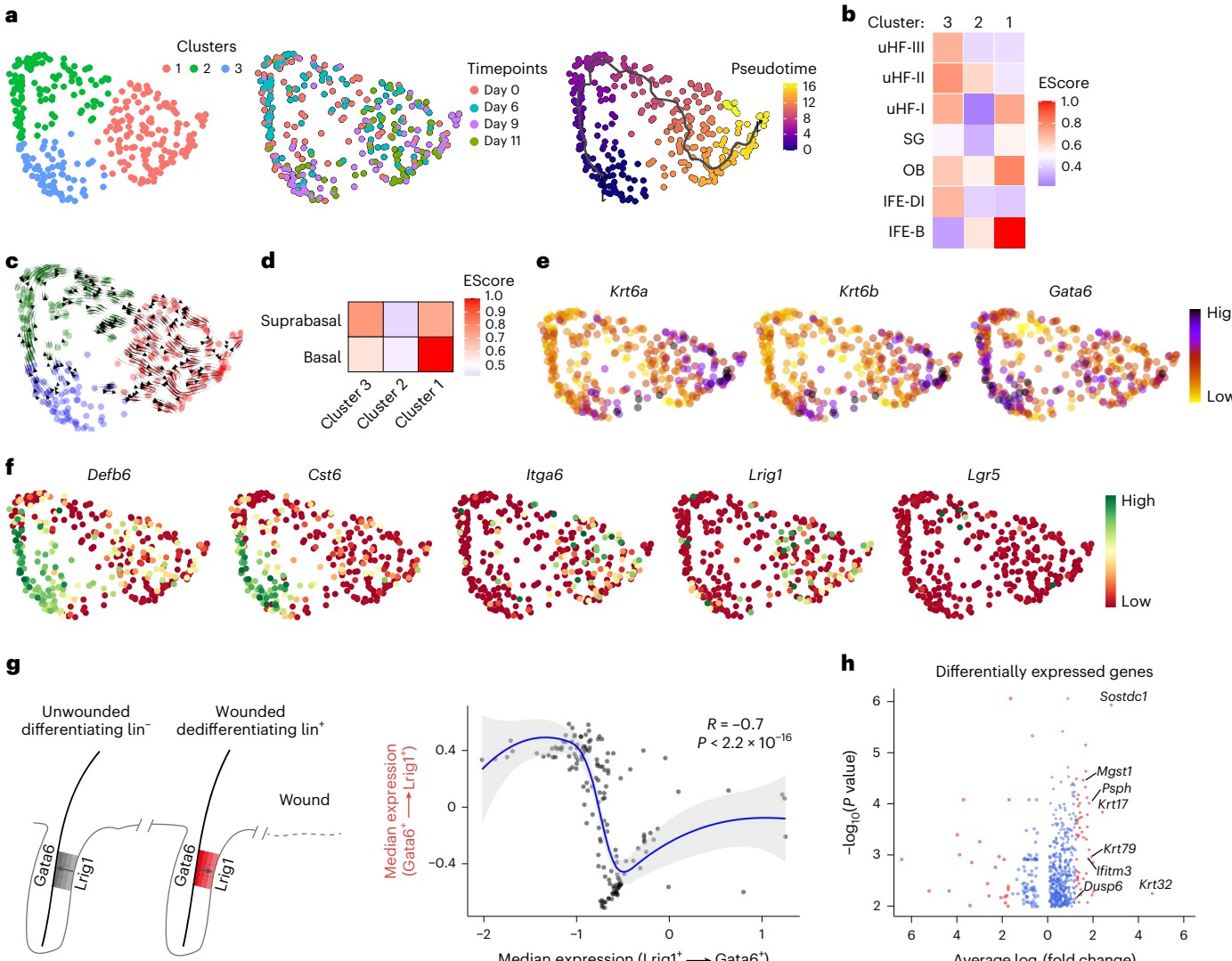

**Fig. 2 | Dedifferentiation occurs via reversal of the normal differentiation process. a**, Gata6lin+ cells were visualized on a tSNE plot. Left: Gata6lin+ cells mapped into three clusters, colour-coded according to unsupervised clustering. Middle: the indicated timepoints after wounding are shown. Right: pseudotime trajectory. **b**, Heat maps showing the correlation between the three clusters identified with Monocle and cell compartments defined in the Joost dataset[15]. **c**, CellRank was used to visualize future states (arrows) on the generated tSNE plot in **a**. **d**, Heat maps showing the correlation between the three clusters and basal and suprabasal epidermal compartments defined in Human Protein Atlas version 20.0. **e**, Expression of the wound markers *Krt6a* and *Krt6b*, and the SD marker *Gata6*. **f**, Expression of the upper HF markers *Defb6*, *Cst6*, basal marker *Itga6* and stem cell markers *Lrig1* and *Lgr5* is shown on the tSNE plot. Data in **a**–**f** are from two independent biological replicates per timepoint. All

Gata6lin+ cells from the scRNA-seq data were analyzed. **g**, Left: schematic of the normal differentiation process (Lrig1 to Gata6, control lin− cells) and the dedifferentiation process (Gata6 to Lrig1, wound lin+ cells). Right: transcriptional profile correlation between Gata6lin+ cells dedifferentiating to Lrig1+ stem cells (wounded skin) and Lrig1 lin− cells differentiating to Gata6+ cells (unwounded skin). Loess regression was used for fitting. Differentially expressed genes between the two populations are represented by dots (545 genes from 112 cells, 32 differentiating and 80 dedifferentiating cells). The solid line represents the Loess regression fit. The grey area indicates the error bands. Two-sided Wilcoxon rank sum test was used to identify differentially expressed genes. Adjustments were made for multiple comparisons. **h**, Volcano plot showing differentially expressed genes identified in **g**. Genes with a fold change greater than 1.6 are shown in red.

The junctional zone stem cell marker *Lrig1*, but not the lower HF stem cell marker *Lgr5*, was increased in Gata6lin+ cells in cluster 1 and at days 9 and 11, consistent with dedifferentiation to Lrig1+ stem cells (Fig. 2f and Extended Data Fig. 2c). Upregulation of *Krt6a*, *Krt6b*, *Krt14*, *Itga6* and *Lrig1* was more pronounced in day 9 and 11 wounds than at day 6, which more closely resembled day 0 (Extended Data Fig. 2b,c). Virtually all day 6 wound lin+ cells were suprabasal (Extended Data Fig. 1b,c), even though some cells expressed basal markers (Fig. 2f). The transition into a less differentiated stage was distinguishable both transcriptionally and spatially 9 days after wounding, when some cells had adhered to the basement membrane (Figs. 1b and 2f and Extended Data Fig. 1b,c).

Since the Gata6 lineage derives from Lrig1+ cells[14], we investigated whether the dedifferentiation trajectory resembled the forward differentiation trajectory (Fig. 2g). Comparing genes differentially expressed between Gata6lin+ cells that dedifferentiated to Lrig1+ stem cells and Lrig1+ lin− cells that differentiated to Gata6+ cells, we found a striking inverse correlation ($R = -0.7$; Fig. 2g). Thus, dedifferentiating Gata6lin+ cells have a similar transcriptome to cells undergoing differentiation, suggesting that dedifferentiation occurs via reversal of the normal differentiation process rather than occurring via a new pathway. Figure 2h shows a volcano plot of all differentially expressed genes in Fig. 2g (Supplementary Table 1). These include genes associated with

cell differentiation, such as *Dusp6*, *Sostdc1* and *Krt32*, and markers of uHF (*Krt79* and *Krt17*), IFE (*Ifitm3*) and SG (*Mgst1* and *Psph*) (Fig. 2h).

We next compared expression of Lrig1 and the proliferation marker Ki67 in Gata6lin+ cells after wounding (Extended Data Fig. 2d,e). At day 1 the number of lin+Lrig1+ cells had increased significantly, whereas the number of lin+Ki67+ cells had not. This indicates that the initial enrichment of lin+Lrig1+ cells was due to the switch of Gata6lin+ cells to Lrig1+ cells, and not to proliferation of Lrig1+ cells. Thus, initiation of the dedifferentiation process did not require cell proliferation.

## Myc as a mediator of dedifferentiation

To explore the dedifferentiation mechanism, we modelled transcriptional changes along the pseudotime trajectory so that we could visualize all the pseudotime-dependent genes and group them on the basis of similar trends. We detected 31 gene modules that were dynamically regulated across the three clusters in the pseudotime trajectory (Extended Data Fig. 3a,b and Supplementary Table 2).

To gain insight into regulation of the genes involved in dedifferentiation of Gata6lin+ cells, we performed a transcription factor (TF) motif analysis using the Transcriptional Regulatory Relationships Unraveled by Sentence-based Text mining (TRRUST) database[17] (Fig. 3a). The most upregulated TFs in cluster 1 were *Hspb1*, *Aft3*, *Id3*, *Myc* and *Gata3*. Myc is known to positively regulate *Hspb1* (ref. [18]), *Aft3* (ref. [19]) and *Id3* (ref. [20]), while Gata3 expression upregulates *Myc*[21]. We also performed ligand-receptor analysis based on the hypothesis that Gata6lin− cells were sender cells and Gata6lin+ cells were target cells. Predicted ligands expressed by lin− cells were used to predict receptors and target genes in Gata6lin+ cells (Extended Data Fig. 3c,d). From this analysis we identified possible regulators of cell state transition, including Myc and Id3.

We focused on Myc because it is known to play a central role in epidermal maintenance, wound healing and differentiation[22]. *Myc* and *Lrig1* expression were both upregulated in cluster 1 dedifferentiated cells (Fig. 3b). Biaxial plots showed that the level of co-expression of Myc and Lrig1 in wound cells was higher than in control cells ($r = 0.24$ and $r = 0.05$, respectively) (Fig. 3c). This was associated with proliferation as indicated by Ki67 expression ($r = -0.09$ in control cells and $r = 0.13$ in wound cells) (Fig. 3c).

To investigate the potential interplay between Gata6 and Myc, we performed colony formation assays, an in vitro measure of stem cell number, with cultured mouse epidermal cells. Keratinocytes from K14MycER mice, in which tamoxifen-inducible Myc is overexpressed in basal cells via the Krt14 promoter[23], were transduced with a doxycycline-inducible GATA6 lentiviral vector (Fig. 3d and Extended Data Fig. 3e–h). The combination of Gata6 and Myc overexpression resulted in increased colony formation (Fig. 3d,e) and reduced expression of the terminal differentiation marker involucrin (*Ivl*) (Fig. 3f). There was an increase, although not significant, in colony area (Extended Data Fig. 3e). These results suggest a synergistic effect of Gata6 and Myc in stem cell expansion.

## Dedifferentiation in wounded and unwounded epidermis

The number of Gata6+ cells increases in intact HFs adjacent to a wound[14]. Immunostaining showed an increased percentage of Gata6lin+Lrig1+ cells in HFs proximal to wounds compared with distal HFs (Fig. 4a,b). In addition, Myc was detected in the nuclei of Gata6lin+ cells, whether or not they co-expressed Lrig1 (Fig. 4a,c). The location of these cells led us to speculate that dedifferentiation of Gata6lin+ cells occurs within the upper HF. Consistent with this, live cell imaging of anaesthetized mice showed that before Gata6lin+ cell migration into the IFE, the Gata6lin+ compartment expanded within the HFs at the wound margin (Extended Data Fig. 4a,b).

To investigate whether Myc was required for dedifferentiation, we analyzed Gata6lin+ cells following deletion of Myc via the Gata6 promoter (Gata6CreER × Myc flox/flox mice). There was a reduced percentage of Lrig1+Gata6lin+ cells and Gata6lin+ cells in the IFE of healing wounds (Fig. 4d–g), indicating that Myc is necessary for dedifferentiation following wounding. Myc deletion in Gata6lin+ cells did not affect Myc expression in Lrig1+ lin− cells (Fig. 4h), or lin− cells in the IFE and bulge (Extended Data Fig. 4c). Although there were fewer Gata6lin+ proliferative cells, there was no impact on the number of Lrig1+ lin− proliferative cells (Extended Data Fig. 4d,e), suggesting that Myc is required for proliferation of the Gata6 lineage after wounding. This is consistent with the finding that Myc overexpression in unwounded epidermis results in Gata6 being expressed in the IFE and in Lrig1 lineage cells extending upwards into the adjacent IFE[14,24].

Epidermal thickening is part of normal wound healing (Extended Data Fig. 4f). Myc deletion in Gata6+ cells impaired injury-induced thickening of the upper HF (Extended Data Fig. 4g). mRNA in situ hybridization for *Lrig1* (upper HF), *Lgr6* (IFE and upper HF), and *Lgr5* (bulge) following injury revealed that Myc deletion resulted in fewer *Lrig1*+ cells (Fig. 4i,j), whereas *Lgr5* and *Lgr6* expression was unaffected (Extended Data Fig. 4h,i). This suggests that, upon wounding, Gata6lin+ cells act as a reservoir of Lrig1+ cells in a Myc-dependent manner.

To explore whether Myc was also required for expansion of Gata6+ cells in unwounded skin, we treated epidermis with RA, which is known to induce expansion of the Gata6 compartment[14] and epidermal hyperplasia[25]. Deletion of Myc prevented RA-induced thickening of the upper HF but not the IFE, and blocked RA-induced dedifferentiation of the Gata6 lineage (Extended Data Fig. 5a–d). We speculate that RA signalling is involved in dedifferentiation of Gata6+ cells, in keeping with reports that RA promotes reprogramming of induced pluripotent stem cells[26,27].

Since wound healing and RA treatment both result in epidermal thickening, we next examined whether mechanical expansion of the epidermis can induce dedifferentiation of Gata6lin+ cells. Mechanical expansion was induced by dermal injection of a methylcellulose-based hydrogel (Extended Data Fig. 5e). Hydrogel implantation induced a mechanical response in the epidermis 1 day after injection, as indicated by an increase in nuclear YES-associated protein 1 (YAP1) and megakaryoblastic leukaemia/myocardin-like 1 (MAL), two modulators of mechanotransduction[28] (Fig. 5a,b). There was a two-fold increase in proliferation 2 days after injection; proliferation returned to control levels by day 7 (Extended Data Fig. 5f,g). Epidermal thickness increased approximately two-fold by day 4 (Fig. 5c,d and Extended Data Fig. 5h). Gata6lin+ cells started migrating into the suprabasal layers of the IFE 1 day after injection, and some were located in the basal layer (Fig. 5e–g). They progressively moved into the basal layer and expressed Lrig1 (Fig. 5f–h). Thus, hydrogel-induced epidermal expansion induces dedifferentiation of Gata6lin+ cells.

Whereas inflammation is part of normal skin wound healing, it has previously been reported that epidermal proliferation after hydrogel implantation is not linked to inflammation[29]. In agreement with that conclusion, based on the number of CD45-positive cells in the skin, inflammation peaked 4 days after hydrogel implantation (Fig. 5i,j), 2 days after the increase in proliferation (Extended Data Fig. 5f,g). To further rule out a causal role for local inflammation in epidermal thickening and dedifferentiation, we injected mice with bleomycin. Bleomycin caused a strong inflammatory response, as judged by increased immune cells and upregulation of the inflammatory markers *Tnfa* and *Ccl2* (Fig. 5i,j and Extended Data Fig. 5i). However, epidermal thickness was not affected (Extended Data Fig. 5j) and bleomycin-treated mice did not show migration of Gata6lin+ cells or conversion of Gata6lin+ cells to the Lrig1 lineage (Fig. 5g–i). Although we cannot exclude the possibility that inflammation can trigger dedifferentiation in some contexts, bleomycin-induced inflammation was not involved in dedifferentiation of Gata6lin+ cells.

As in the case of wounding and RA treatment, Myc deletion in the Gata6 lineage blocked the increase in thickness of the upper HF caused by mechanical expansion (Extended Data Fig. 5k). The number

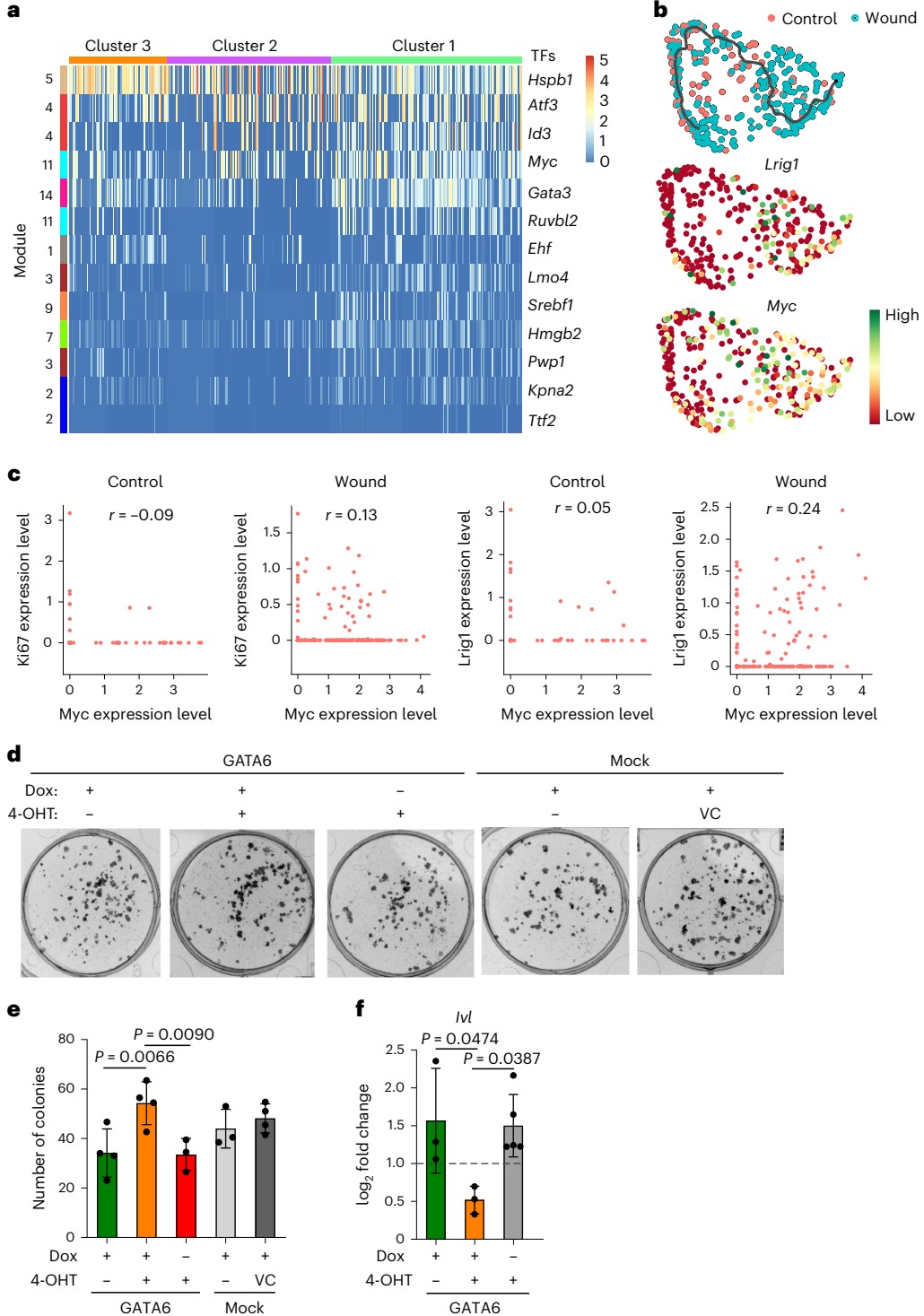

**Fig. 3 | Evidence for upregulation of Myc during dedifferentiation. a**, Heat map showing expression of TFs obtained from the TRRUST database after analysis of each gene module. TFs were selected on the basis of *q* value <0.5 and Maron value >0.9. Gata6lin⁺ cells distributed into the different clusters obtained in Fig. 2a are represented along the horizontal axis, and gene modules are shown along the vertical axis. **b**, Expression of *Lrig1* and *Myc* on the pseudotime trajectory. **c**, Scatter plots showing expression of *Ki67* and *Myc*, and *Lrig1* and *Myc* in control and wound cells. Spearman *r* coefficient is shown on the plots. Note that wound cells show a higher correlation between *Myc* and *Ki67*, and *Myc* and *Lrig1* compared with control cells. **d**, Keratinocytes from K14MycER transgenic mice were transfected with a doxycycline (Dox)-inducible GATA6 construct or

with mock lentiviruses and seeded at clonal density. Representative images of dishes showing colony formation 12 days after plating are shown. GATA6 or Myc expression was induced by addition of 2 µg ml⁻¹ Dox or 25 nM 4-OHT, respectively. One day after GATA6 induction, Myc expression was induced by adding 25 nM 4-OHT. Cells transfected with mock lentiviruses were pre-treated with or without ethanol (VC, vehicle control). **e,f**, Bar graphs showing number of colonies in each of the indicated conditions (**e**) and expression of *Ivl* measured by RT–qPCR (**f**). Data are the mean ± s.d. from three to four independent experiments. One-way ANOVA with Šidák's multiple comparisons test was used to determine statistical significance.

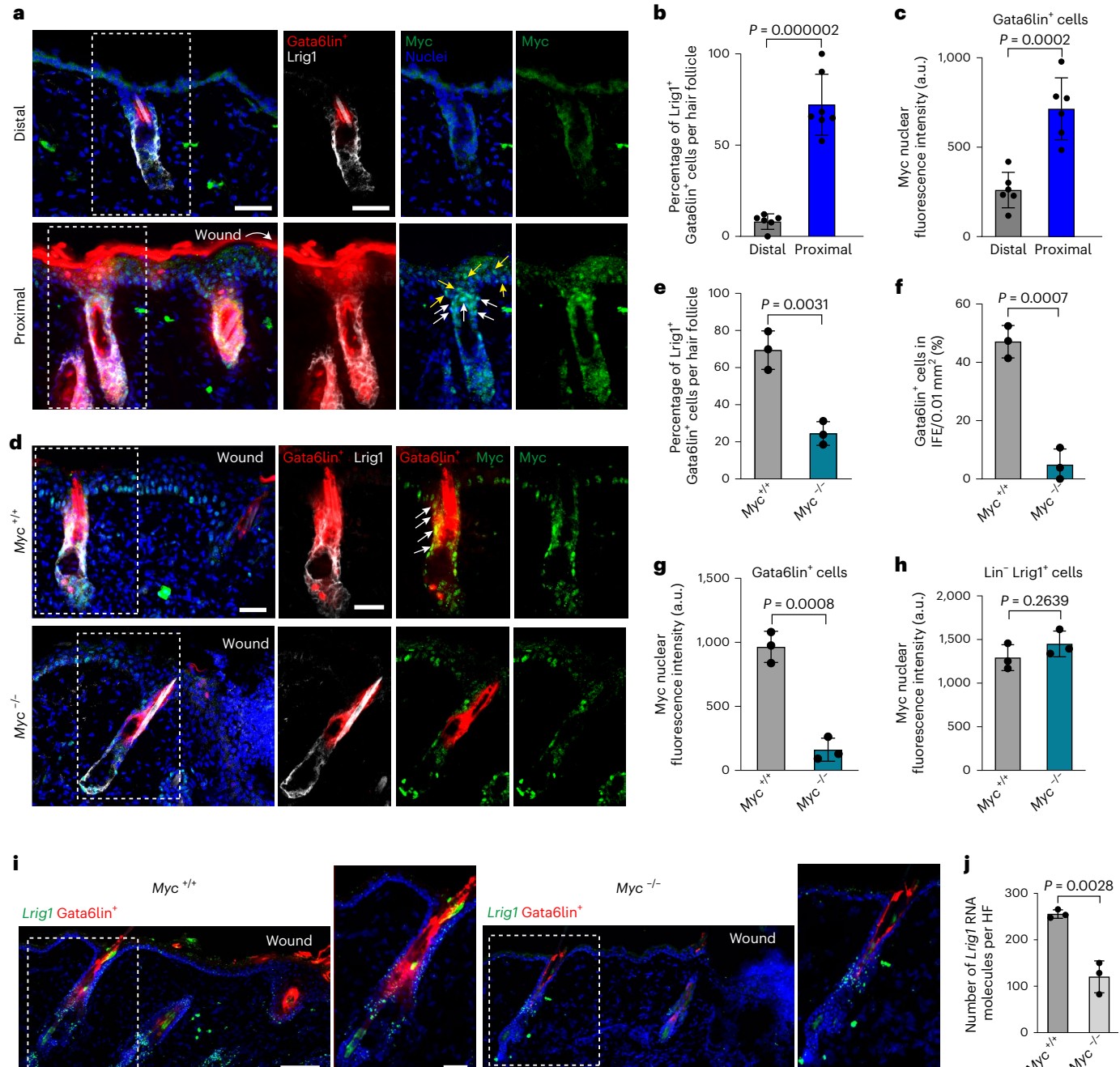

**Fig. 4 | Myc is required for wound-induced dedifferentiation of Gata6lin⁺ cells. a**, Representative sections of HFs distal and proximal to the wound site showing tdTomato Gata6lin⁺ cells stained for Lrig1 (grey) and Myc (green). Boxed regions show separation of channels. White and yellow arrows indicate Myc-expressing cells from the uHF and the IFE, respectively. Scale bar, 40 μm. *n* = 7 independent experiments. **b,c**, Bar graphs showing the percentage of Gata6lin⁺ cells expressing Lrig1 per HF (**b**) or Myc nuclear intensity (**c**). Data are the mean ± s.d. *n* = 7 mice (**b**); *n* = 6 mice (**c**). **d**, Skin of *Myc*⁺/⁺ and *Myc*⁻/⁻ mice showing tdTomato Gata6lin⁺ cells expressing Myc (green) and Lrig1 (grey). Arrows indicate Gata6lin⁺ cells expressing Myc. Note that Gata6lin⁺ cells in *Myc*⁻/⁻ mice do not express Myc. Scale bar, 40 μm. Representative images from *n* = 3 independent

experiments. **e,f**, Bar graphs showing the percentage of Gata6lin⁺ cells expressing Lrig1 per HF adjacent to wound (**e**) and per 0.01 mm² of healing IFE (**f**) in *Myc*⁺/⁺ and *Myc*⁻/⁻ mice. Data are the mean ± s.d. *n* = 3 mice per group. **g,h**, Bar graphs showing fluorescence intensity of nuclear Myc in Gata6lin⁺ cells (**g**) and lin⁻ Lrig1⁺ cells present in HFs (**h**). Data are the mean ± s.d. *n* = 3 mice per group. **i**, Detection of *Lrig1* by mRNA in situ hybridization in skin sections of *Myc*⁺/⁺ and *Myc*⁻/⁻ mice showing tdTomato⁺ Gata6lin⁺ cells. Scale bars, 200 μm (overview) and 40 μm (magnification). Representative images from *n* = 3 independent experiments. **j**, Quantification of the number of *Lrig1* RNA molecules per HF. Data are the mean ± s.d. *n* = 3 mice per group. Two-tailed Student's unpaired *t*-test was used to determine statistical significance in **b**, **c**, **e**–**h** and **j**.

of Lrig1⁺Gata6lin⁺ cells in the HF and Gata6lin⁺ cells in the IFE was significantly reduced by loss of Myc (Fig. 5k–m). Myc deletion also impaired activation of YAP and MAL in Gata6lin⁺ cells (Fig. 5l,o and Extended Data Fig. 5l,m), suggesting that Myc is required for mechanotransduction associated with dedifferentiation. Hydrogel-expanded control

epidermis showed an increase in Ki67⁺ Gata6lin⁺ cells compared with untreated mice (Extended Data Fig. 5n,o). However, Myc deletion led to a reduced number of proliferative Gata6lin⁺ cells (Extended Data Fig. 5n,o), indicating that Myc is necessary for Gata6lin⁺ proliferation upon hydrogel injection.

Altogether, we provide evidence that in three different contexts (wound healing, RA treatment and mechanical expansion), Myc is required for dedifferentiation of Gata6+ epidermal cells.

### Mechanical response associated with dedifferentiation

There is a growing appreciation that physical properties are a key regulatory component of the stem cell niche[30], and we have recently reported that stiffness of epidermal cells is modulated by their neighbours and the topography of the basal layer[31]. We therefore investigated whether wounding affects the physical properties of cells in the upper HFs, where Gata6 lineage cells reside. Atomic force microscopy (AFM) has been used to assess the mechanical properties of mouse epidermis[32] and human HFs[33] in histological sections. We measured epidermal stiffness in the upper HF, excluding hair shafts and dermis from the measurements. Distal HFs of $Myc^{-/-}$ mice were stiffer than those of wild-type mice (Fig. 6a–c). Proximal HFs in wild-type mice were stiffer than distal HFs, whereas in $Myc^{-/-}$ mice there was no significant difference (Fig. 6a–c). Thus, the difference in stiffness between upper distal and proximal HF found in wild-type skin was abolished on deletion of Myc, indicating that Myc is required for the change in upper HF stiffness induced by injury.

We also flow sorted Gata6lin+ cells from $Myc^{+/+}$ and $Myc^{-/-}$ mice, allowed them to adhere and spread and then analyzed them by AFM at single-cell resolution (Fig. 6d). This allowed us to rule out potential effects of the extracellular matrix or neighbouring cells in skin sections. Consistent with the results observed in distal HF, Myc-depleted cells showed an increased stiffness (Fig. 6d). This correlated with a decrease in cell area and cortical actin (Fig. 6e–g), in agreement with reports that Myc knockout reduces epidermal cell size[34] and that this correlates with stiffer substrates[35].

The keratin cytoskeleton is known to modulate cell stiffness and keratin-free cells are more deformable than cells with depolymerized actin[36]. Deletion of Myc in the Gata6 lineage resulted in changes in the keratin cytoskeleton, as judged by expression of Krt10 (Extended Data Fig. 6a,b). Additionally, expression of the suprabasal uHF marker $Cst6$ in the basal layer indicated premature differentiation (Fig. 6h,i), consistent with an earlier report that Myc deletion induces differentiation[34]. The spinous and granular layers of the IFE are stiffer than the basal layer[32], and so premature differentiation and associated changes in the keratin cytoskeleton could potentially explain the increase in stiffness in Myc-depleted keratinocytes we observed. This is consistent with a report showing that keratin networks regulate actin re-organization[37].

Quantitation of F-actin revealed a higher number of filamentous structures and lower lacunarity in wound proximal than distal upper HFs (Fig. 7a–c), reflecting increased F-actin polymerization and stress fibres[38]. The actin cytoskeleton was disrupted in Myc-depleted Gata6lin+ HFs, which correlated with a reduced number of filaments and higher lacunarity (Fig. 7a–c). Similar effects of Myc deletion were observed in RA-treated (Extended Data Fig. 6c–e) and hydrogel-treated

skin (Extended Data Fig. 6f–h), indicating that in three different contexts actin remodelling is associated with dedifferentiation of Gata6lin+ cells and is dependent on Myc expression.

To assess the effect of Myc on contractility of Gata6lin+ cells following wounding, we analyzed the levels of myosin-II phosphorylation by immunofluorescence labelling. Myc-depleted cells in HFs proximal to wounds showed reduced levels of myosin-II phosphorylation compared with control cells, indicating reduced contractility[39] (Fig. 7d,e). We observed a similar phenotype in hydrogel-injected mice, where Myc impairment prevented the activation of pMLC2 caused by hydrogels (Fig. 7f,g). This is consistent with the evidence that hydrogels cause mechanical changes in the epidermal layer that in Gata6lin+ cells are prevented by Myc deletion (Fig. 5a,b,n,o and Extended Data Fig. 5l,m).

It has been reported that regulators of epidermal cell stretching are upregulated during wound healing and that cell stretching induces actomyosin cytoskeleton remodelling[29]. This led us to predict that actomyosin regulators would be upregulated in dedifferentiating Gata6lin+ cells. We found that the small GTPases $Rac1$, $RhoA$ and $Cdc42$, as well as other actomyosin regulators, such as $Arpc4$ and $Cnbp$, were upregulated in dedifferentiating cells (Fig. 7h and Extended Data Fig. 6i). This is in agreement with the finding that Gata6 regulates genes involved in processes such as tube development and cell motility[14]. Moreover, Myc is known to regulate many genes involved in keratinocyte adhesion[40]. Consistent with this, cell adhesion genes $Lad1$, $Itgb1$ and $Tln1$, upregulated in stretched-mediated skin expansion[29], were also upregulated in dedifferentiating cells (Fig. 7i).

Together our data demonstrate that epidermal cell stretching and dedifferentiation share common transcriptional profiles and that Myc regulates cell contractility associated with dedifferentiation.

## Discussion

Here we have obtained insights into the molecular mechanisms underlying epidermal dedifferentiation. Transcriptionally, Gata6lin+ cells that had undergone dedifferentiation to Lrig1+ stem cells were indistinguishable from Gata6lin−Lrig1+ cells. Dedifferentiation occurred via reversal of the normal differentiation trajectory (Fig. 2g), contrasting with recent findings that reprogramming of induced pluripotent stem cells to naïve pluripotency is not a reversion of the developmental pathway[41].

So far, the Gata6 lineage of the SD is the only differentiated epidermal subpopulation that is known to revert to stem cells in vivo. The fact that dedifferentiation is not detected in lineage-traced and photo-labelled suprabasal IFE cells upon injury[42–44] probably reflects the unique characteristics of the duct microenvironment. It is also possible that the onset of dedifferentiation is more rapid in the SD than the IFE[44]. Our findings, however, are consistent with evidence of dedifferentiation by cultured keratinocytes on transplantation[45], and the ability of Blimp1+ differentiated sebocytes to dedifferentiate and initiate self-renewal in culture[46].

**Fig. 5 | Myc-dependent hydrogel-induced dedifferentiation. a**, Skin sections of control and hydrogel-injected mice stained for YAP (top) and MAL (bottom). Scale bar, 20 μm. **b**, Percentage of cells showing nuclear YAP or MAL (N > C), even distribution of YAP or MAL (N = C) and cytoplasmic YAP or MAL (N < C). Data are the mean ± s.d. $n$ = 3 mice. **c**, Epidermal sections of control and hydrogel-injected mice stained for Ivl and Krt14. Scale bar, 10 μm. **d**, Bar graph showing IFE thickness (μm) of control and hydrogel-injected mice. Data are the mean ± s.d. $n$ = 4 mice. **e**, Representative skin section showing position of the injected hydrogel (dashed line). Gata6lin+ cells and DAPI nuclear labelling are shown. Scale bar, 500 μm. Boxed region shows magnified HF stained for Lrig1. Scale bar, 40 μm. $n$ = 3 independent experiments. **f–h**, Bar graphs showing the percentage of suprabasal and basal Gata6lin+ cells (**f**), the percentage of Gata6lin+ cells in IFE (**g**) and the percentage of Gata6lin+ cells expressing Lrig1 per HF (**h**) in control, hydrogel-injected and bleomycin-treated mice. Data are the mean ± s.d. $n$ = 3

mice. **i**, Skin sections showing tdTomato Gata6lin+ cells stained for CD45. Scale bar, 40 μm. **j**, Bar graph showing CD45 fluorescence intensity (a.u.) in hydrogel and bleomycin-treated mice. Data are the mean ± s.d. $n$ = 3 mice (day 2, day 7, bleomycin); $n$ = 4 mice (control, day 1, day 4, PBS). **k**, Skin sections of $Myc^{+/+}$ and $Myc^{-/-}$ hydrogel-injected mice showing tdTomato Gata6lin+ cells stained for Lrig1. Scale bar, 40 μm. **l,m**, Percentage of Gata6lin+ cells in the IFE (**l**) and Gata6lin+ cells expressing Lrig1 per HF (**m**). Data are the mean ± s.d. $n$ = 3 mice. **n**, Skin sections of $Myc^{+/+}$ and $Myc^{-/-}$ hydrogel-injected mice showing tdTomato Gata6lin+ cells stained for YAP. Scale bar, 20 μm. **o**, Percentage of cells showing nuclear YAP (N > C), even distribution of YAP (N = C) and cytoplasmic YAP (N < C). Data are the mean ± s.d. $n$ = 3 mice. Two-tailed Student's unpaired $t$-test was used to determine statistical significance in **b**, **l**, **m** and **o**. One-way ANOVA with Šidák's multiple comparisons test was used to determine statistical significance in **d**, **g**, **h** and **j**.

We identified the master TF Myc as a driver of dedifferentiation. Effects of Myc expression on undifferentiated epidermal cells are dose and context dependent[22,47]. Myc expression varied along the pseudotime trajectory from cluster 3 to cluster 1 (Fig. 3b). Thus, we speculate that, upon damage, Myc has a dual role in Gata-6lin+ cells: low levels stimulate dedifferentiation and proliferation

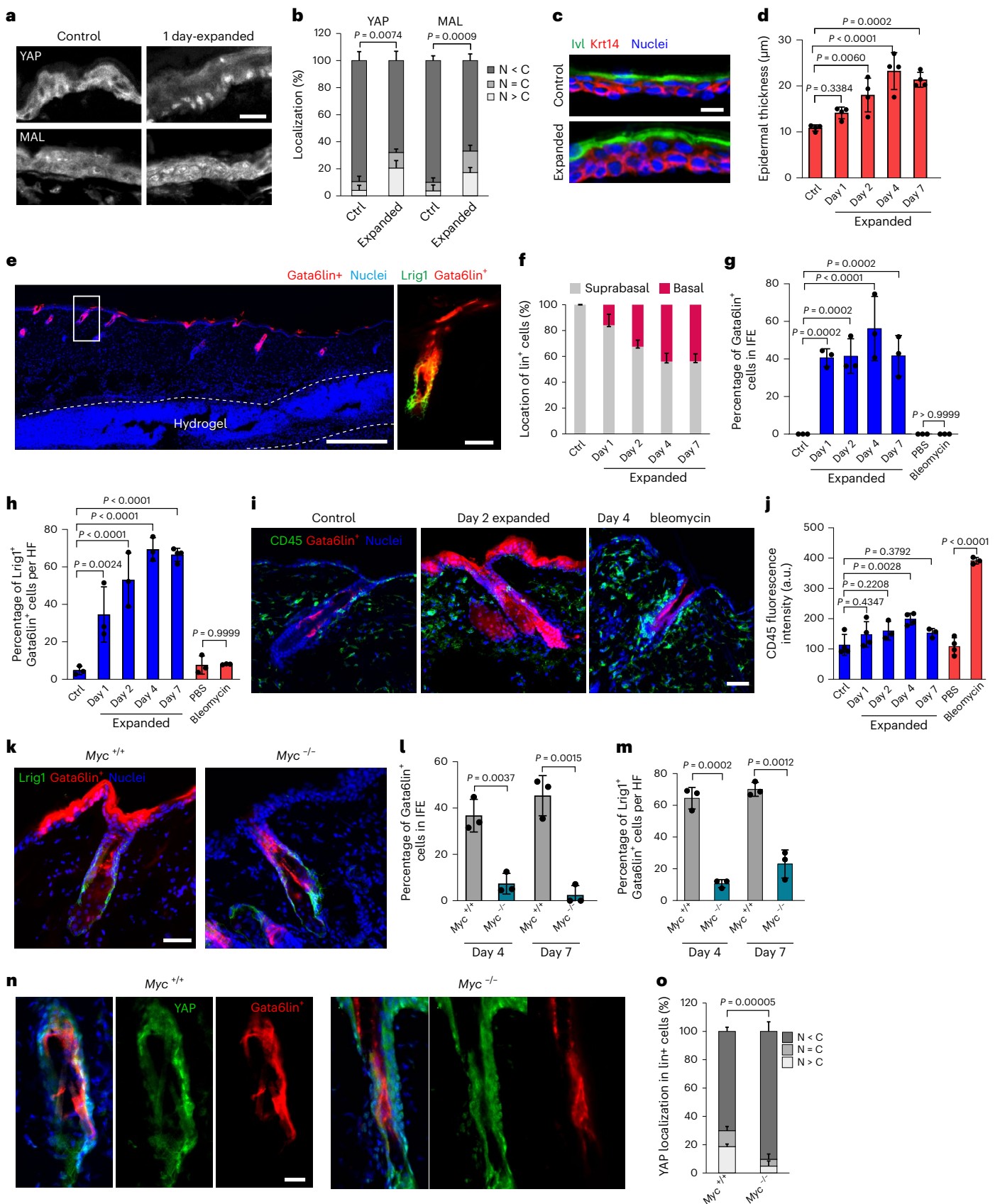

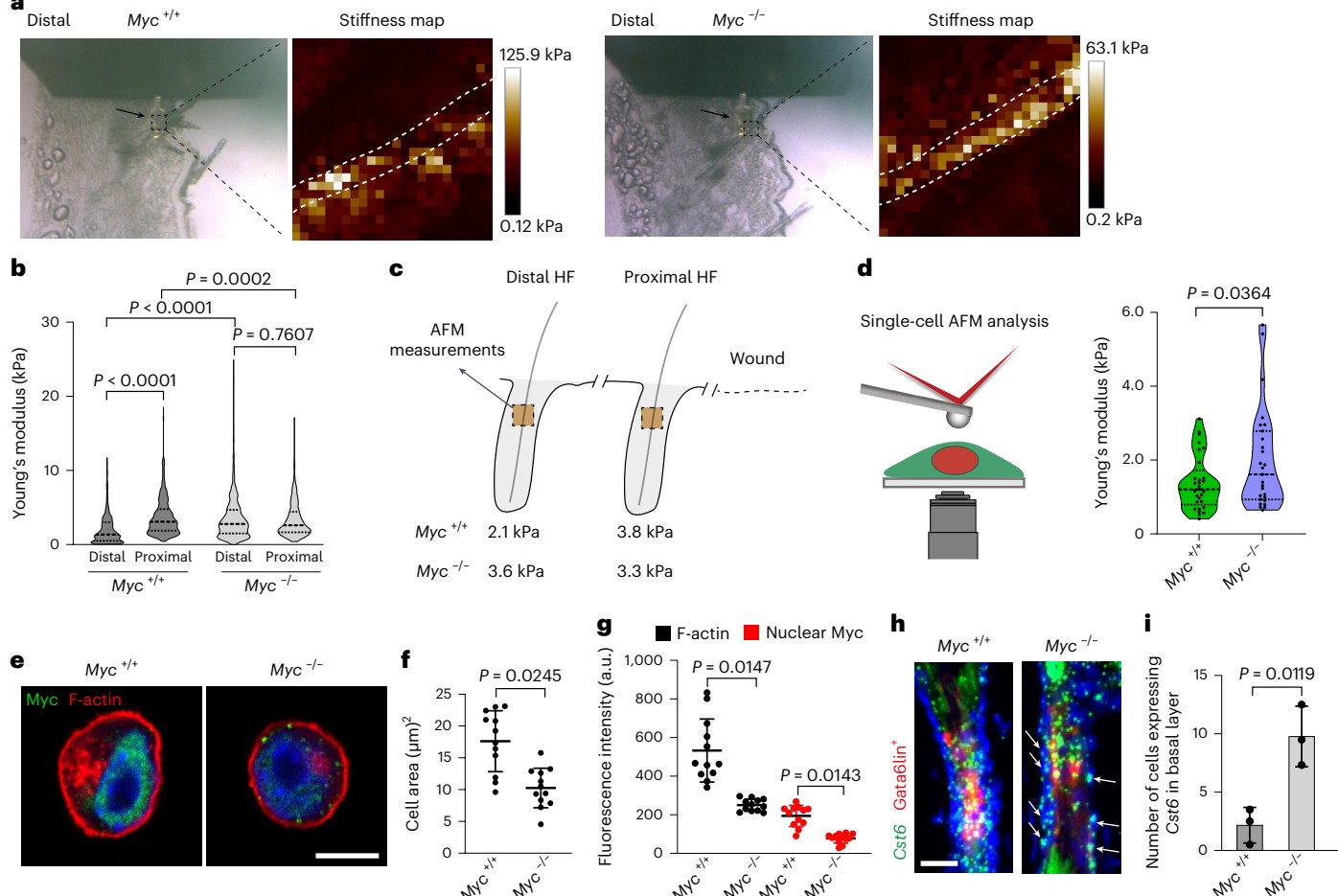

**Fig. 6 | Effect of Myc deletion on cell stiffness and the actin cytoskeleton.**
**a**, Representative brightfield images and Young's modulus maps of HFs distal to wounds in *Myc*[+/+] and *Myc*[−/−] mice. Boxed regions show the upper HF regions that were measured by AFM. Each square corresponds to the Young's modulus obtained from one force–distance curve. The dashed lines mark hair shafts. The arrows indicate the tip of the cantilever. Representative images from *n* = 3 independent experiments. **b**, Epidermal cell stiffness was analyzed in HFs by AFM. Violin plots show Young's modulus (kPa) in distal and proximal HFs of *Myc*[+/+] and *Myc*[−/−] mice (deleted via the Gata6 promoter). A total of 150 measurements were analyzed per region. At least 18 HFs (9 distal and 9 proximal) from 3–4 mice per group were analyzed. **c**, Schematic showing Young's modulus means (kPa) obtained in **b**. Note that in the absence of Myc there is no difference in stiffness between distal and proximal HFs. **d**, Schematic of single-cell AFM analysis

and violin plots showing Young's modulus (kPa) measurements of *Myc*[+/+] and *Myc*[−/−] keratinocytes. A total of 29 *Myc*[+/+] cells and 27 *Myc*[−/−] cells from three independent experiments were analyzed. **e**, Cells stained with phalloidin (red), anti-Myc (green) and DAPI (blue). Scale bar, 10 μm. **f,g**, Bar graphs show cell area (μm²) (**f**) and fluorescence intensity of F-actin and nuclear Myc (**g**) in *Myc*[+/+] and *Myc*[−/−] keratinocytes. *n* = 12 cells per condition from 2 independent experiments. **h**, Detection of *Cst6* by mRNA in situ hybridization in HFs of *Myc*[+/+] and *Myc*[−/−] mice showing tdTomato Gata6lin[+] cells. Arrows indicate cells in the basal layer expressing *Cst6*. Scale bar, 20 μm. **i**, Quantification of the number of cells per HF expressing *Cst6* in the basal layer. Data are the mean ± s.d. *n* = 3 mice per group. Two-tailed Mann–Whitney test was used to determine statistical significance in **b**. Two-tailed Student's unpaired *t*-test was used to determine statistical significance in **d**, **f**, **g** and **i**.

while high levels promote subsequent differentiation in the HF and the IFE.

The interplay between Myc and Gata6 is supported by other studies. In embryonic stem cells, Myc suppresses Gata6, impairing endoderm differentiation while favouring pluripotency[48]. In addition, Myc is an oncogene, and Gata6 acts as a tumour suppressor in sebaceous carcinogenesis[49].

Hydrogel-mediated epidermal expansion had a more pronounced effect on dedifferentiation than wound healing, since basal lin[+] cells were detected 1 day after implantation (Fig. 5f), compared with 9 days after wounding (Extended Data Fig. 1b,c). Yap, MAL and pMLC2 were activated in Gata6lin[+] cells after hydrogel injection, indicating that the SD responds to mechanical cues. Myc deletion abrogated this response, in line with reports that Myc and Yap cooperate to integrate mechanical and mitogenic signals[50]. The exact mechanisms by which mechanical forces are transmitted to the Gata6 population to induce

Myc activation remain unclear. HF orientation, re-arrangement of cell junctions or changes in cell stiffness could potentially be involved. The extent to which mechano-signalling contributes to dedifferentiation compared with damage signalling pathways also requires further investigation.

We propose that physical space availability is a key feature of dedifferentiation. This would explain why when Lgr5[+] lower bulge cells are ablated, they can be replaced by upper bulge cells[51]. Similarly, when bulge HF stem cells are laser ablated, cells at the junctional zone can repopulate the bulge[52]. Following injury, Lrig1lin[+] cells reach the IFE before Gata6lin[+] cells[14]. Therefore, it is plausible that during wound healing the Lrig1[+] cells that migrate into the IFE leave space for Gata6lin[+] cells to migrate downward within the uHF, providing a reservoir that fuels the HF with Lrig1[+] cells that subsequently migrate into the IFE. In undamaged epidermis the cells are tightly packed and constrained by extensive cell–cell and cell–extracellular matrix adhesions.

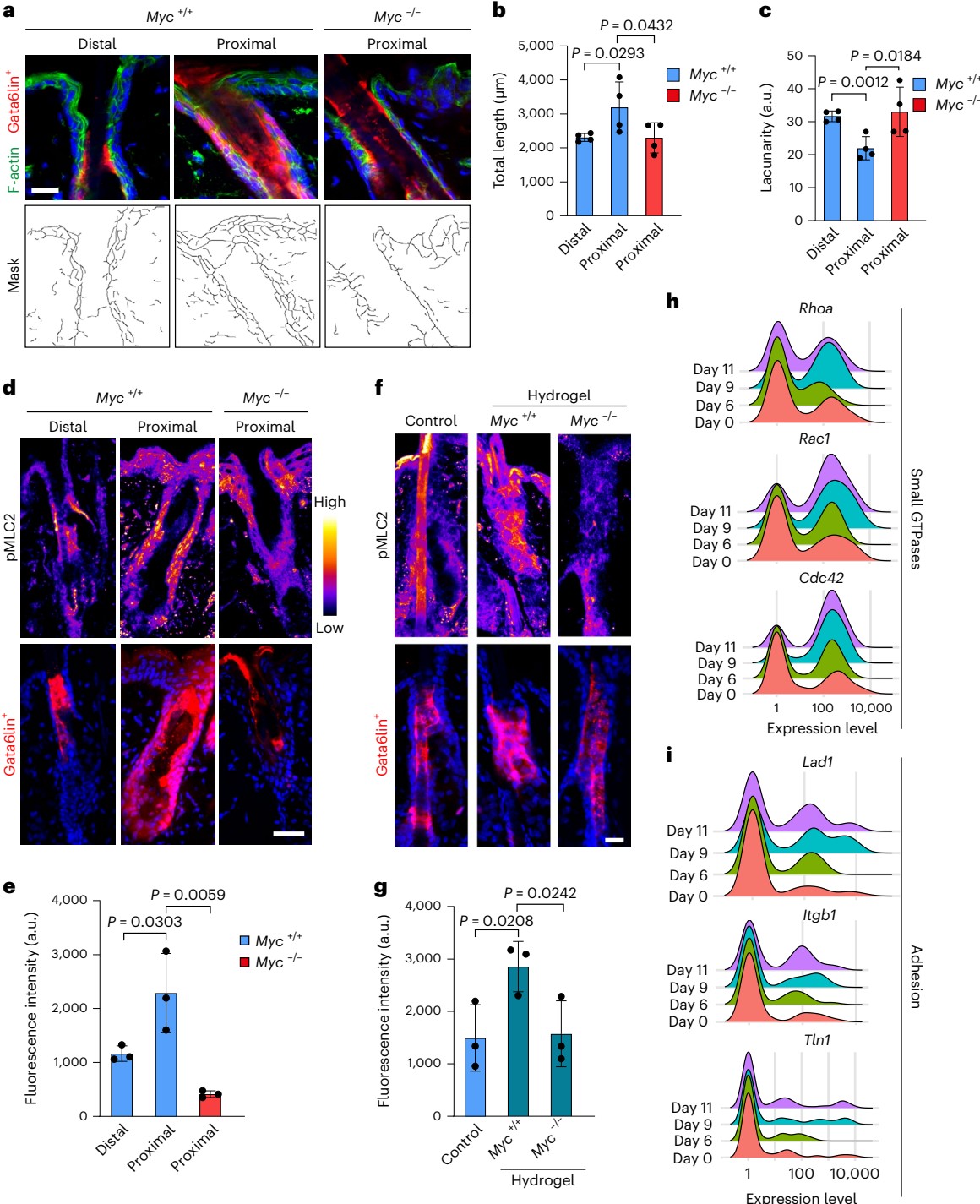

**Fig. 7 | Myc-dependent actin network remodelling and cell contractility.**
**a**, Top: skin sections of HFs distal and proximal to wounds showing tdTomato Gata6lin⁺ cells stained with phalloidin (green) and DAPI (blue). Bottom: masks were obtained using the ImageJ plugin TWOMBLI. Scale bar, 20 µm. Representative images from *n* = 4 independent experiments. **b,c**, Bar graphs show total length (µm) (**b**) and lacunarity (a.u.) (**c**) obtained in distal and proximal HFs of *Myc*⁺/⁺ mice and proximal HFs of *Myc*⁻/⁻ mice. Data are the mean ± s.d. *n* = 4 mice per condition. **d–g**, Wounded skin sections (**d**) and skin sections of hydrogel-injected mice (**f**) showing tdTomato Gata6lin⁺ cells were stained for pMLC2 and colour-coded for signal intensity with ImageJ. Bar graphs

show fluorescence intensity of pMLC2 in wounded (**e**) and hydrogel-injected (**g**) epidermis of *Myc*⁺/⁺ and *Myc*⁻/⁻ mice. The hair shaft was excluded from the analysis and only areas containing tdTomato⁺ cells were measured. Data are the mean ± s.d. *n* = 3 mice per condition. Scale bars, 20 µm. **h,i**, Ridgeline plots showing expression of the small GTPases *RhoA*, *Rac1* and *Cdc42* (**h**), and the adhesion markers *Lad1*, *Itgb1* and *Tln1* (**i**) at days 0, 6, 9 and 11 post-wounding. Data are from two independent biological replicates per timepoint. All Gata6lin⁺ cells from the scRNA-seq data were analyzed. Two-tailed Student's unpaired *t*-test was used to determine statistical significance in **b**, **c**, **e** and **g**.

During durotaxis cell migration is directed via stiffness gradients. Durotaxis in vivo is understudied[53]. Following wounding there is a difference in stiffness between distal and proximal HFs that is lost on Myc deletion (Fig. 6b). This suggests that durotaxis may occur during epidermal wound healing and that Myc is required to generate dynamic stiffness gradients. The fact that Gata6lin+ cells lacking Myc do not respond to stiffness stimuli is indicative of the mechanical properties of the SD. Since the cytoplasm of epidermal cells is relatively small, the mechanical response observed upon wounding is probably dominated by nuclear rigidity. Further AFM studies will be required to determine the extent to which Myc affects cytoplasmic and nuclear stiffness.

Our findings are consistent with increased cell contractility and stiffness positively regulating epithelial cell migration during wound healing[54], with the abundance of stress fibres regulating cell stiffness[55] and the finding that increased substrate stiffness induces Myc[56]. Mechanical stress regulates stem cell differentiation in *Drosophila*[57] and stretching affects the balance between epidermal renewal and differentiation[29]. Changes in the actin cytoskeleton can prevent the nucleus from deformation[58], leading us to speculate that mechanical factors upregulate Myc, which is in turn involved in actin network remodelling linked to dedifferentiation. Given the widespread expression of Myc in different tissues, the mechanism of dedifferentiation we have uncovered in the epidermis may turn out to be a general phenomenon.

## Online content

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

## Methods

The research in this manuscript is in compliance with relevant ethical regulations.

### Mouse strains

All mouse procedures were subjected to local ethical approval at King's College London (UK) and performed under a UK Government Home Office licence (PP70/8474 or PP0313918). Rosa26-fl/STOP/fl-tdTomato43 (ref. 59), CAGGS eGFP mice[60,61] (The Jackson Laboratory, #003291), and Gata6EGFPCreERT2 mice[14] have been described previously. Mice in which Myc was knocked out via the Gata6 promoter mice were generated by crossbreeding *c-Myc*[fl/fl] mice[62] (The Jackson Laboratory, #032046) with Gata6EFGPCreERT2 Rosa26-fl/STOP/fl-tdTomato43 mice. All mice used in the experiments were on C57BL/6J or mixed backgrounds. Experiments were carried out with male and female mice. No gender-specific differences were observed. For lineage-tracing experiments, 8–12-week-old mice received a topical dose of 4-hydroxy-tamoxifen (Sigma): 1 mg dissolved in 100 µl acetone applied to clipped back skin. To achieve *Myc* knockout (*Myc*[−/−]) in cells where epidermal *Gata6* was expressed and visualize Gata6+ cells, *c-Myc*[fl/fl] Gata6EFGPCreERT2 Rosa26-fl/STOP/fl-tdTomato43 mice were treated topically with three doses of 4-hydroxy-tamoxifen every second day. *Myc*[+/+] littermates were used as controls. All mice were housed under specific pathogen-free conditions in individually ventilated cages and a 12 h light–dark cycle. Room temperature was maintained at 22 ± 2 °C with 30–70% humidity. The investigators were not blinded to allocation during experiments and outcome assessment.

### Full-thickness skin wound and RA treatment

One week post tamoxifen treatment (day 0) full-thickness wounds were made with a circular biopsy punch (Stiefel) in dorsal (6 mm) skin under analgesia and anaesthesia. Smaller wounds (2 mm) were made for assessment of the early timepoints of wound healing, as this allows a faster closure (Extended Data Fig. 2d,e). To investigate the effects of RA, we treated the clipped back skin of 8–12-week-old mice in the telogen phase of the hair cycle with 200 µl of 0.5 mM all-*trans*-RA (Sigma) dissolved in acetone. Treatment was carried out daily for 5 days. Age-matched mice treated with acetone alone were used as controls.

### Hydrogel and bleomycin treatments

Eight- to 12-week-old mice were subcutaneously injected with a methylcellulose-based gel (4% methylcellulose and 4.5% ammonium sulfate in phosphate-buffered saline (PBS) mixed 1:1 with protamine sulfate (0.25 mg ml$^{-1}$), which forms a gel at body temperature[63]. A 27-gauge needle was used to inject the mice (100 µl). Tissue was collected 1, 2, 4 and 7 days after injection. For bleomycin-induced inflammation, bleomycin was dissolved in PBS at a concentration of 0.05 mg ml$^{-1}$ and injected subcutaneously into the back skin using a 27-gauge needle at a volume of 100 µl for 4 consecutive days. Inflammation was analyzed by CD45 immunostaining and *Tnfa* and *Ccl2* expression.

### Immunofluorescence staining and histology

Mouse tissue samples were embedded in optimal cutting temperature compound (OCT) (Life Technologies), processed and stained by conventional methods[64,65]. Frozen sections were fixed in 4% paraformaldehyde/PBS pH 7.4, permeabilized with 0.1% Triton X-100/PBS for 10 min at room temperature and blocked for 1 h at room temperature with immunofluorescence buffer containing 10% foetal bovine serum (Sigma-Aldrich), 3% bovine serum albumin (Sigma-Aldrich), 0.02% fish skin gelatin (Sigma-Aldrich), 0.05% Triton X-100, and 0.05% Tween 20 (Sigma-Aldrich) in PBS. Sections were then incubated overnight at 4 °C with the following primary antibodies: Krt14 (1:1,000, LL002 clone, Abcam ab7800 and 1:1,000, Covance SIG-3476); Itga6 (1:200, GoH3 clone, eBioscience 14-0495-81); Myc (1:100, Abcam ab32072); Ki67 (1:50, Novus Biologicals NB600-1252); Lrig1 (1:200, R&D Systems AF3688);

Involucrin (1:500, ERLI-3 clone, in-house); CD45 (1:200, 30-F11 clone, eBioscience 14-0451-82); pMLC (1:100, Cell Signaling 3674); YAP (1:200, Cell Signaling 14074); MAL (1:200, Proteintech 21166-1-AP). Samples were labelled for 1 h at room temperature with the following secondary antibodies (1:500, Invitrogen): Donkey anti-Rabbit IgG Alexa Fluor 647 (A32795); Donkey anti-Rat IgG Alexa Fluor 647 (A48272); Donkey anti-Goat IgG Alexa Fluor 488 (A48272); Goat anti-Chicken Alexa Fluor 488 (A21449); Goat anti-Rabbit IgG Alexa Fluor 488 (A11034). Alexa 488 phalloidin (1:500, Invitrogen A12379) was used to label F-actin. 4',6-Diamidino-2-phenylindole dihydrochloride (DAPI, Thermo Fisher Scientific) was used as a nuclear counterstain. Sections were mounted with ProLong Gold Antifade Mountant (Thermo Fisher). Images were obtained using a Nikon A1 or a Nikon A1R confocal microscope (Nikon) with 20× or 40× objectives and analyzed with ImageJ (National Institutes of Health, v1.53). To quantify the position of Gata6lin+ cells in HFs and in the wound bed after wounding, serial sections of half of each wound bed were examined. For haematoxylin and eosin staining, tissues were embedded in OCT, sectioned and post-fixed in 4% paraformaldehyde/PBS pH 7.4 for 10 min before staining by conventional methods. Images were acquired using a NanoZoomer 2.0RS Digital Slide Scanner (Hamamatsu Photonics K.K.) and analyzed using NPD viewer software (Hamamatsu, v2.7.43).

### Flow cytometry

Single keratinocytes from wounded and unwounded skin of Gata6EFGPCreERT2 Rosa26-fl/STOP/fl-tdTomato43 mice were isolated by flow sorting 6, 9 and 11 days after wounding, essentially as previously described[66], and labelled with anti-CD45 APC (1:500, 30-F11 clone, eBioscience 17-0451-82) and anti-CD49f PE/Cyanine7 (Itga6) (1:500, BioLegend, clone GoH3 313611). Given the small number of cells that could be collected from individual wounds, samples were buffered by mixing with epidermal cells from CAGGS eGFP mice in which GFP is expressed in all cells via the CMV-β-actin promoter[61]. GFP+ epidermal cells were subsequently removed by sorting (Extended Data Fig. 1d). Cell sorting was performed on the BD FACSAria II cell sorter. For gate setting and compensation, unlabelled and single-labelled cells were used as controls. Data were analyzed using FlowJo software (v10.7.2). For single-cell sequencing, ten mice were used from two independent experiments per timepoint ($n$ = 2 mice (day 9, one per replicate); $n$ = 2 mice (day 11, one per replicate); $n$ = 6 mice (day 6, 3 per replicate)). Control samples ($n$ = 2 mice) were taken from unwounded skin of mice used for the day 9 and 11 timepoints.

### Live imaging of wound healing

Two-photon excitation microscopy was performed with a Nikon A1RMP upright microscope, equipped with a 25×/1.10 water-immersion objective lens (CF175 Apo LWD 25XW Nikon) and a Ti:sapphire laser (0.95 W at 900 nm) (Coherent Chameleon II laser). The laser power used for observation was 2–10%. Scan speed was 4 µs per pixel. The excitation wavelength was 770 nm for second harmonic generation, Hoechst 33342 and tdTomato signals. An IR-cut filter, IR-DM, was used. Hoechst 33342 and collagen (autofluorescence via second harmonic generation) signals were detected by 492 SP (Chroma); the tdTomato signal was detected by 575/25 (Chroma). *Z*-stack images were acquired with a view field of 0.257 mm$^2$ in 5 µm steps. Mice aged 9–12 weeks were anaesthetized throughout imaging by inhalation of vapourized 1.5% isoflurane (MERIAL) and placed in the prone position on a heating pad. The ear was stabilized between a cover glass and a thermal conductive soft silicon sheet as previously described[67]. For nuclear staining, mice were injected with 200 µl of Hoechst 33342 (Molecular Probes) dissolved in PBS at 2 mg ml$^{-1}$ via the tail vein 1 h before imaging. To image wound healing, 2 days after tamoxifen treatment, hairs were removed from mouse ear skin with depilation cream 1 h before wounding. Ear skin wounds were created with a mini router No. 28600 (Kisopowertool MFG). The wound healing process was imaged 3 days after wounding.

Time-lapse images were acquired every 1 h. A total of three mice were examined. Acquired images were analyzed with ImageJ. Orthogonal images (Extended Data Fig. 4a) were reconstructed from Z-stack images with 20 pixels of Linewidth.

## Cell culture

Spontaneously immortalized keratinocytes isolated from K14My-cER transgenic mouse founder line 2184C.1 (ref. 23) were cultured in calcium-free complete FAD medium comprising one part Ham's F12, three parts Dulbecco's modified Eagle medium and $10^{-4}$ M adenine (Gibco) supplemented with 10% foetal bovine serum (Gibco), 1% penicillin/streptomycin (Gibco), 1% L-glutamine (Gibco), 10 ng ml$^{-1}$ epidermal growth factor (Peprotech), 0.5 µg ml$^{-1}$ hydrocortisone (Sigma-Aldrich), 5 µg ml$^{-1}$ insulin (Sigma-Aldrich) and $10^{-10}$ M cholera enterotoxin (Sigma-Aldrich) as described previously[68]. Keratinocytes were maintained on mitotically inactivated 3T3-J2 cells pre-treated with 4 µg ml$^{-1}$ mitomycin (Sigma-Aldrich) for 2–3 h unless otherwise specified. 3T3-J2 cells were cultured in high-glucose Dulbecco's modified Eagle medium (Sigma-Aldrich) with 10% adult bovine serum (Thermo Fisher Scientific) and 1% penicillin/streptomycin (Gibco). 3T3-J2 cells were originally obtained from Dr James Rheinwald (Department of Dermatology, Harvard Skin Research Center, the United States). Lentivirus-mediated transduction was performed as described previously[69]. The K14MycER keratinocyte line was infected with GATA6 lentivirus (pCW57–GFP–2A-GATA6) containing human GATA6 open reading frame (National Center for Biotechnology Information reference sequence NM_005257.5). pCW57–GFP–2A-MCS was used for mock lentiviral infection. The plasmids and vector maps are detailed in ref. 69. Keratinocyte stocks were maintained without 4-hydroxytamoxifen (Sigma-Aldrich) and/or doxycycline (Sigma-Aldrich). All cell stocks were routinely tested for mycoplasma contamination and found to be negative. For experiments cells were seeded on plates pre-coated with 20 µg ml$^{-1}$ rat-tail collagen type I in PBS (BD Biosciences). A total of 2 µg ml$^{-1}$ doxycycline (diluted in sterile water) was added to the growth medium to induce expression of *GATA6*, while 25 nM 4-OHT (diluted in ethanol) was used to induce *Myc* expression.

## Colony formation assay

To determine colony forming efficiency, 1,000 keratinocytes were seeded on mitotically inactivated 3T3-J2 cells per well in triplicate wells of six-well plates. After 12 days, feeders were removed and colonies were fixed in 4% paraformaldehyde (Sigma-Aldrich) for 10 min, then stained with 1% Rhodanile Blue (1:1 mixture of Rhodamine B and Nile Blue chloride) (Sigma-Aldrich).

## Real-time qRT–PCR

Total RNA extraction from cultured cells and complementary DNA synthesis was performed using Qiagen RNeasy Mini Kit (Qiagen) and the QuantiTect Reverse Transcription kit (Qiagen) respectively, according to the manufacturer's instructions. Quantitative real-time reverse transcriptase polymerase chain reactions (qRT–PCR) were performed on a CFX384 Real-Time System (Bio-Rad Laboratories) using TaqMan Fast Universal PCR Master Mix (Life Technologies). Values were normalized to housekeeping genes (*Rn18*, *Gapdh* and/or *Tbp*) and relative quantification of gene expression was performed using either the $2^{-\Delta Ct}$ or $2^{-\Delta\Delta Ct}$ method. For each biological replicate, the reaction was performed in technical duplicates. Primers used in this study are listed in Supplementary Table 3.

## Statistics and reproducibility

Statistical analysis was performed with GraphPad Prism software (v9.5.1). Unless stated otherwise, data are expressed as the mean ± standard deviation (s.d.) of at least three independent experiments. Statistical significance was determined with the two-tailed Student's unpaired *t*-test, ordinary one-way analysis of variance (ANOVA) with Šidák's multiple comparisons test, or Mann–Whitney test. No statistical methods were used to pre-determine sample sizes, but our sample sizes are similar to those reported in previous publications[6,14,29]. No data points were excluded. Data distribution was assumed to be normal, but this was not formally tested. No randomization was done. Mice were categorized on the basis of genotype. To quantify the fluorescence intensity of nuclear Myc in an unbiased manner, we used DAPI as a nuclear mask and quantified the nuclear intensity of Myc in Gata6lin$^+$ cells and lin$^-$ cells. To quantify CD45 staining in skin sections, mean fluorescence was determined with ImageJ software and normalized to background. To quantify pMLC, ImageJ software was used to measure the integrated intensity. For calculation of epidermal thickness, we took measurements in the IFE and in the upper HF using Krt14 and Ivl as basal and suprabasal layer markers, respectively. In the case of haematoxylin-and-eosin-stained sections, we only measured epidermal thickness in regions that were vertically oriented as judged by the thickness of the basement membrane. Data collection and analysis were not performed blind to the conditions of the experiments.

## Library preparation and RNA sequencing

For single-cell sequencing, cells were sorted into individual wells of a 96-well plate containing 2 µl lysis buffer (0.8% (volume/volume), Triton X-100 and 2 U µl$^{-1}$ recombinant RNase inhibitor (Clontech). Library preparation was performed with SmartSeq2 followed by the Nextera XT protocol (Illumina). Sequencing was performed on the Illumina HiSeq4000 75 PE. Two biological replicates per timepoint were analyzed, and batch-corrected (Extended Data Fig. 1g). In total, 684 cells were analyzed by scRNA-seq.

## Alignment, quantification and quality control of scRNA-seq data

Smart-seq2 sequencing data were aligned with STAR (version 2.2), using the STAR index and aligned to the GRCm38 reference genome. Gene-specific counts were calculated using the featureCounts method from Rsubread (version 3.7)[70] with mm10 RefSeq annotation, and analyzed with Seurat version 4.1.1 (principal component analysis (PCA), Cluster, tSNE and UMAP)[71]. In the standard pre-processing workflow of Seurat, we selected 5,000 variable genes for PCA. Then we performed cell clustering and UMAP. The top 15 principal components (PCs) were used in cell clusters with the resolution parameter set at 0.5. Marker genes of each cell cluster/state were outputted for enrichment analysis using fgsea version 1.16.0 package in R, which were used to define the cell types. Cell clusters were annotated on the basis of cell types from Joost et al. (2016)[15]. For most plots we used ggplot2 (v3.3.6).

## Processing and normalization of data

All Gata6lin$^+$ and lin$^-$ cells from day 0 (control, unwounded skin), day 6, day 9 and day 11 wounds were selected. Two biological replicates from two independent batches per timepoint were analyzed. Cells with fewer than 200 genes were removed (684 cells remained). Unexpressed gene counts were removed, and the dataset was normalized according to size factors and log-transformed. For combined analysis of different batches, we performed batch correction using the Bayesian-based method ComBat from the sva R package. The corrected data were used for further downstream analysis. Marker genes were identified with log(fold change) and min.pct ≥0.25 as cut-off by performing differential gene expression analysis between the clusters using Wilcoxon rank sum test. To present high-dimensional data in two-dimensional space, we generated UMAP using the results of PCA with significant 15 PCs as input.

## Pseudotime and trajectory analysis

We performed pseudotemporal ordering of all lineage-positive cells based on previous methods[72]. Monocle 2 v2.18.0 and Monocle 3 v0.1.3 were used for pseudotime analysis. For Monocle 2, batch effect information was passed into the residualModelFormulaStr option in the

'reduceDimension' function. Pseudotemporal ordering was performed on Combat-batch corrected data. The corrected data were scaled using the ScaleData function, regressing out mitochondrial genes that were manually curated from the Mouse MitoCarta2.0 database and taking only the top 100 genes based on MCARTA2.0_score. Following that step, the data were used as an input for dimension reduction using PCA and UMAP, performed using the Seurat package. The top 15 PCs were used in UMAP with default parameters. We considered all genes that were highly variable using Seurat's FindVariableFeatures function to be pseudotime dependent. The UMAP space from the Seurat package was used as an input of the reduced dimensional function in Monocle 3.

## Cell fate transition analysis

We applied CellRank[73] (v1.5.1) to assess cell dynamics based on our transcription profiles and trajectory analysis. Like Monocle 2, Monocle 3, Diffusion pseudotime[74] and PAGA[75], CellRank makes the assumption that cell transitions take place in small steps. CellRank models these transitions using Markov chain and computes directed transition probabilities based on a $k$-nearest neighbour (KNN) graph and PseudotimeKernel. Computing cell fate using CellRank involves two steps. The first step is to initialize the PseudotimeKernel to create a computing transition matrix. In the second step, computing projections are visualized on UMAP. We used pseudotime generated by Monocle 3 to initialize PseudotimeKernel and calculated the transition matrix using the compute_transition_matrix function. Then we computed and projected these transitions on UMAP using compute_projection function. Finally, we visualized the transitions using scvelo's velocity_embedding_stream function.

## Transcriptional profile correlation between dedifferentiating and differentiating cells

Differentiating cells (Gata6lin⁻ cells expressing Lrig1 <1 from unwounded epidermis, $n$ = 32 cells) and dedifferentiating cells (Gata6lin⁺ cells expressing Lrig1 >1 from wounds, $n$ = 80 cells) were selected from the whole dataset and then subjected to selection of highly variable and differentially expressed genes between the two groups, as discussed in Results. Loess regression was used to understand the trend of gene expression correlation between differentiating and dedifferentiating cells. We obtained 545 genes (Supplementary Table 1) that were used to calculate the Loess regression on median expression values for both populations using the ggscatter function in ggpurb package in R with method=loess.

## Calculation of signature score of a gene set

For gene scoring analysis, gene sets were acquired from the MSigDB database, the Mouse Genome Informatics Gene Ontology Browser. Specific genes in each gene set related to cell identities are listed according to Joost et al. (2016)[15]. The TransferData function in Seurat R package was then used to calculate the anchors of each geneset in each cell. The two-sided Wilcoxon rank sum test was used to evaluate whether there were significant differences in the computed signature scores between two groups of cells.

## TFs controlling gene modules

After extracting the information related to the interaction of various mouse TFs and their effect on downstream or upstream genes, every gene module was subjected to comparison using the TRRUST database[17] to filter out the TFs controlling a particular gene module. Next, we arranged all the Gata6lin⁺ cells according to pseudotime from left to right, to show the expression of those TFs with a spatial signature.

## Ligand-receptor analysis

To gain a computational perspective on cell–cell interactions, nichenetr version 1.0.0 was used[76] to probe the intercellular influence on gene expression. The package not only predicts extracellular upstream regulators but also the affected target genes. The nichenetr pipeline was used to analyze the effect of the niche on dedifferentiation of Gata6lin⁺ cells. We assumed the surrounding cell population was the sender cells (Gata6lin⁻ cells) and those in the wound area (Gata6lin⁺ cells) were the target cells. First, we defined the genes expressed in both populations. Next, we selected the genes that were expressed (that is, receptors) in the Gata6lin⁺ wound cells. Then, we selected the set of ligands expressed in the surrounding Gata6lin⁻ cells that would bind the receptors. Finally, we ranked these ligand-receptor activities and selected the top predicted target genes for these ligands.

## AFM

AFM measurements were carried out using a Bioscope atomic force microscope (BioScope Resolve BioAFM, Bruker), coupled with an optical microscope (Nikon Eclipse Ti-U). Fresh-frozen skin sections of 10 μm thickness adhered to Superfrost microscope slides were used. Before measuring, the sections were washed with PBS three times to remove residual OCT. The measurements for each group ($Myc^{+/+}$ and $Myc^{-/-}$) included three HF distal and three HF proximal to the wound. A spherical nitride tip (5 μm) on a nitride cantilever with a nominal spring constant of 0.25 N m⁻¹ (SAA-SPH-5UM, Bruker) was used. Before the experiment, the deflection sensitivity of the AFM cantilever was individually calibrated. For each sample, an area of 40 μm × 40 μm was selected in the upper HF conducting 24 × 24 force–extension measurements to probe the stiffness of the region. Each measurement consisted of a 10 μm ramp to a maximum trigger force of 10 nN, approached/retracted at a velocity of 10 μm s⁻¹. The Young's modulus for each probed region was calculated by fitting the force–extension curves with a Hertz model for a spherical geometry, which allows calculation of the Young's modulus $E$, from the evolution of the force $F$, as a function of the indentation depth $\delta$ as:

$$F = \frac{4ER^{1/2}}{3(1 - v^2)}\delta^{3/2}$$

where $R$ is the radius of the spherical tip and $v = 0.5$ the Poisson's ratio. Only the region between 20% and 80% of the maximum force was employed for fitting. Only curves showing a clean approach/retraction pattern were selected. Measurements on the hair shaft were discarded (Fig. 6a). Typically, ~150 measurements were included for each area.

For single-cell measurements, Gata6lin⁺ cells from the epidermis of $Myc^{+/+}$ and $Myc^{-/-}$ mice were isolated by FACS as described in Extended Data Fig. 1d and seeded on glass-bottom dishes (40 mm aperture, Willco Wells) coated with collagen. A total of $1.5 \times 10^4$ to $2.5 \times 10^4$ cells per dish were plated. Two days later, the stiffness of the individual cells was probed by AFM, using the same instrument and procedure described above. Since the tip of the cantilever was similar to the size of the cells, the stiffness of the cytoplasm/nucleus could not be independently assessed, and our measurements are an average of the cytoplasm and nucleus. Data were analyzed following the same criteria as for the tissue sections. Artefactual force–extension curves, here typically consisting of substrate measurements exhibiting unphysically stiff values, were discarded.

## RNAscope multiplex fluorescent assay

Skin sections of 10 μm from OCT cryoblocks were fixed in 4% formalin and then analyzed by RNA hybridization using the RNAscope Multiplex Fluorescent Detection Kit v2 (ACDBio, cat. no. 323100), following the manufacturer's instructions. Probes against mouse *Lrig1* (ACDBio, reference: 310521), *Lgr5* (ACDBio, reference: 312171-C2), Cst6 (ACDBio, reference: 436181), *Lgr6* (ACDBio, reference: 404961-C3) and *Defb6* (ACDBio, reference: 430141-C2) mRNA molecules were used. Opal dyes (Akoya Biosciences) were used at a dilution of 1:1,000 for the fluorophore step to develop each channel: Opal 520 Reagent Pack (FP1487001KT), Opal 570 Reagent Pack (FP1488001KT) and Opal 650

Reagent Pack (FP1496001KT). Nuclei were counterstained with DAPI and mounted using ProLong Gold Antifade Mountant. Images were obtained using a Nikon A1R confocal microscope (Nikon) with a 20× objective and were processed using ImageJ.

### F-actin quantification

For F-actin quantification the FIJI macro TWOMBLI[38] was used. The pipeline was developed for quantifying patterns in extracellular matrix, but it can also be applied to analyze the F-actin cytoskeleton. The macro metrics total length and lacunarity were used to analyze the F-actin network.

### Reporting summary

Further information on research design is available in the Nature Portfolio Reporting Summary linked to this article.

## Data availability

Sequencing data that support the findings of this study have been deposited in the Gene Expression Omnibus under the accession code GSE174857. The mouse reference genome sequence (GRCm38) was downloaded from Ensembl and used for alignment of the scRNA-seq data. All other data supporting the findings of this study are available from the corresponding author on reasonable request. Source data are provided with this paper.

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

## Acknowledgements

We thank our many CGTRM colleagues for advice and helpful discussions. We thank the Oxford Genomics Centre at the Wellcome Centre for Human Genetics (funded by Wellcome Trust grant reference 203141/Z/16/Z) for the generation and initial processing of the sequencing data. F.M.W. gratefully acknowledges financial support from Cancer Research UK (C219/A23522), the Medical Research Council (G1100073), the Wellcome Trust (096540/Z/11/Z and 211276/E/18/Z) and the Biotechnology and Biological Sciences Research Council (BB/M007219/1). M.B.-R. is the recipient of an EMBO Postdoctoral Fellowship (ALTF 578-2017). This work was supported in part by the Francis Crick Institute which receives its core funding from Cancer Research UK (CC0102), the UK Medical Research Council (CC0102) and the Wellcome Trust (CC0102). R.T.-R. is the recipient of a King's Prize Fellowship. T.Z. is the recipient of a CSC-King's doctoral studentship and funding through the KCL British Heart Foundation Centre of Research Excellence. This work is supported by the EPSRC Strategic Equipment Grant (EP/M022536/1), the European Commission (Mechanocontrol, grant agreement 731957), BBSRC sLoLa (BB/V003518/1), Leverhulme Trust Research Leadership Award (RL 2016-015), Wellcome Trust Investigator Award (212218/Z/18/Z) and Royal Society Wolfson Fellowship (RSWF/R3/183006) to S.G.-M. We are also grateful to the National Institute for Health Research (NIHR) Biomedical Research Centre based at Guy's and St Thomas' NHS Foundation Trust and King's College London. The views expressed are those of the author(s) and not necessarily those of the NHS, the NIHR or the Department of Health.

## Author contributions

M.B.-R. and F.M.W. designed the study, analyzed the data and wrote the manuscript. M.B.-R. performed most of the experiments and did the quantitative data analysis assisted by P.G.B., G.G., I.M.T., C.G. and M.B. S.A. performed the computational analysis. S.A.M., R.T.-R. and T.Z. conducted and analyzed the AFM experiments, which were supervised by S.G.-M. T.H. performed the live imaging experiments. F.M.W. supervised the study.

## Competing interests

The authors declare no competing interests.

## Additional information

**Extended data** is available for this paper at https://doi.org/10.1038/s41556-023-01234-5.

**Correspondence and requests for materials** should be addressed to Fiona M. Watt.

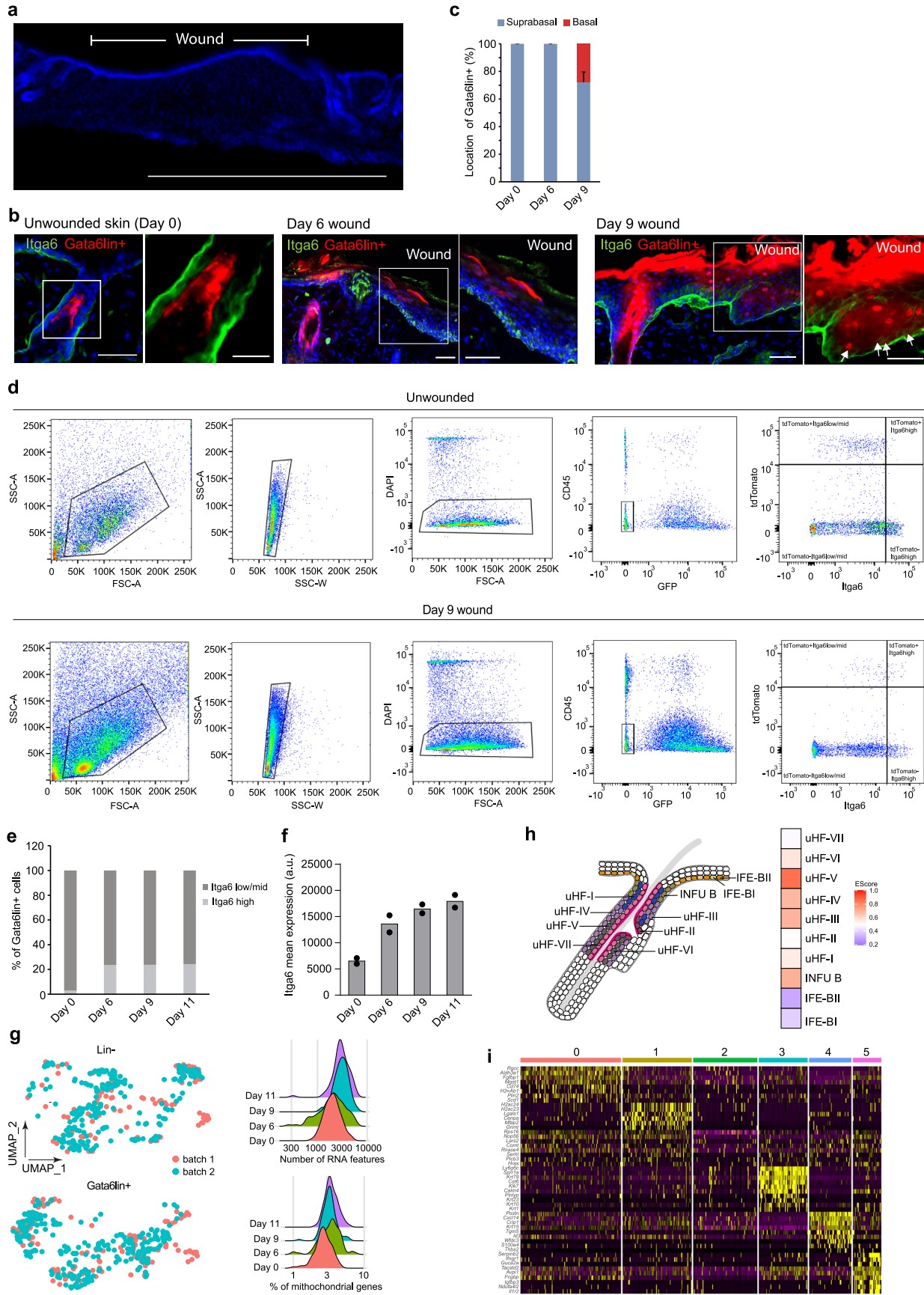

**Extended Data Fig. 1 | See next page for caption.**

**Extended Data Fig. 1 | Flow sorting strategy. (a)** Representative section of wounded skin 9 days after wounding with a 6 mm biopsy punch. DAPI was used to visualize nuclei. Scale bar, 4 mm. n = 4 independent experiments. **(b)** Representative sections of unwounded and wounded skin showing tdTomato Gata6lin+ cells stained for Itga6. Nuclei are visualized with DAPI staining. Boxed regions are shown at higher magnification. White arrows indicate lin+ cells attached to the basement membrane. Scale bars, 40 μm. Representative images from n = 4 independent experiments per timepoint. **(c)** Bar graphs showing the percentage of suprabasal and basal Gata6lin+ cells in unwounded (day 0) and wound cells (day 6 and day 9). Data are means ± s.d. n = 4 mice **(d)** Flow cytometry of epidermal cells from unwounded and wounded skin of Gata6EFGPCreERT2 Rosa26-fl/STOP/fl-tdTomato43 mice buffered with GFP+ cells from CAGGS eGFP mice. GFP⁻CD45⁻, tdTomato⁺Itg6⁻, tdTomato⁺Itg6⁺, tdTomato⁻Itg6⁻, and tdTomato⁻Itg6⁺ cells were analyzed. **(e)** Quantification of low/mid-Itga6 and high-Itga6 Gata6lin+ cells at the indicated timepoints after wounding. n = 2 independent experiments. **(f)** Bar graph showing Itga6 expression (a.u) at different timepoints after wounding. n = 2 independent experiments. **(g)** t-SNE plot showing batch 1 and batch 2 used for single-cell RNA analysis after batch correction (left). Ridgeline plots showing the number of mRNA features and percentage of mitochondrial genes sequenced from every cell of the scRNAseq dataset (right). Data are from two independent biological replicates per timepoint. **(h)** Schematic of location of epidermal populations in undamaged skin based on the second level of clustering introduced in Joost et al., 2016 (left panel). Heatmaps showing the correlation between the lin+ cells and cell compartments defined in the Joost dataset[15] (right panel). **(i)** Heatmap showing the most highly enriched genes in each cluster in Fig. 1d. Clusters were colour-coded along the horizontal axis.

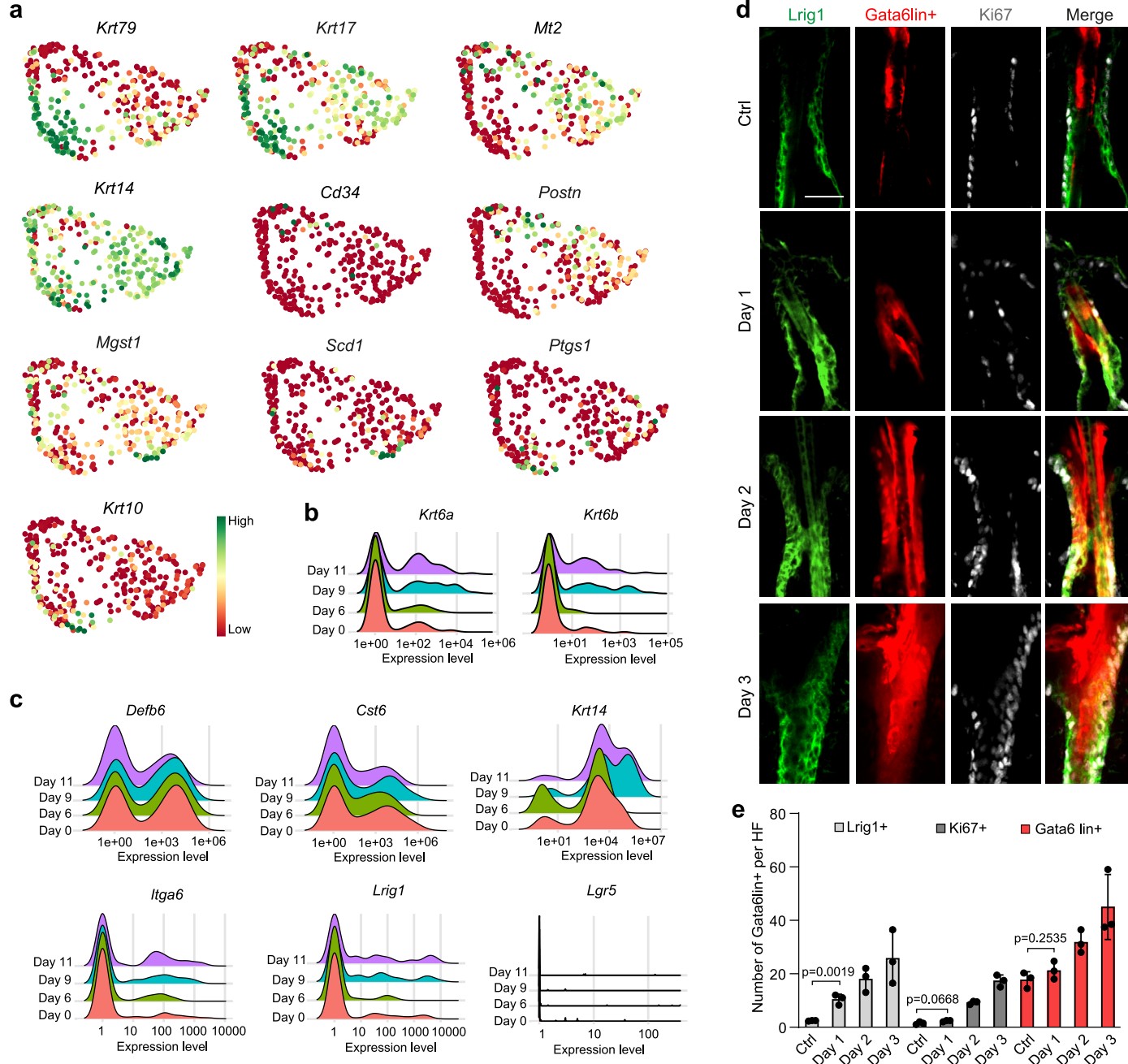

**Extended Data Fig. 2 | Trajectory of Gata6lin+ cells from uHF-like to IFE-like states. (a)** Expression of the upper HF markers *Krt79 and* Krt17, IFE markers *Mt2* and *Krt14*, OB markers *Cd34* and *Postn*, SG markers *Mgst1* and *Scd1*, and IFE-D markers *Ptgs1* and *Krt10* is shown along the pseudotime trajectory. **(b, c)** Ridgeline plots showing the expression of the wound markers *Krt6a* and *Krt6b* (b), the uHF markers *Defb6* and *Cst6* and the basal layer markers *Itga6*, *Krt14*, *Lrig1* and *Lgr5* (c) at days 0, 6, 9, and 11 post-wounding. Data in (a), (b) and (c) are from two independent biological replicates per timepoint. All Gata6lin+

cells from the scRNAseq data were analyzed. **(d)** Sections of HFs proximal to the wound site showing tdTomato Gata6lin+ cells stained for Lrig1 (green) and Ki67 (grey) at days 1, 2, and 3 after wounding. 2 mm wounds were made to assess the early timepoints of wound healing. Scale bar, 40 μm. **(e)** Bar graph shows the number of Gata6lin+ cells expressing Lrig1 (light grey bars), Ki67 (dark grey bars), and the total number of lin+ cells (red bars) in HFs proximal to a wound and ctrl HFs. Data are means ± s.d. n = 3 mice per group. Two-tailed Student's unpaired t-test was used to determine statistical significance.

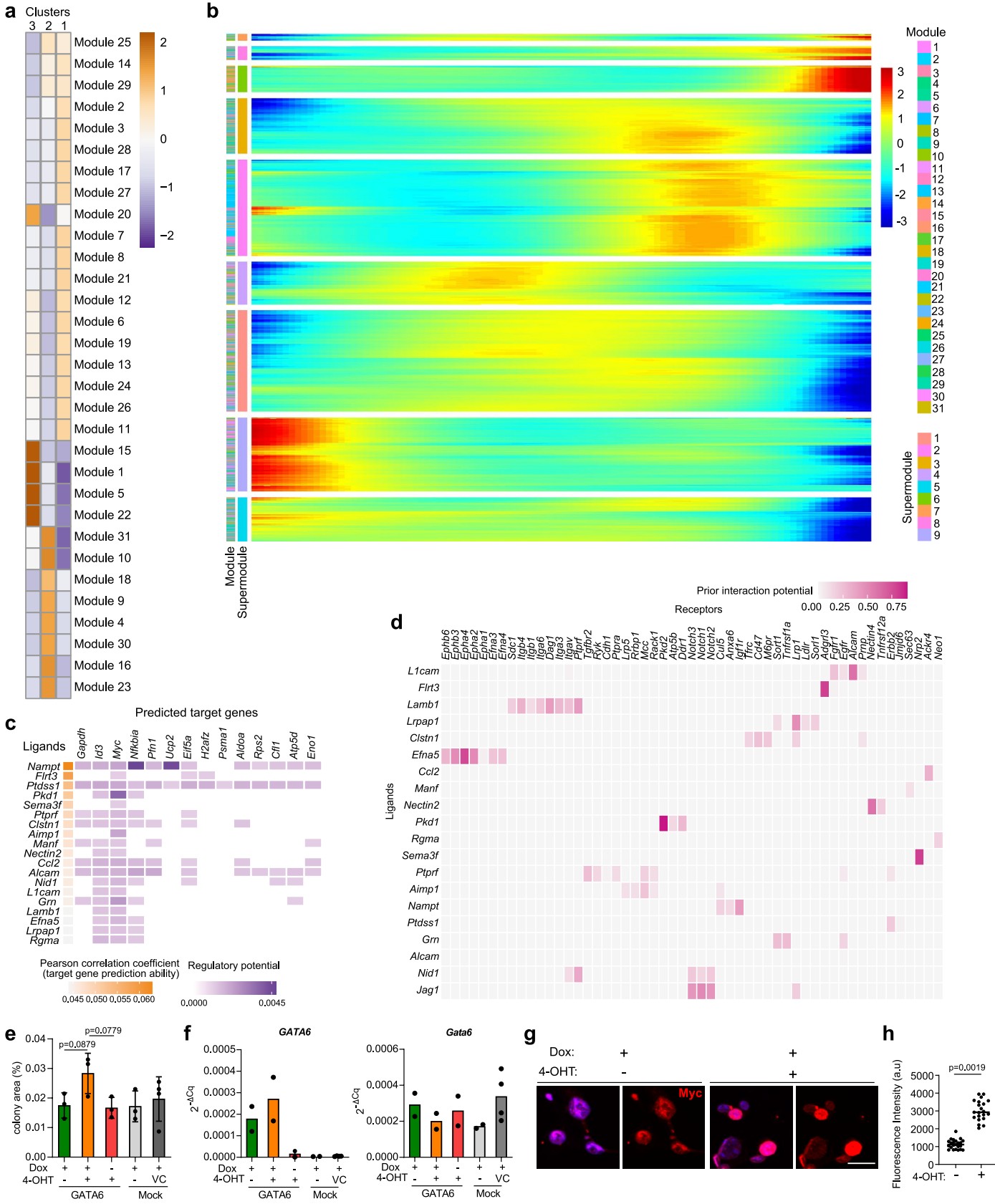

**Extended Data Fig. 3 | See next page for caption.**

**Extended Data Fig. 3 | Pseudotransition-dependent genes along the state trajectory. (a)** Gene modules defined in pseudotime analysis visualized on a heatmap. **(b)** Heatmap showing smoothed expression of pseudotransition-dependent genes ordered by hierarchical clustering and maximum expression. Genes (rows) are ordered by peak expression from cluster 3 to cluster 1. **(c)** Ligand-receptor analysis. In the wound Gata6lin- cells were assumed to be signal sender cells and Gata6lin+ cells receiving cells. Prioritized ligands expressed by sender cells are shown on the vertical axis and predicted target genes on the horizontal axis. **(d)** Ligands expressed by lin- cells are shown on the vertical axis and predicted receptors expressed by Gata6lin+ cells on the horizontal axis. **(e)** Bar graphs showing colony area in each of the indicated conditions. Data are means ± s.d. n = 3 independent experiments **(f)** Bar graphs show gene expression levels of Dox-inducible human *GATA6* (left) and endogenous mouse *Gata6* (right) after 2 μg/ml Dox treatment. n = 2 independent experiments. **(g)** Keratinocytes from K14MycER mice transfected with GATA6 lentivirus were treated with 4-OHT and stained with DAPI and anti-Myc (red). Scale bar, 40 μm **(h)** Bar graph shows fluorescence intensity of nuclear Myc in cells ± 25 nM 4-OHT. n = 23 cells (-) and n = 21 cells (+) from 3 independent experiments. One-way ANOVA with Šidák's multiple comparisons test was used to determine statistical significance in (e). Two-tailed Student's unpaired t-test was used to determine statistical significance in (h).

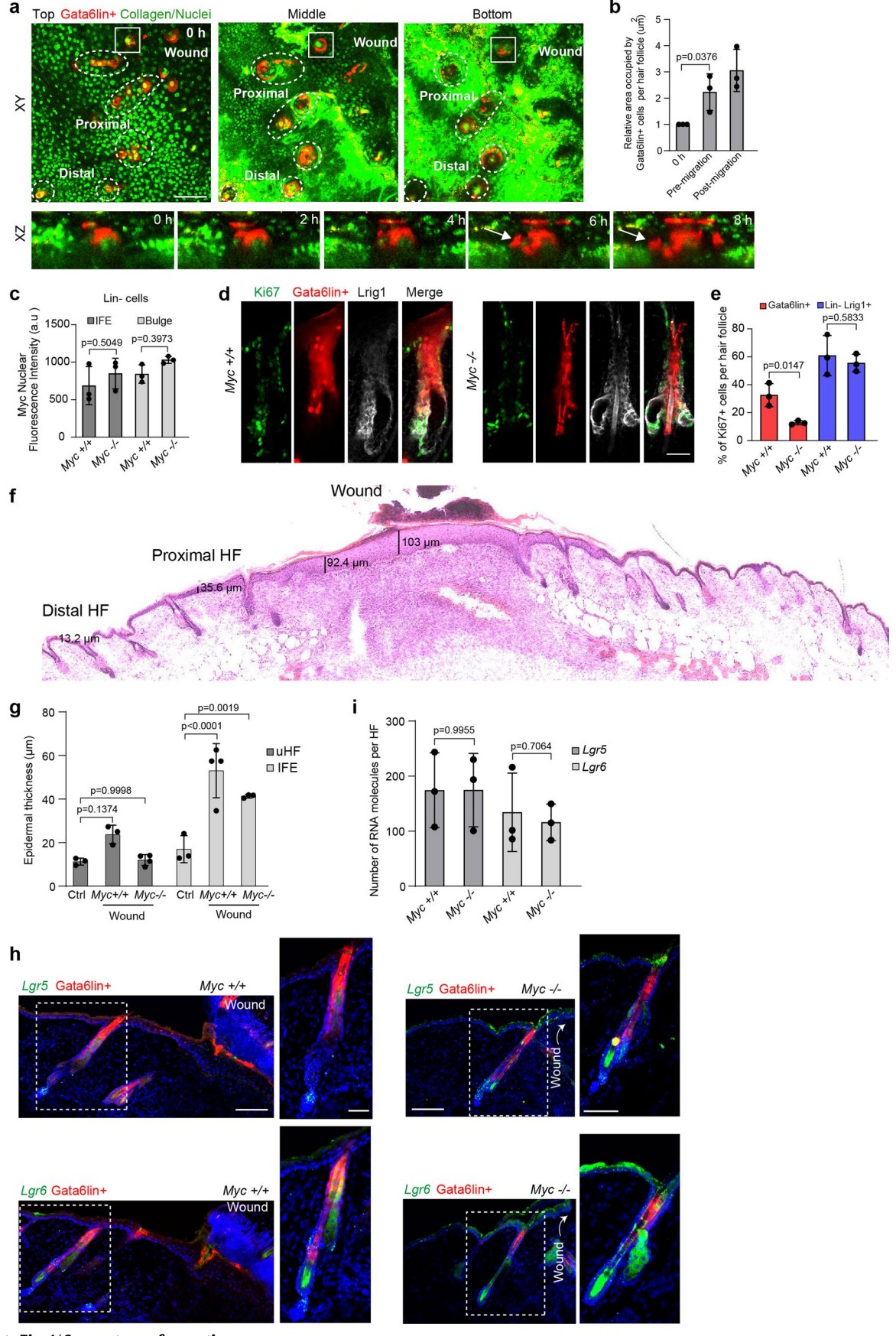

**Extended Data Fig. 4 | See next page for caption.**

**Extended Data Fig. 4 | Effects of Myc depletion in Gata6lin+ cells. (a)** Top view of healing epidermis (top panels). Top, middle and bottom planes are shown. Dashed lines mark proximal and distal HFs to the wound. Boxed region indicates the HF shown in the stills (bottom panels). Stills from time-lapse *in vivo* recording showing expansion of tdTomato Gata6lin+ cells (red) in the HF and subsequent migration (arrows) into the IFE of a healing wildtype wound (to left of HF). Nuclei (Hoechst) and collagen are visualized in green. Scale bar, 100 μm. **(b)** Bar graph shows the relative area occupied by tdTomato Gata6lin+ cells in the uHF at pre-migration and post-migration timepoints. Data are means ± s.d. n = 3 mice. **(c)** Bar graphs showing fluorescence intensity of nuclear Myc in lin- cells present in IFE and bulge. Data are means ± s.d. n = 3 mice. **(d)** Skin of *Myc* +/+ and *Myc* -/- mice showing tdTomato Gata6lin+ cells stained for Ki67 and Lrig1. Scale bar, 40 μm. **(e)** Bar graph shows the percentage of tdTomato Gata6lin+ and lin- Lrig1+ cells expressing Ki67 per HF adjacent to a wound. Data are means ± s.d. n = 3 mice **(f)** Representative H&E staining of day 9 wounded epidermis. n = 4 independent experiments. **(g)** Bar graph showing epidermal thickness in uHFs and IFE after wounding. Data are means ± s.d. n = 3 mice (ctrl uHF, *Myc* +/+ wound uHF, ctrl IFE, *Myc* -/- wound IFE); n = 4 mice (*Myc*-/- wound uHF, *Myc* +/+ wound IFE). **(h)** Detection of *Lgr5* and *Lgr6* by mRNA in situ hybridisation in skin sections of *Myc* +/+ and *Myc* -/- mice showing tdTomato Gata6lin+ cells. Scale bars, 200 μm for the overview and 40 μm for the magnification. **(i)** Quantification of the number of *Lgr5* and *Lgr6* RNA molecules per HF. Data are means ± s.d. n = 3 mice. Two-tailed Student's unpaired t-test was used to determine statistical significance in (b), (c), (e), and (i). One-way ANOVA with Šidák's multiple comparisons test was used to determine statistical significance in (g).

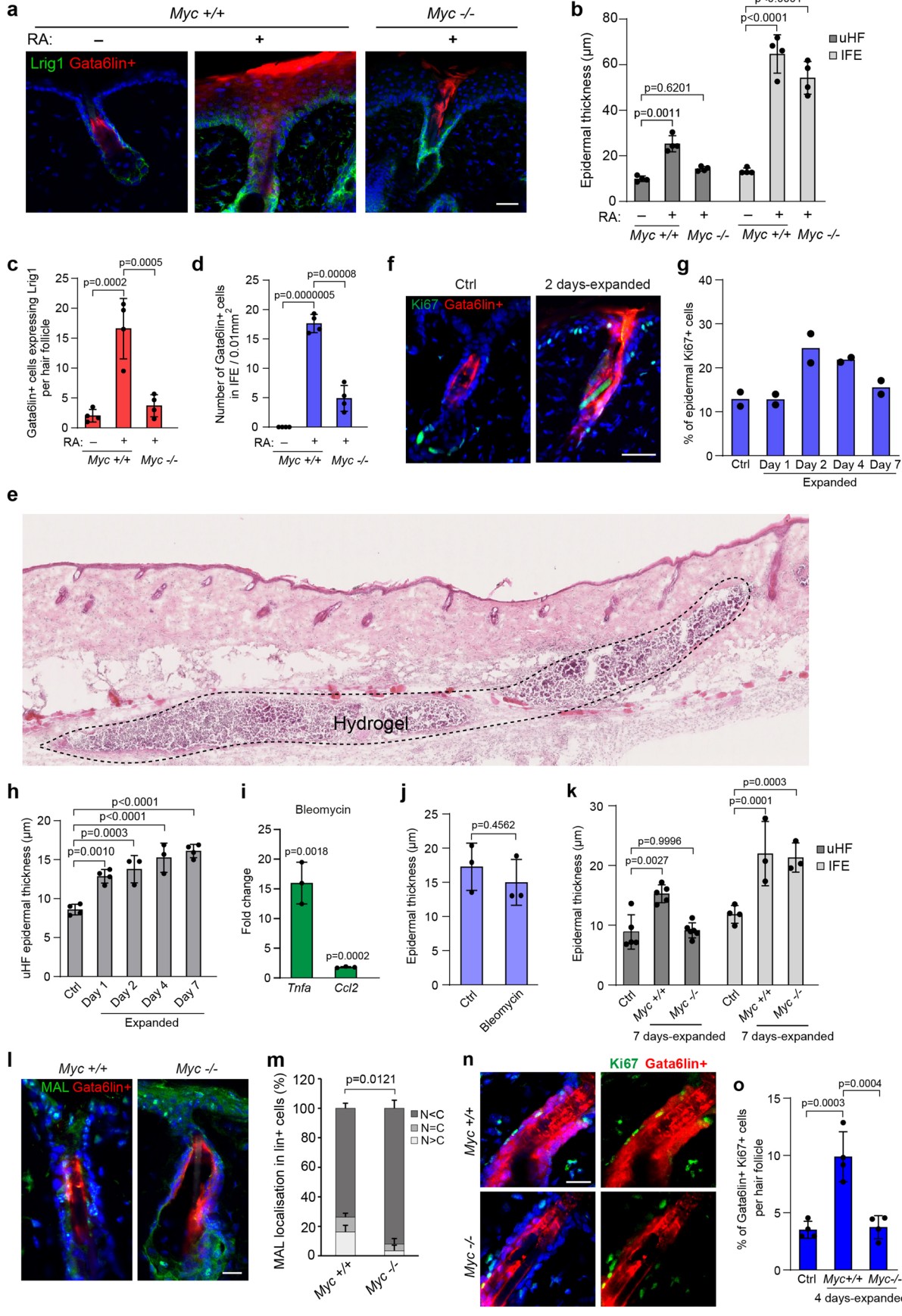

**Extended Data Fig. 5 | See next page for caption.**

**Extended Data Fig. 5 | RA treatment induces dedifferentiation. (a)** Skin was treated with acetone or RA and skin sections were stained for Lrig1 with DAPI counterstain. Scale bar, 40 µm. **(b)** Bar graph shows epidermal thickness upon RA treatment. Data are the mean ± s.d. n = 4 mice. **(c, d)** Bar graphs showing the number of Gata6lin+ cells expressing Lrig1 per HF (c) and Gata6lin+ cells in the IFE (d). Data are the mean ± s.d. n = 4 mice. **(e)** Representative H&E staining of skin of hydrogel-injected mice. n = 4 independent experiments. **(f)** Skin sections showing tdTomato Gata6lin+ cells stained for Ki67. Representative images from n = 2 independent experiments. **(g)** Bar graph showing the percentage of epidermal Ki67+ cells per HF at the indicated timepoints after hydrogel injection. n = 2 mice. **(h)** Bar graph showing epidermal thickness (µm) in the uHF. Data are the mean ± s.d. n = 4 mice (ctrl, day 1, day 7), n = 3 mice (day 2 and day 4). **(i)** Bar graph showing expression of the inflammation markers *Tnfa* and *Ccl2* measured by RT-qPCR. Data are the mean ± s.d. n = 3 mice. **(j)** Bar graph showing epidermal thickness (µm) in control and bleomycin-treated mice. Data are the mean ± s.d. n = 3 mice. **(k)** Bar graph shows epidermal thickness upon hydrogel injection. Data are the mean ± s.d. n = 5 mice (ctrl uHF, *Myc* +/+ expanded uHF); n = 6 mice (*Myc*-/- expanded uHF); n = 4 mice (ctrl IFE); n = 3 mice (*Myc* +/+ and *Myc* -/- expanded IFE). **(l)** Skin sections of *Myc* +/+ and *Myc* -/- hydrogel-injected mice showing tdTomato Gata6lin+ cells stained for MAL. Scale bar, 20 µm. **(m)** Bar graphs showing the percentage of cells showing nuclear MAL (N > C), even distribution of MAL in nucleus and cytoplasm (N = C), and cytoplasmic MAL (N < C). Data are the mean ± s.d. n = 3 mice. **(n)** Skin of *Myc* +/+ or *Myc* -/- hydrogel-injected mice showing tdTomato Gata6lin+ cells stained for Ki67. Scale bar, 20 µm. **(o)** Bar graph shows the number of tdTomato Gata6lin+ cells expressing Ki67 per HF in the indicated conditions. Data are the mean ± s.d. n = 4 mice. One-way ANOVA with Šidák's multiple comparisons test was used to determine statistical significance in (b), (h), and (k). Two-tailed Student's unpaired t-test was used to determine statistical significance in (c), (d), (i), (j), (m), and (o).

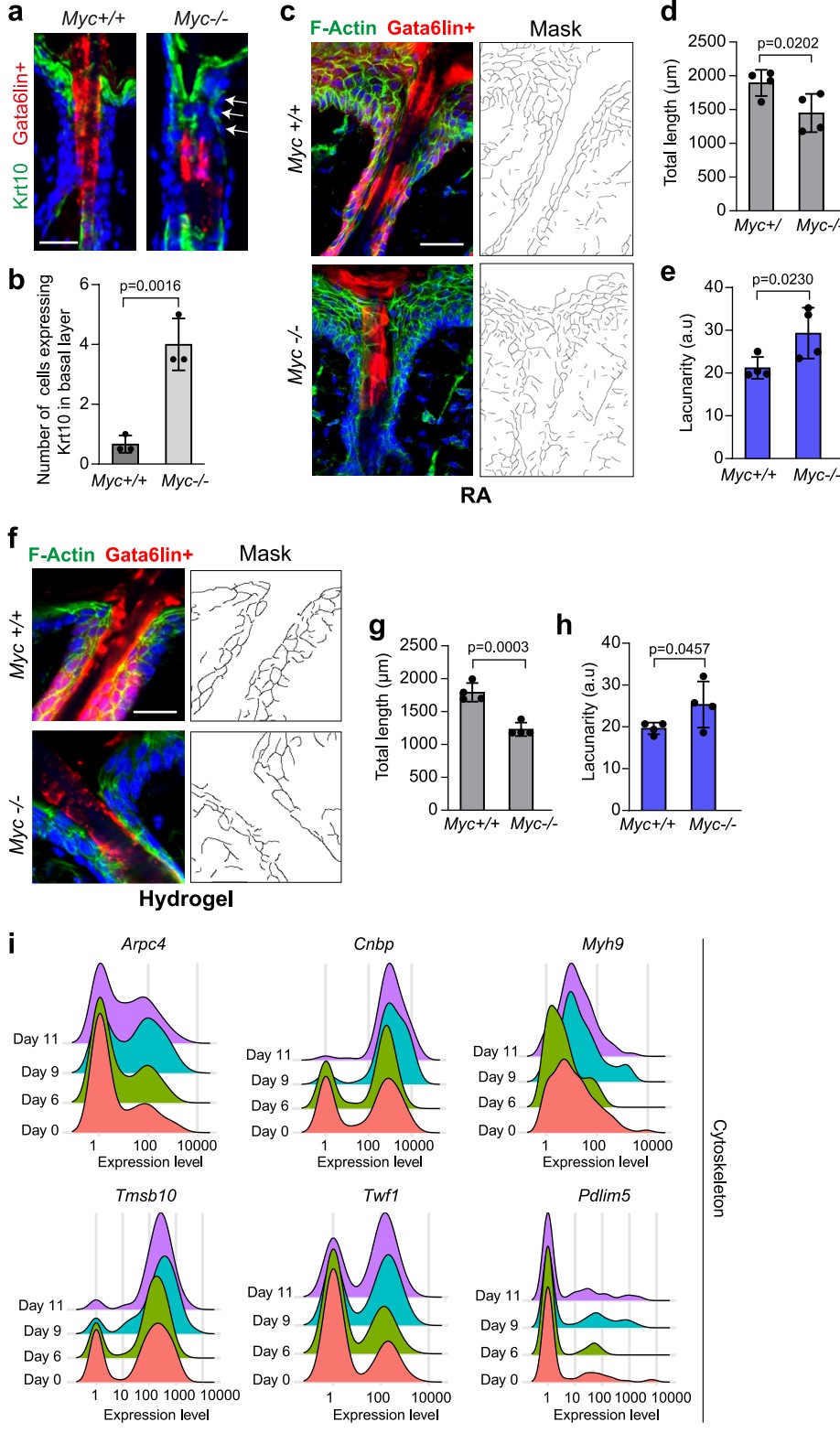

**Extended Data Fig. 6 | See next page for caption.**

**Extended Data Fig. 6 | F-Actin remodelling is linked to dedifferentiation.**
**(a)** Detection of Krt10 in HFs of *Myc + /+* and *Myc-/-* mice showing tdTomato Gata6lin+ cells. Arrows indicate cells in the basal layer expressing Krt10. Scale bar, 20 μm. Representative images from n = 3 independent experiments. **(b)** Quantification of the number of cells per HF expressing Krt10 in the basal layer. Data are the mean ± s.d. n = 3 mice per group. **(c)** Skin sections of RA-treated *Myc + /+* and *Myc -/-* mice showing tdTomato Gata6lin+ cells stained with phalloidin and DAPI (left panels). Masks were obtained using the ImageJ plugin TWOMBLI (right panels). Scale bar, 40 μm. Representative images from n = 4 independent experiments. **(d, e)** Bar graphs show total length (μm) (d) and lacunarity (a.u) (e) in RA-treated epidermis using TWOMBLI. Data are the mean ± s.d. n = 4 mice per group. **(f)** Skin sections of expanded skin of *Myc + /+* and *Myc -/-* mice showing tdTomato Gata6lin+ cells were stained with phalloidin and DAPI (left panels). Masks were obtained using TWOMBLI (right panels). Scale bar, 20 μm. Representative images from n = 4 independent experiments. **(g, h)** Bar graphs show total length (μm) (g) and lacunarity (a.u) (h) in mechanically expanded epidermis using TWOMBLI. Data are the mean ± s.d. n = 4 mice per group. **(i)** Ridgeline plots showing the expression of the cytoskeleton regulators *Arpc4*, *Cnbp*, *Myh9*, *Tmsb10, Twf1*, and *Pdlim5*. Data are from two independent biological replicates per timepoint. All Gata6lin+ cells from the scRNAseq data were analyzed. Two-tailed Student's unpaired t-test was used to determine statistical significance in (b), (d), (e), (g), and (h).

# Reporting Summary

## Statistics

For all statistical analyses, confirm that the following items are present in the figure legend, table legend, main text, or Methods section.

| n/a | Confirmed | |
|---|---|---|
| ☐ | ☒ | The exact sample size (*n*) for each experimental group/condition, given as a discrete number and unit of measurement |
| ☐ | ☒ | A statement on whether measurements were taken from distinct samples or whether the same sample was measured repeatedly |
| ☐ | ☒ | The statistical test(s) used AND whether they are one- or two-sided *Only common tests should be described solely by name; describe more complex techniques in the Methods section.* |
| ☒ | ☐ | A description of all covariates tested |
| ☐ | ☒ | A description of any assumptions or corrections, such as tests of normality and adjustment for multiple comparisons |
| ☐ | ☒ | A full description of the statistical parameters including central tendency (e.g. means) or other basic estimates (e.g. regression coefficient) AND variation (e.g. standard deviation) or associated estimates of uncertainty (e.g. confidence intervals) |
| ☐ | ☒ | For null hypothesis testing, the test statistic (e.g. *F*, *t*, *r*) with confidence intervals, effect sizes, degrees of freedom and *P* value noted *Give P values as exact values whenever suitable.* |
| ☒ | ☐ | For Bayesian analysis, information on the choice of priors and Markov chain Monte Carlo settings |
| ☒ | ☐ | For hierarchical and complex designs, identification of the appropriate level for tests and full reporting of outcomes |
| ☒ | ☐ | Estimates of effect sizes (e.g. Cohen's *d*, Pearson's *r*), indicating how they were calculated |

*Our web collection on statistics for biologists contains articles on many of the points above.*

## Software and code

Policy information about availability of computer code

| | |
|---|---|
| Data collection | Confocal microscopy: Nikon A1 confocal microscope, Nikon A1R confocal, Nikon A1RMP upright microscope. Atomic Force Microscopy: AFM measurements were carried out using a Bioscope atomic force microscope (Bioscope resolveTM BioAFM, Bruker), coupled with an optical microscope (Nikon Eclipse Ti-U) Flow cytometry: BD FACSAria II cell sorter NanoZoomer 2.0RS Digital Slide Scanner (Hamamatsu Photonics K.K) Sequencing: Illumina HiSeq4000 75 PE |
| Data analysis | Data and statistical analysis: Excel (Microsoft) (Version 2305 Build 16.0.16501.20074) Data and statistical analysis: Prism (Graphpad version 9.5.1) Image analysis: Fiji or Image J (NIH) (v1.53) Image analysis: NPD viewer software (Hamamatsu, v2.7.43) RNA-seq: Smart-seq2 sequencing data was aligned with STAR (version 2.2), using the STAR index and aligned to GRCm38 reference genome. Gene-specific counts were calculated using featureCounts function from Rsubread (version3.7) with mm10 RefSeq annotation, and analysed with Seurat version4.1.1. Marker genes of each cell cluster/state were outputted for enrichment analysis using fgsea version1.16.0 package in R. Monocle 2 v2.18.0 and Monocle 3 v0.1.3 (pseudotime/trajectory analysis). CellRank (v1.5.1) was used to asses cell dynamics. For most plots we used ggplot2 (v3.3.6). Flow cytometry analysis: FlowJo software (v10.7.2) |

For manuscripts utilizing custom algorithms or software that are central to the research but not yet described in published literature, software must be made available to editors and reviewers. We strongly encourage code deposition in a community repository (e.g. GitHub). See the Nature Portfolio guidelines for submitting code & software for further information.

## Data

Policy information about availability of data

All manuscripts must include a data availability statement. This statement should provide the following information, where applicable:
- Accession codes, unique identifiers, or web links for publicly available datasets
- A description of any restrictions on data availability
- For clinical datasets or third party data, please ensure that the statement adheres to our policy

The single-cell RNA-sequencing data have been deposited in the Gene Expression Omnibus (GEO) under the accession code GSE174857. The mouse reference genome sequence (GRCm38) was downloaded from Ensembl (http://ensembl.org/Mus_musculus) and used for alignment of the single-cell RNA-seq data.

## Human research participants

Policy information about studies involving human research participants and Sex and Gender in Research.

| | |
|---|---|
| Reporting on sex and gender | N/A |
| Population characteristics | N/A |
| Recruitment | N/A |
| Ethics oversight | N/A |

Note that full information on the approval of the study protocol must also be provided in the manuscript.

# Field-specific reporting

Please select the one below that is the best fit for your research. If you are not sure, read the appropriate sections before making your selection.

☒ Life sciences          ☐ Behavioural & social sciences          ☐ Ecological, evolutionary & environmental sciences

For a reference copy of the document with all sections, see nature.com/documents/nr-reporting-summary-flat.pdf

# Life sciences study design

All studies must disclose on these points even when the disclosure is negative.

| | |
|---|---|
| Sample size | No statistical methods were used to predetermine sample sizes. The sample size was determined based on previous experience with preliminary experiments and previous studies (doi.org/10.1016/j.celrep.2018.09.059; doi.org/10.1038/ncb3532; doi.org/10.1038/s41586-020-2555-7). In accordance with local animal ethics, the experiments were designed to use the smallest number of mice needed to obtain the requested data. All n values are clearly stated in the Figure legends. |
| Data exclusions | No exclusion was applied |
| Replication | All experiments were repeated at least three independent times with independent samples, unless otherwise specified. Precise n values are stated in the Figure legends. All attempts at replication were successful. |
| Randomization | Based on their genotype, the mice were allocated randomly to experimental groups. There was no allocation into experimental groups in other experiments, thus randomization is not relevant beyond mouse experiments in this study. |
| Blinding | The investigators were not blinded to allocation during experiments and outcome assessment |

# Reporting for specific materials, systems and methods

We require information from authors about some types of materials, experimental systems and methods used in many studies. Here, indicate whether each material, system or method listed is relevant to your study. If you are not sure if a list item applies to your research, read the appropriate section before selecting a response.

## Materials & experimental systems

| n/a | Involved in the study |
|---|---|
| ☐ | ☒ Antibodies |
| ☐ | ☒ Eukaryotic cell lines |
| ☒ | ☐ Palaeontology and archaeology |
| ☐ | ☒ Animals and other organisms |
| ☒ | ☐ Clinical data |
| ☒ | ☐ Dual use research of concern |

## Methods

| n/a | Involved in the study |
|---|---|
| ☒ | ☐ ChIP-seq |
| ☐ | ☒ Flow cytometry |
| ☒ | ☐ MRI-based neuroimaging |

## Antibodies

| | |
|---|---|
| Antibodies used | Krt14 (1:1,000, LL002 clone, Abeam ab7800 and 1:1,000, Covance SIG-3476); Itga6 (1:200, GoH3 clone, eBioscience 14-0495-81); Myc (1:100, Abeam ab32072); Ki67 (1:50, Novus BIologicals, NB600-1252); Lrig1 (1:200, R&D Systems AF3688); Involucrin (1:500, ERLI-3 clone, in-house); CD45 (1:200, 30-F11 clone, eBioscience 14-0451-82); pMLC (1:100, Cell Signaling, #3674); YAP (1:200, Cell Signaling, #14074); MAL (1:200, Proteintech, 21166-1-AP); anti-CD45 APC (1:1500, eBioscience, #17-0451-82); anti-CD49f (1:500, Itga6, Biolegend, clone GoH3, #313611). Secondary antibodies conjugated to Alexa Fluor 488 or 647 (1:500) were purchased from Invitrogen: Donkey anti-Rabbit IgG (H+L), Alexa Fluor 647, A32795; Donkey anti-Rat IgG (H+L), Alexa Fluor 647, A48272; Donkey anti-Goat IgG (H+L), Alexa Fluor 488, A48272; Goat anti-Chicken IgG (H+L), Alexa Fluor 488, A21449; Goat anti-Rabbit IgG (H+L), Alexa Fluor 488, A11034 |
| Validation | All antibodies were published and validated in previous studies. Validation statements can be found on the manufacturer's website.<br>Krt14. https://www.abcam.com/cytokeratin-14-antibody-11002-ab7800.html<br>Itga6. https://www.thermofisher.com/antibody/product/CD49f-Integrin-alpha-6-Antibody-done-eBioGoH3-GoH3-Monoclonal/14-0495-82<br>Myc. https://www.abcam.com/c-myc-antibody-y69-ab32072.html<br>Ki67. https://www.novusbio.com/products/ki67-mki67-antibody-sp6_nb600-1252<br>Lrig1. https://www.rndsystems.com/products/mouse-Irig1-anti body_af3688<br>Involucrin. doi: 10.1083/jcb.200706187<br>CD45. https://www.thermofisher.com/antibody/product/CD45-Antibody-clone-30-F11-Monoclonal/14-0451-82<br>pMLC. https://www.cellsignal.co.uk/products/primary-antibodies/phospho-myosin-light-chain-2-thr18-ser19-antibody/3674<br>YAP. https://www.cellsignal.com/products/primary-antibodies/yap-d8h1x-xp-rabbit-mab/14074<br>MAL. https://www.ptglab.com/products/MKL1-Antibody-21166-1-AP.htm<br>CD45 APC. https://www.thermofisher.com/antibody/product/CD45-Antibody-clone-30-F11-Monoclonal/17-0451-82<br>CD49f. https://www.biolegend.com/en-us/search-results/pe-anti-human-mouse-cd49f-antibody-4108<br>Donkey anti-Rabbit, Alexa Fluor 647. https://www.thermofisher.com/antibody/product/Donkey-anti-Rabbit-IgG-H-L-Highly-Cross-Adsorbed-Secondary-Antibody-Polyclonal/A32795<br>Donkey anti-Rat, Alexa Fluor 647, A48272. https://www.thermofisher.com/antibody/product/Donkey-anti-Rat-IgG-H-L-Highly-Cross-Adsorbed-Secondary-Antibody-Polyclonal/A48272<br>Donkey anti-Goat, Alexa Fluor 488. https://www.thermofisher.com/antibody/product/Donkey-anti-Goat-IgG-H-L-Cross-Adsorbed-Secondary-Antibody-Polyclonal/A-11055<br>Goat anti-Chicken, Alexa Fluor 488. https://www.thermofisher.com/antibody/product/Goat-anti-Chicken-IgY-H-L-Secondary-Antibody-Polyclonal/A-21449<br>https://www.thermofisher.com/antibody/product/Goat-anti-Rabbit-IgG-H-L-Highly-Cross-Adsorbed-Secondary-Antibody-Polyclonal/A-11034 |

## Eukaryotic cell lines

Policy information about cell lines and Sex and Gender in Research

| | |
|---|---|
| Cell line source(s) | - Spontaneously immortalised keratinocytes isolated from K14MycER transgenic mouse founder line 2184C.1 (doi.org/10.1016/S0960-9822(01)00154-3).<br>- 3T3-J2 cells were originally obtained from Dr. James Rheinwald (Department of Dermatology, Harvard Skin Research Centre, USA). |
| Authentication | The identity of the keratinocytes isolated from K14MycER transgenic mice was validated by activation of Myc on Tamoxifen treatment. Both keratinocytes and 3T3-J2 cells grew and showed the expected morphology. No additional specific authentication was performed. |
| Mycoplasma contamination | All cell stocks were routinely tested for mycoplasma contamination and found to be negative. |
| Commonly misidentified lines<br>(See ICLAC register) | No commonly misidentified cell lines were used in the study |

# Animals and other research organisms

Policy information about studies involving animals; ARRIVE guidelines recommended for reporting animal research, and Sex and Gender in Research

| | |
|---|---|
| Laboratory animals | Mus musculus. This study includes adult mice (8-12 weeks). The following strains were used: Rosa26-fl/STOP/fltdTomato43, CAGGS eGFP, Gata6EGFPCreERT2, c-Myc fl/fl mice. |
| Wild animals | No wild animals were used in this study. |
| Reporting on sex | Experiments were carried out with male and female mice. No gender-specific differences were observed. |
| Field-collected samples | No field-collected samples were used in the study. |
| Ethics oversight | All mouse procedures were subjected to local ethical approval at King's College London (UK) and performed under a UK Government Home Office (PP70/8474 or PP0313918). |

Note that full information on the approval of the study protocol must also be provided in the manuscript.

# Flow Cytometry

## Plots

Confirm that:

☒ The axis labels state the marker and fluorochrome used (e.g. CD4-FITC).

☒ The axis scales are clearly visible. Include numbers along axes only for bottom left plot of group (a 'group' is an analysis of identical markers).

☒ All plots are contour plots with outliers or pseudocolor plots.

☒ A numerical value for number of cells or percentage (with statistics) is provided.

## Methodology

| | |
|---|---|
| Sample preparation | Single keratinocytes from wounded and unwounded skin of a Gata6EFGPCreERT2 Rosa26-fl/STOP/fl-tdTomato43 mouse were isolated by flow sorting, and labelled with anti-CD45 and anti-CD49f (ltga6). The samples were buffered with GFP+ epidermal cells from CAGGS eGFP mice, in which GFP is expressed in all cells via the CMV-b-actin promoter. GFP+ epidermal cells were subsequently removed by sorting. |
| Instrument | Cell sorting was performed on the BD FACSAria II cell sorter. |
| Software | Data were analysed using FlowJo software. |
| Cell population abundance | Sorted samples were >95% pure. A small fraction of fibroblasts was discarded from the analysis. |
| Gating strategy | Epidermal cells were gated on FSC/SSC- area and width, live (DAPI), GFP- CD45-, tdTomato+ltg6-, tdTomato+ltg6+, tdTomato-ltg6-, and tdTomato-ltg6+. |

☒ Tick this box to confirm that a figure exemplifying the gating strategy is provided in the Supplementary Information.

