## [Peer Review File · Nature Cell Biology]

Peer Review Information

Journal: Nature Cell Biology

Manuscript Title: Mechanical factors induce Myc-dependent dedifferentiation of adult epidermal cells via direct reversal of the terminal differentiation trajectory

Corresponding author name(s): Professor Fiona Watt

Editorial Notes:

Reviewer Comments & Decisions:

Decision Letter, initial version:
--

Dear Professor Watt,

Your manuscript "Mechanical factors induce Myc-dependent dedifferentiation of adult epidermal cells via direct reversal of the terminal differentiation trajectory", has now been seen by 3 referees, who are experts in mechanobiology, epidermis, stiffening (referee 1); single cell transcriptomics of epidermis, wound healing (referee 2); and Myc (referee 3), and whose comments are pasted below. In light of their advice, we regret that we cannot offer to publish the study in Nature Cell Biology.

As you will see, although the reviewers find this work interesting, they raise a number of concerns that question the strength of the data and of the conclusions that can be drawn, and in light of the points they raise, we find the present data-set too preliminary to pursue at this stage.

We are very sorry that we could not be more positive on this occasion, but we thank you for the opportunity to consider this work.

With kind regards,
Stelios

Stylianos Lefkopoulos, PhD
He/him/his
Associate Editor
Nature Cell Biology
Springer Nature
Heidelberger Platz 3, 14197 Berlin, Germany

E-mail: stylianos.lefkopoulos@springernature.com
Twitter: @s_lefkopoulos

Reviewers' comments:

Reviewer #1 (Remarks to the Author):

In this manuscript Bernabé-Rubio and coworkers address the mechanisms by which Gata6+ upper hair follicle cells alter their fate to become cells of the interfollicular epidermis upon wounding or upon other stimuli that trigger epidermal hyperthickening. Using single cell sequencing and associated analyses they show that this fate conversion follows a similar trajectory than the steady state differentiation of Lrig+ upper hair follicle cells into Gata6+ and further demonstrate a critical role for Myc in this process using genetic ablation. Finally they link the process in the mechanosignaling by analyzing expression of actin regulator genes, cell shape and by measuring tissue stiffness.

Overall the manuscript contains a number of interesting observations, and although it seems that the

conversion of Gata6+ cells into epidermis is not essential for tissue repair, this study provides interesting new knowledge into how these types of transitions occur on the transcriptional level. The role of Myc in this process is convincingly demonstrated. On the other hand, the mechanosignaling aspect and particularly its link to Myc appears relatively superficial and underdeveloped and compelling evidence to support the conclusions and statement in the title of mechanical factors inducing Myc-dependent dedifferentiation of adult epidermal cells is essentially lacking.

Specifically:

1. The hydrogel expansion experiment appears superficial, with only quite late time point (d7) presented, and where a lot of secondary effects to produce the hyperproliferation already are involved (most prominently inflammation). The expansion should be carefully described and properly quantified. Also it is not clear where from the hydrogel-injected areas the samples are analysed and images presented are, as forces will be very different at various sites. It would be helpful to show low-magnification images of whole sections for an overview (similar to Extended Fig 5d) to demonstrate the claimed mechanical expansion directly after injection. Also potential secondary effects for example through local inflammation caused by the hydrogel injection directly, or by tissue injury caused by the potential tissue expansion should be very carefully excluded in this type of experiment. As it stands, this experiment does not very directly address mechanical force, which is a problem given the central role of mechanics in the authors claims. Further, panel 4m do not appear compelling as the epidermis does not appear thickened but the hair follicles themselves appear highly abnormal (it is not clear where the hydrogel is positioned here) and consistently Gata6lin+ are quantified from the hair follicles but not from the epidermis. Epidermal Gata6lin+ should be shown and quantified.

2. For the AFM measurements it is not clear what structure is measured: the hair shaft, the dead cells of the section or the ECM? It would be important to clarify.

3. It is perhaps not very surprising that wounding leads to upregulation of cytoskeletal regulators as there is a large increase in migration. The authors do not present any evidence that matrix rigidity is actively influencing this, for example by independently perturbing matrix stiffness. Again, given that this is one of the main conclusions of the manuscript, it would be critical to demonstrate that this local stiffening that the authors measure by AFM is in fact, in a Myc-dependent manner causing the dedifferentiation. Alternatively, the authors should substantially tone down the conclusions) and at minimum, show that 1) upregulation of these actin-regulatory genes occurs Gata6lin+ population 2) coincides spatially with the locally increased stiffness and that 3) deletion of Myc affects the actin cytoskeleton, although given the pleiotropic effects of Myc, even these experiments will not directly demonstrate a role for stiffness. Some in vitro experiments on soft vs stiff hydrogels could be considered

Other points

4. The purpose of the receptor-ligand analyses is not clear as these are not mentioned or discussed, rather the authors seem to use this analysis to further focus on myc, which is a target gene of a very large number of pathways. None of the receptor or ligand candidates are mentioned in the text. The authors should consider clarifying the logic here and elaborating the result in more detail, or alternatively using a more appropriate method such as a transcription factor or gene set enrichment analysis to further provide evidence for the involvement of specific transcription factors

5. The myc staining in Fig 4a does not look compelling as there appears to be a lot of cytoplasmic staining and other potential background signals whereas one would expect mostly nuclear localization for a transcription factor even in the control condition. It also contrasts panel 4d where abundant nuclear myc is seen all along the IFE and multiple hair follicles. Would be helpful to improve the quality of this staining and also show the channel separately without Dapi for the reader to better be able to assess the finding. The quantification would also be more compelling if instead of quantification

of “number of cells with nuclear myc” which is potentially very subjective, the authors would use dapi to generate a nuclear mask and simply quantify nuclear myc intensity in an unbiased manner.

6. “After wounding, the expression of Lrig1 preceded that of the proliferation marker Ki67 in Gata6lin+ cells (Extended Data Fig. 2b, c), indicating that the dedifferentiation process did not require cell proliferation.” This conclusion appears overstated given that it is based on a mRNA levels of a single cell cycle regulator in a very limited number of cells. I would recommend either experimental demonstration of this point or removal of this claim.

7. In figure legends the authors consistently refer to simple bar graphs as “histograms” which is not accurate and should be corrected.

Reviewer #2 (Remarks to the Author):

With this study, Bernabe-Rubio et al set out to investigate how wound healing mediates dedifferentiation of post-mitotic (differentiated) cells to regain stem cell capacity. Previously, the same laboratory has shown that the hair follicle in mouse skin harbors a differentiated (Gata6+) keratinocyte population, which, upon wounding, is recruited towards the interfollicular epidermis (IFE). During this process, some of the traced Gata6 post-mitotic cells changed from the suprabasal to basal layer, which also endowed them with long-term self-renewal in the wound epidermis. This previous work opened up a next most fascinating question: how is this dedifferentiation accomplished on a molecular level. This study builds on the previous finding, here using the same Gata6 mouse model for lineage tracing combined with single-cell RNA-seq (scRNAseq) analysis. Even though this combination seems ideal to answer this question, it is technically very challenging and rigorous controls would be necessary at all levels of experimentation and analysis. Unfortunately, as seen by the comments below, the claims throughout the manuscript are far from being supported by their data, starting from Gata6-lineage tracing, presented tracing patterns and stainings, isolated cell populations for scRNAseq analysis, scRNAseq analysis itself as well as its interpretation.

Identifying the mechanism(s) governing dedifferentiation in a tissue context is obviously fascinating, which the authors tackle in mouse skin. Being in great support for finding an answer, the following comments are meant as constructive feedback for the authors to potentially redesign and expand their study in order to convincingly show the molecular basis of dedifferentiation in skin.

Figure 1 & Extended Data 1:

1. The key prerequisite for the entire study is that Gata6 cells in unwounded skin are exclusively suprabasal as the presence of basal cells would jeopardize the entire endeavor to study dedifferentiation in the chosen experimental setup, which unfortunately seems to be the case. In Fig1a/S1b (Gata6lin+ population FACS plots) appear to be only a couple (<5) cells that might be considered tdTomato+Itg6-. From this one would conclude that the (vast majority of) sorted Gata6lin+ cells are basal. Fig1b shows at least 3 cells that are basal in unwounded skin, which in the sequencing data cannot be distinguished from potentially dedifferentiated cells when contributing the wound epidermis (i.e. the traced wound IFE cells could come from originally basal cells and not from dedifferentiates ones). Consistently, Fig 1c shows a number of Gata6lin+ cells that are characterized as IFE-B.

2. The authors sorted for: “tdTomato+Itg6-, tdTomato+Itg6+, tdTomato-Itg6-, and tdTomato-Itg6+ cells (FigS1b).” It is unclear what these 4 populations represent in healthy and wounded samples and

if all of them were used in the analysis (needs to be clarified in main text). Please include exemplar sorting strategy plots for all population, give a more detailed sorting strategy description, and provide feature plots showing the fluorescence intensity for markers used in sorting (ITGA6, tdTomato) on the tSNE plots.

3. How many mice were used for scRNAseq? It seems that the cells come from one (!) post-wounding 9d and one (!) post-wounding 11d mouse, with respective 0-day control samples from both mice taken distal from the wound site, resulting in ~300 cells. First, analyzing such a small dataset from a single biological replicate per time point is not sufficient; the authors should include biological replicates for each time point. Further, could the authors explain why they think that 9 and 11d timepoints are enough to reconstruct the de-differentiation trajectory (lacking transcriptional information of earlier wound healing time points)?

4. How did the authors exclude (experimentally or computationally) that the isolated wound area indeed did not contain surrounding non-wound/HF cells? Even though FigS1a shows a wound with a 4mm scale bar about the size of the re-epithelialized area, experimentally its not easy to collect wound-epithelium only. To test purity, the authors could include feature plots showing Krt6a and Krt6b expression.

5. Fig1c. The authors should also provide feature plots for other well-known markers for different HF cell populations to make sure that all the cell types come from the expected (annotated) cell types (e.g., Cd34, Ly6a, Mgst1, Cidea, Postn, etc). Also, plots showing the expression level of the markers used for cell annotation would help to clarify whether all cells marked as IFE-B (Krt14/Mt2) are indeed IFE-B or parts of them are HF cells (as could be expected as some cells are mapping close to other uHF cells) because both Krt14 and Mt2 are also expressed by HF cells. Overall, a more thorough annotation incl. more genes per population would solve this issue.

6. Fig1d,e. In the analysis of the wound-derived epidermal cell separation into basal and differentiated clusters it seems that numerous cells are still within their original context (i.e., junctional zone), as they seem to have very high expression of markers indicating that identity (Krt79, Cst6, Defb6). The authors need to substantiate their claims with in situ stainings, validating the expression of these typical JZ-genes in Gata6lin+ basal wound epidermis.

7. Minor: please show an overview of Fig1b (right image) to put the placement of the wound-IFE clone in perspective with HFs and the wound edge.

Figure 2 & Extended Data 2:

8. The authors analyze the differentiation trajectory of their sequenced cells, showing downregulation of uHF markers Cst6 and Defb6, as compared to day 0 control, with an upregulation of Itga6 and Krt14. How does this correspond with their shown sorting strategy that almost exclusively included ITGA6+ cells?

9. Additionally, in Fig2b, the authors show each timepoint as a single violinplot, although each timepoint likely includes multiple cell types/states (see e.g. Gonzales et al 2021; Joost et al 2018; Haensel et al 2020). It would be better to already use a state-separation here (as shown in 2d) to make sure that the changes happen within the same cells.

10. Looking at the correlation between dedifferentiation and differentiation (Fig 2c), it's very difficult to understand what was done. Which cells exactly where chosen for the DEG analysis? What exactly is shown on the axes, is it a median expression of the gene in different cell populations? Are the axis

values based on another measure, rather than the actual median expression of the gene? Could the authors also provide a volcano plot showing the identified DEGs? As the panel is right now, it's not possible to understand what the strong anti-correlation actually means.

11. For the pseudotime analysis in Fig2d, can the authors also include the wound time point annotation? Moreover, it's curious how the described pseudotime seems to progress through the OB cell state. Which genes were used for the annotation of cell types/states and how robust is it against potential changes occurring during wounding i.e., higher Krt14 in certain wound-cell states?

12. Please include the annotation of Gata6lin identity on the pseudotime, i.e., which cells are from Gata6lin+ and which from Gata6lin- populations?

13. The authors claim that Lrig1 expression preceded Ki67 in Gata6lin+ cells after wounding (FigS2b,c). From the provided images it is difficult to say where exactly the Lrig1 and Gata6lin+ cells are and how much they overlap. The authors should provide higher quality images and zoom-ins detailing the exact identities of the cells of interest. Minor: in FigS2b wounds, are there indeed all cells Ki67+ or are the images overexposed?

Figure 3 & Extended Data 3:

14. Minor: In FigS3, please improve quality, as it's impossible to read the gene names from the figure.

15. In Fig 3a, it's difficult to understand what the rows are. In figure legend it says that cell states are shown on the vertical axes, but how does it match with the annotation on the horizontal axis and the gene expression in rows?

16. To identify co-expression of Myc and other genes, it would be easier to look at scatterplots of respective gene expressions. Especially since Myc expression can be found in many cells, it would be interesting to see how its expression level correlates with the other genes. Fig3e is an interesting depiction of the co-expression levels, but it's unclear what was considered a positive Myc expression cutoff? Relating to Fig3d, which is the actual endpoint the authors consider for pseudotime in Fig3e? A more detailed description would be important.

Figure 4 & Extended data 5:

17. Please provide overviews of the depicted areas to see the wound site and overall expression patterns together with the detailed zoom-ins. Higher-quality images are desirable.

18. For the intravital imaging shown in FigS5a, please include a larger area to be able to see/identify what is shown and where in the tissue this occurs. It would be best to have this timelapse as a supplementary video, showing the bigger context and changes over time.

19. In quantifications of Myc expression in different populations, could the authors explain better how they decide where the IFE starts and which cells still belong to the uHF? There is an example in Fig4a, but only in the wound proximal HF, so it's difficult to extrapolate what the authors would consider IFE in the distal HF context.

20. In Fig4g, the authors show that Myc+/+ mice have more Gata6lin+ cells in the IFE. Could they also provide an image supporting this finding, since the existing image in 4d shows only a red hair shaft in the IFE area and the rest of the cells seem to be confined to the infundibulum.

21. (!) In Fig4m, the authors say that “this resulted in dedifferentiation of Gata6lin+ cells, as indicated by Lrig1 expression and migration into the IFE”. However, the provided image shows highly likely a touch dome next to a guard hair, not a normal IFE as the authors claim. Please include an example of normal IFE to substantiate the claim. Moreover, since the authors see Gata6lin+ cells in the touch dome and guard hair, could they describe in more detail how the baseline Gata6 and Lrig1 expression levels are in those compartments? Alternatively, if the IFE with Gata6lin+ traced cells becomes morphologically touch-dome like, could the authors provide stainings showing the lack of other touch-dome related cell types (e.g.: Krt8 or Gli1 stainings)?

Figure 5:

22. In Fig5d, the authors show Gata6lin+ cells migrating from the HFs (even from second row HFs) to the wound site, covering the IFE areas. With the presented experimental images, this cartoon is not supported and the most generous interpretation would be that the Gata6lin+ cells stay within the immediate HF opening area. Could the authors provide whole mount (top-down) images of the re-epithelialized wound, showing the existence of Gata6lin+ cells between 1st and 2nd row HFs and in the wound site?

Data deposition:

Data: It would be good to also deposit the fastq files from sequencing experiments, together with the already deposited count matrices and metadata.

Minor comments:

- Barplots and violinplots are labeled in figure legends as histograms, while there’s only a single actual histogram panel in Fig 3e.
- How strict was sorting in making sure that no duplicates and contaminant populations entered the wells? Is there any additional step to computationally remove duplicates?
- As understood, there are control cells coming from 2 different mice (9d and 11d post-wounding). Was there any batch effect with these different controls and how was it identified and corrected for?
- What clustering method was used in the scRNA-seq analysis?

Reviewer #3 (Remarks to the Author):

Bernabe-Rubio et al. investigate the process of dedifferentiation of Gata6+ epidermal cells upon skin wounding. The manuscript is a follow-up study from Donati et al. (2017) that first demonstrated a contribution of Gata6lin+ cells to skin wound repair. By combining lineage tracing and single-cell transcriptomics, the authors first show that the progeny of Gata6+ cells (Gata6lin+ cells) is transcriptionally indistinguishable from non-progeny epidermal cells in the wound site. Single-cell trajectory analysis then suggests that the dedifferentiation of Gata6+ cells occurs via a reversal of normal differentiation from Gata6+ differentiated cells to Lrig1+ stem cells. Pseudotime analysis further revealed varying expression of different transcription factors over the course of the regenerative response. In line with ligand-receptor analysis, the authors identify c-MYC as a potential mediator of the dedifferentiation of Gata6+ cells. Cell culture experiments subsequently demonstrate that overexpression of c-MYC and GATA6 promotes colony formation and reduces terminal differentiation. In contrast, depletion of c-MYC within Gata6+ cells in vivo results in a reduction of the number of Gata6+ cells and Lrig1+Gata6lin+ cells in the wound site. In addition to the injury model, the authors use treatment with retinoic acid (RA) as well as a hydrogel-based mechanical expansion

model to show that MYC is required for dedifferentiation Gata6+ cells. Lastly, the authors assess the influence of mechanical factors in the process. Atomic force microscopy (AFM) analysis showed increased epidermal cell stiffness in hair follicles close to the skin wound in c-MYC proficient skin, while no change in epidermal cell stiffness was observed between hair follicles proximal and distal to the wound in mice with c-MYC depletion targeted to Gata6+ cells. Based on these data, the authors propose that dedifferentiation and epidermal cell stretching have similar transcriptional profiles.

Overall, the manuscript provides some new insights into the molecular process of epidermal cell dedifferentiation. It demonstrates that perturbations of epidermal homeostasis through physical stress (injury, mechanical expansion) or RA treatment induce c-MYC-dependent dedifferentiation of epidermal cells. However, the link between Gata6 and c-MYC has already been made in Donati et al. (2017), raising questions about the novelty of the story. Furthermore, the manuscript requires significant revisions to improve its flow as well as data presentation. In addition, some controls are missing for the c-MYC depletion experiments.

Comments:

1) Statistical power is critical to draw robust conclusions from single-cell RNA-seq data. However, the datasets in the study, in particular, those from the time course experiment (Fig 2), appear to contain only relatively small cell numbers. Did the authors use any analytical test to ascertain statistical power? Further details on the number of mice used for the single-cell RNA-seq analysis are missing. In addition, several statements and presented data require further statistical analyses: e.g., how was defined that the transcriptomes of Gata6lin+ and Gata6lin- cells are indistinguishable? Some violin plots are missing details on the statistical significance of differences described (see Figs 2b, 3b, 5h, 5i).

2) Can the authors provide some information on the quality control for the single-cell dataset (i.e., number of detected genes and transcripts/UMIs, percentage of mitochondrial genes)

3) Figure 1: Two recent studies (Cockburn et al., bioRxiv, 2021; Kretzschmar et al., Stem Cell Reports, 2021) suggest that the population "IFE-DI" with low (dim) expression of Krt10 and Ptgs1 are actually basal keratinocytes. Did the authors make similar findings? Also, can the authors display the transcript counts for Gata6 in a tSNE plot? How exactly did the authors annotate the cell types based on Joost et al. (2016)? Was this done based on marker gene expression or using a specific tool such as Seurat's projection of reference data? Can the authors visualise the enrichment of Gata6lin+ cells per annotated cell type?

4) Figure 2: The trajectory/pseudotime analysis is missing some critical data: Please visualise the collection time points in the trajectory analysis in Fig 2d and in the pseudotime data shown in the extended data figure 3. Also, state 2 is missing in the extended data figure 3. Is it an error? Otherwise, please explain. Did the authors find Gata6lin+ cells in the outer bulge of wounded/unwounded skin? If not, could the authors explain the enrichment for genes associated with the outer bulge in state 2? The manuscript would greatly improve if the authors performed RNA velocity to show the change of direction in differentiation process.

5) Figure 3: For their ligand-receptor analysis, the authors assume that Gata6lin- cells are sender cells and Gata6lin+ cells are receiving cells. Could the authors clarify the rationale for this? As single cell suspensions do not allow to formally assess directly interacting cells, can the authors apply sequencing of physically interacting cells (e.g., Boisset et al., Nature Methods, 2018; Giladi et al., Nature Biotechnology, 2020) to provide evidence for this rationale? For Fig. 3d, can the authors provide correlation data on the expression of Lrig1 and Myc? For instance, using Seurat's biaxial plots? Likewise, the correlation between Lrig1 and Mki67 expression in Myc+ and Myc- cells should be tested

statistically. Panels f and g show increased colony forming efficiency upon overexpression of GATA6 and c-MYC. Do the authors also find evidence for increased colony size? This would fit well with the observed increase in proliferation.

6) Figure 4: How were the quantifications shown here performed? The method part is missing a detailed description on this. In addition, the skin in some of the images provided appears crosscut. This could influence the values obtained for some of the measurements, in particular, the measurement of epidermal thickness. Could the authors please comment? Since the thickness of the interfollicular epidermis is not affected by depletion of c-MYC in Gata6lin+ cells, did the authors observe any delay in wound closure or increased proliferation of Gata6lin- cells (e.g., tdTomato-Lrig1+ stem cells)? Is there some other type of redundancy, for instance, by increased proliferation and migration of hair follicle bulge stem cells and their progeny? Considering that Gata6lin- Lrig1+ stem cells in the upper hair follicle should still be able to proliferate and contribute to skin regeneration, the reduction of epidermal thickness of the upper hair follicle upon depletion of c-MYC in Gata6lin+ cells is quite surprising. Can the authors confirm that Myc expression is fully intact in Gata6lin- Lrig1+ stem cells in the upper hair follicle? Is the number of Lrig1+ stem cells or their proliferative activity changed upon depletion of c-MYC in Gata6lin+ cells? What is the percentage of Lrig1+ stem cells expressing tdTomato with and without c-MYC depletion in the Gata6lin?

7) Figure 5: In line with my comments above, the strong change in epidermal cell stiffness upon depletion c-MYC in Gata6lin+ cells appears somewhat surprising. Could the authors please explain this robust phenotype given the well documented contribution of multiple stem cell compartments to wound repair?

8) The staining showing that MYC is expressed in Gata6lin+ cells (4a and 4d) is questionable. On the one hand, MYC staining is more intense in the cytosol than in the nucleus and the overall intensity is higher in the -/- cells. It is also difficult to interpret because it is not convincing that GATA6-positive cells are also MYC-positive.

9) The last author's laboratory has by now exhaustively documented that deletion or overexpression of MYC has dramatic effects on all aspects of MYC function in skin homeostasis, and that an extensive literature covers the links between MYC and stem cell function: that some level of MYC function is required for dedifferentiation and subsequent expansion of cells after wounding seems to me to be a very, very expected result. In detail, some of the results seem to differ from previous claims: for example, the authors have shown that MYC induces differentiation and suppresses colony formation of keratinocytes, but now show a moderate increase (Figure 3f,g).

**Although we cannot publish your paper, it may be appropriate for another journal in the Nature Portfolio. If you wish to explore the journals and transfer your manuscript please use our manuscript transfer portal. If you transfer to Nature journals or the Communications journals, you will not have to re-supply manuscript metadata and files. This link can only be used once and remains active until used.

All Nature Portfolio journals are editorially independent, and the decision on your manuscript will be taken by their editors. For more information, please see our manuscript transfer FAQ page.

Note that any decision to opt in to In Review at the original journal is not sent to the receiving journal on transfer. You can opt in to In Review at receiving journals that support this service by choosing to modify your manuscript on transfer. In Review is available for primary research manuscript types only.

**For Nature Research general information and news for authors, see <http://npg.nature.com/authors>.

Author Rebuttal to Initial comments

Point-by-point response

Reviewers' comments:

Reviewer #1 (Remarks to the Author):

In this manuscript Bernabé-Rubio and coworkers address the mechanisms by which Gata6+ upper hair follicle cells alter their fate to become cells of the interfollicular epidermis upon wounding or upon other stimuli that trigger epidermal hyperthickening. Using single cell sequencing and associated analyses they show that this fate conversion follows a similar trajectory than the steady state differentiation of Lrig+ upper hair follicle cells into Gata6+ and further demonstrate a critical role for Myc in this process using genetic ablation. Finally they link the process in the mechanosignaling by analyzing expression of actin regulator genes, cell shape and by measuring tissue stiffness.

Overall the manuscript contains a number of interesting observations, and although it seems that the conversion of Gata6+ cells into epidermis is not essential for tissue repair, this study provides interesting new knowledge into how these types of transitions occur on the transcriptional level. The role of Myc in this process is convincingly demonstrated. On the other hand, the mechanosignaling aspect and particularly its link to Myc appears relatively superficial and underdeveloped and compelling evidence to support the conclusions and statement in the title of mechanical factors inducing Myc-dependent dedifferentiation of adult epidermal cells is essentially lacking.

We thank the reviewer for the constructive comments and acknowledge that a more comprehensive analysis on the role of Myc was needed.

Specifically:

1. The hydrogel expansion experiment appears superficial, with only quite late time point (d7) presented, and where a lot of secondary effects to produce the hyperproliferation already are involved (most prominently inflammation). The expansion should be carefully described and properly quantified.

We have now carried out hydrogel injections at earlier time points (day 1, day 2, day 4) and have assessed cell proliferation (Fig. 5a, b), epidermal thickening (Fig. 5d and Extended Data Fig. 5e) and cell contractility in wt and Myc knockout mice (Fig. 7f, g).

Also it is not clear where from the hydrogel-injected areas the samples are analysed and images presented are, as forces will be very different at various sites. It would be helpful to show low-magnification images of whole sections for an overview (similar to Extended Fig 5d) to demonstrate the claimed mechanical expansion directly after injection.

We now provide H&E staining and overview images showing where the hydrogels are positioned within the skin (Fig. 5e, Extended Data Fig. 6f)

Also potential secondary effects for example through local inflammation caused by the hydrogel injection directly, or by tissue injury caused by the potential tissue expansion should be very carefully excluded in this type of experiment. As it stands, this experiment does not very directly address mechanical force, which is a problem given the central role of mechanics in the authors claims.

We have now investigated whether local inflammation influences epidermal thickening. Inflammation peaks 4 days after hydrogel injection. However, Gata6lin+ cells start dedifferentiating one day after hydrogel injection, when there is no inflammation (Fig. 5g-j).

In an additional approach we injected mice with bleomycin to induce inflammation. 4 days after injection inflammation was higher in bleomycin-treated than in hydrogel-treated mice, yet we did not observe dedifferentiation of Gata6lin+ cells (Fig. 5g-j).

As an indirect measure of cell contractility after hydrogel implantation, we have analysed pMLC2 expression in HFs. We show that pMLC2 levels are greater in hydrogel-treated mice than in untreated mice (Fig. 7f, g).

Further, panel 4m do not appear compelling as the epidermis does not appear thickened but the hair follicles themselves appear highly abnormal (it is not clear where the hydrogel is positioned here) and consistently Gata6lin+ are quantified from the hair follicles but not from the epidermis. Epidermal Gata6lin+ should be shown and quantified.

Regarding panel 4m, Gata6lin+ cells were quantified in the HFs and the IFE, as shown in histograms 4n and 4o. We have replaced the micrograph in panel m, and the new micrograph shows morphologically normal hair follicles (new Fig 5k-m).

2. For the AFM measurements it is not clear what structure is measured: the hair shaft, the dead cells of the section or the ECM? It would be important to clarify.

The AFM measurements were made in the HF infundibulum, where Gata6+ cells reside. The hair shaft and the ECM were excluded from the analysis. We have clarified this in the Results and Methods sections and have included new images that are more informative than previously (Fig. 6a).

3. It is perhaps not very surprising that wounding leads to upregulation of cytoskeletal regulators as there is a large increase in migration. The authors do not present any evidence that matrix rigidity is actively influencing this, for example by independently perturbing matrix stiffness. Again, given that this is one of the main conclusions of the manuscript, it would be critical to demonstrate that this local stiffening that the authors measure by AFM is in fact, in a Myc-dependent manner causing the dedifferentiation.

We did not intend to suggest that matrix stiffness is altered, because we have measured the stiffness of the epidermal cells themselves and not the extracellular matrix (see point 2 above). In addition, our AFM measurements taken in the upper HF – although cells undoubtedly migrate from there into the IFE we would contrast this with the mass cell migration involving the leading front of keratinocytes moving into the wound bed. We have now clarified these points in the text.

Alternatively, the authors should substantially tone down the conclusions) and at minimum, show that 1) upregulation of these actin-regulatory genes occurs Gata6lin+ population 2) coincides spatially with the locally increased stiffness and that 3) deletion of Myc affects the actin cytoskeleton, although given the pleiotropic effects of Myc, even these experiments will not directly demonstrate a role for stiffness. Some in vitro experiments on soft vs stiff hydrogels could be considered

We have revised the manuscript thoroughly and believe that the conclusions are now a better match for the data.

Regarding suggestion (1) we have previously identified Gata6 target genes using ChIP-Seq (Donati et al., 2017) and reported that Gata6 regulates genes involved in processes such as "tube development" and "cell motility", consistent with our observation that Gata6 promotes migration of cultured mouse epidermal cells. We have now clarified this in the manuscript.

While our AFM measurements on tissue sections did not have sufficient resolution to determine the stiffness of individual cells, we have been able to analyse the stiffness of cells in the upper HF, which is the location of the Gata6+ cells (suggestion 2). Our revised manuscript compares upper HF stiffness in wt mice and in mice in which Myc has been selectively deleted in the Gata6 lineage (Myc-/-). We show that the stiffness of distal follicles is higher in Myc-/- than wt but the stiffness of proximal follicles is the same in Myc-/- and wt. Thus Myc deletion not only increases uHF stiffness distal to the wound but also results in a loss of stiffness differential between distal and proximal upper HF.

To complement the measurements on tissue sections we have now performed single-cell AFM on primary keratinocytes derived from wt and Myc knockout mice (Fig. 6d-g). Myc-depleted cells were

stiffer and showed a reduction in cortical actin relative to control cells (reviewer's suggestion 3). This is consistent with our new data showing that the upper HFs of mice in which Myc is deleted in Gata6lin+ cells have reduced levels of pMLC2, indicating loss of contractility. Thus, the increase in stiffness observed in Myc-depleted Gata6lin+ cells is independent of the actin network and cell contractility. We believe that the explanation lies with the finding that the spinous and the granular layers of the epidermis are stiffer than the basal layer (Fiore et al., 2020) and epidermal Myc-knockout results in premature differentiation (Zanet et al., 2005). In line with these studies, we found that the basal layer in the upper HF of Myc-depleted mice expressed suprabasal markers (Fig. 6h, I and Extended Data Fig. 6a, b), thereby potentially explaining the greater stiffness of distal HF and the loss of the differential between wt and Myc-/- in proximal HF.

Other points

4. The purpose of the receptor-ligand analyses is not clear as these are not mentioned or discussed, rather the authors seem to use this analysis to further focus on myc, which is a target gene of a very large number of pathways. None of the receptor or ligand candidates are mentioned in the text. The authors should consider clarifying the logic here and elaborating the result in more detail, or alternatively using a more appropriate method such as a transcription factor or gene set enrichment analysis to further provide evidence for the involvement of specific transcription factors

We accept this criticism and in the revised manuscript we have clarified the rationale for including the ligand-receptor analysis. We have also performed a transcription factor enrichment analysis, which further strengthens the case for focusing on Myc (Fig. 3a).

5. The myc staining in Fig 4a does not look compelling as there appears to be a lot of cytoplasmic staining and other potential background signals whereas one would expect mostly nuclear localization for a transcription factor even in the control condition. It also contrasts panel 4d where abundant nuclear myc is seen all along the IFE and multiple hair follicles. Would be helpful to improve the quality of this staining and also show the channel separately without Dapi for the reader to better be able to assess the finding. The quantification would also be more compelling if instead of quantification of "number of cells with nuclear myc" which is potentially very subjective, the authors would use dapi to generate a nuclear mask and simply quantify nuclear myc intensity in an unbiased manner.

We have now provided new images and quantitation to show the specificity of the Myc antibody and used DAPI as a nuclear mask, as suggested by the reviewer (Fig. 4a, e).

6. "After wounding, the expression of Lrig1 preceded that of the proliferation marker Ki67 in Gata6lin+ cells (Extended Data Fig. 2b, c), indicating that the dedifferentiation process did not require cell proliferation." This conclusion appears overstated given that it is based on a mRNA levels of a single cell cycle regulator in a very limited number of cells. I would recommend either experimental demonstration of this point or removal of this claim.

The conclusion is based on quantification of skin sections from 12 mice (3 per group). We did not analyse mRNA levels but stained skin sections of Gata6 lineage traced mice with antibodies to Lrig1 and Ki67 after 1, 2, and 3 days of wounding. Rather than being a cell cycle regulator Ki67 is expressed in all phases of the cell cycle apart from G0, which is why it is widely used a marker of proliferating cells. We have clarified our observations in the text and provided new images and quantification (Extended Data Fig. 2d, e).

7. In figure legends the authors consistently refer to simple bar graphs as "histograms" which is not accurate and should be corrected.

We have corrected this throughout the text.

Reviewer #2 (Remarks to the Author):

With this study, Bernabe-Rubio et al set out to investigate how wound healing mediates dedifferentiation of post-mitotic (differentiated) cells to regain stem cell capacity. Previously, the same laboratory has shown that the hair follicle in mouse skin harbors a differentiated (Gata6+) keratinocyte population, which, upon wounding, is recruited towards the interfollicular epidermis (IFE). During this process, some of the traced Gata6 post-mitotic cells changed from the suprabasal to basal layer, which also endowed them with long-term self-renewal in the wound epidermis. This previous work opened up a next most fascinating question: how is this dedifferentiation accomplished on a molecular level. This study builds on the previous finding, here using the same Gata6 mouse model for lineage tracing combined with single-cell RNA-seq (scRNAseq) analysis. Even though this combination seems ideal to answer this question, it is technically very challenging and rigorous controls would be necessary at all levels of experimentation and analysis. Unfortunately, as seen by the comments below, the claims throughout the manuscript are far from being supported by their data, starting from Gata6-lineage tracing, presented tracing patterns and stainings, isolated cell populations for scRNAseq analysis, scRNAseq analysis itself as well as its interpretation.

Identifying the mechanism(s) governing dedifferentiation in a tissue context is obviously fascinating, which the authors tackle in mouse skin. Being in great support for finding an answer, the following comments are meant as constructive feedback for the authors to potentially redesign and expand their study in order to convincingly show the molecular basis of dedifferentiation in skin.

We thank the reviewer for the constructive feedback and have now addressed the specific points raised.

Figure 1 & Extended Data 1:

1. The key prerequisite for the entire study is that Gata6 cells in unwounded skin are exclusively suprabasal as the presence of basal cells would jeopardize the entire endeavor to study dedifferentiation in the chosen experimental setup, which unfortunately seems to be the case. In Fig1a/S1b (Gata6lin+ population FACS plots) appear to be only a couple (<5) cells that might be considered tdTomato+Itg6-. From this one would conclude that the (vast majority of) sorted Gata6lin+ cells are basal. Fig1b shows at least 3 cells that are basal in unwounded skin, which in the sequencing data cannot be distinguished from potentially dedifferentiated cells when contributing the wound epidermis (i.e. the traced wound IFE cells could come from originally basal cells and not from dedifferentiated ones). Consistently, Fig 1c shows a number of Gata6lin+ cells that are characterized as IFE-B.

We fully agree with the reviewer that it is a prerequisite of the study that Gata6 lineage-traced cells in unwounded skin are predominantly suprabasal. We have previously shown that in unwounded skin a low concentration of tamoxifen exclusively labels suprabasal Gata6+ cells and does not label basal Gata6+ cells (Donati et al., 2017). Consistent with the previous report, Fig. 1b in the revised manuscript and old Fig. 1b (now Extended Data Fig. 1b) show that the vast majority of Gata6+ cells in undamaged epidermis are suprabasal and do not co-express Itga6.

Although the FACS plots in Figs. 1a/S1b (old version) do show some Itga6 expression in Gata6+ cells of unwounded skin, there is no significant representation of these cells in the intensity range 10^4 - 10^5 on the X axis (Itga6high) (Extended Data Fig. 1c, top far right panel), which is where a significant proportion of tdTomato+ Itga6+ cells in wounded skin fall (Extended Fig. 1c, bottom far right panel). The higher Itga6 levels are fully consistent with Gata6lin+ cells undergoing dedifferentiation. We now show FACS plots for all time points (Fig. 1a) and statistics showing the percentage of cells gated for expression of low/mid or high levels of Itga6 and Itga6 (Extended Fig. 1d, e). Our data indicates that around 20% of the cells from wounds exhibited a shift to high levels of Itga6 expression (Extended Fig. 1d, e), which is consistent with cells acquiring basal-like properties.

Moreover, in the unwounded sample there is a high representation of Tomato- Itga6high cells, which contrasts with the lack of Tomato+ Itga6high cells (Extended Fig. 1c, top far right panel).

Our conclusions regarding Itga6 expression are not based solely on antibody labelling. We now visualise Itga6 mRNA expression at the single cell level in Gata6lin+ cells and confirm that there is no expression in cells from control skin whereas there Itga6 upregulation in wound cells (Fig. 2c).

An independent confirmation of the validity of our approach is that in tSNE plots of control skin Gata6lin+ cells and lin- cluster separately whereas in wounds they co-cluster (previous Fig. 2a).

There are indeed Gata6lin+ cells categorised as IFE-B cells in wounds (Fig. 2) whereas in unwounded skin Gata6lin+ cells map primarily to the suprabasal layers of the upper HF (uHF-II, uHF-III) (new Fig. 1c).

2. The authors sorted for: "tdTomato+Itg6-, tdTomato+Itg6+, tdTomato-Itg6-, and tdTomato-Itg6+ cells (FigS1b)." It is unclear what these 4 populations represent in healthy and wounded samples and if all of them were used in the analysis (needs to be clarified in main text). Please include exemplar sorting strategy plots for all population, give a more detailed sorting strategy description, and provide feature plots showing the fluorescence intensity for markers used in sorting (ITGA6, tdTomato) on the tSNE plots.

As described above all four populations are present in healthy and wounded skin but their relative abundance differs. All four populations were used in our single-cell analysis, a point we have clarified in the text. We have now included exemplar sorting strategy plots and a detailed description of the sorting strategy. We also provide feature plots showing Itga6 and Gata6 mRNA levels in tSNE plots. We have not included tdTomato RNA expression. Tamoxifen treatment induces Tomato translation by Cre recombinase-mediated cleavage of a stop codon in the Tomato gene sequence (Donati et al., 2017); thus there is no difference in tdTomato RNA expression in lin- cells and lin+ cells.

3. How many mice were used for scRNAseq? It seems that the cells come from one (!) post-wounding 9d and one (!) post-wounding 11d mouse, with respective 0-day control samples from both mice taken distal from the wound site, resulting in ~300 cells. First, analyzing such a small dataset from a single biological replicate per time point is not sufficient; the authors should include biological replicates for each time point. Further, could the authors explain why they think that 9 and 11d timepoints are enough to reconstruct the de-differentiation trajectory (lacking transcriptional information of earlier wound healing time points)?

We agree that more cells and an additional time point were required to strengthen the conclusions of the manuscript. In the previous version, 2 mice were used, one for day 9 and one for day 11; the control samples came from unwounded skin of the same mice. We chose day 9 since this is the time point when we start observing Gata6lin+ cells on the basement membrane (Extended Data Fig. 1b).

To address the reviewer's concern we have now obtained additional biological replicates for the day 9 and 11 time points, together with unwounded skin from the same mice. We have also included an earlier time point (day 6). We only begin to observe wound closure at day 6, which is thus the earliest time point at which we could isolate epidermal cells from wounds. The number of cells we could isolate from day 6 wounds was much lower than from the later time points.

In total, scRNAseq has now been performed on 10 mice. This comprises two for day 9, two for day 11 and 6 for day 6. Control samples were taken from unwounded skin of the mice used at days 9 and 11. In total, 684 cells were analysed, compared with 282 in the previous version of the manuscript.

4. How did the authors exclude (experimentally or computationally) that the isolated wound area indeed did not contain surrounding non-wound/HF cells? Even though FigS1a shows a wound with a 4mm scale bar about the size of the re-epithelialized area, experimentally its not easy to collect wound-epithelium only. To test purity, the authors could include feature plots showing Krt6a and Krt6b expression.

We agree that it was technically challenging to isolate cells from the wounds; however, by making a 4mm biopsy in the centre of a 6mm wound we managed to exclude non-wound cells. In addition,

following the suggestion of the reviewer, we now include feature plots showing *Krt6a* and *Krt6b* expression and confirm that the majority of the cells analysed come from wounds (Fig. 2e, Extended Data Fig. 2b).

5. Fig1c. The authors should also provide feature plots for other well-known markers for different HF cell populations to make sure that all the cell types come from the expected (annotated) cell types (e.g., *Cd34*, *Ly6a*, *Mgst1*, *Cidea*, *Postn*, etc). Also, plots showing the expression level of the markers used for cell annotation would help to clarify whether all cells marked as IFE-B (*Krt14/Mt2*) are indeed IFE-B or parts of them are HF cells (as could be expected as some cells are mapping close to other uHF cells) because both *Krt14* and *Mt2* are also expressed by HF cells. Overall, a more thorough annotation incl. more genes per population would solve this issue.

In current Fig. 1c we now show a projection of control cells from our single-cell dataset on the previously annotated tSNE plot of Joost et al. 2016. Annotations for wound cells are now shown in Extended Data Fig. 2a. We have included feature plots of known markers in our analysis for a more comprehensive annotation. We have used the TransferLabel method from Seurat which uses correlation as the basis for calculating the score for each cell type in the reference dataset with the identified anchors as stated in the standard workflow of Seurat.

6. Fig1d,e. In the analysis of the wound-derived epidermal cell separation into basal and differentiated clusters it seems that numerous cells are still within their original context (i.e., junctional zone), as they seem to have very high expression of markers indicating that identity (*Krt79*, *Cst6*, *Defb6*). The authors need to substantiate their claims with in situ stainings, validating the expression of these typical JZ-genes in *Gata6lin+* basal wound epidermis.

It has been shown that some cells express Krt79 in early wounds (Donati et al., 2017). We now include mRNA in-situ labelling of Cst6 and Defb6 which shows that in addition to the expected uHF location, some Gata6lin+ cells expressing these markers are present in the suprabasal layer of the wounded epidermis (Fig. 1e).

7. Minor: please show an overview of Fig1b (right image) to put the placement of the wound-IFE clone in perspective with HFs and the wound edge.

We have included the requested image (Fig. 1b).

Figure 2 & Extended Data 2:

8. The authors analyze the differentiation trajectory of their sequenced cells, showing downregulation of uHF markers *Cst6* and *Defb6*, as compared to day 0 control, with an upregulation of *Itga6* and *Krt14*. How does this correspond with their shown sorting strategy that almost exclusively included *ITGA6+* cells?

*As discussed above, our FACS analysis shows that Gata6+ cells of unwounded epidermis do not express high levels (10^4 - 10^5) of *Itga6*, whereas a significant fraction of *Gata6+* cells derived from wounded epidermis do. In addition, *Itga6* mRNA and protein levels are significantly higher in wound cells than in control cells (Extended Fig. 1d, e and Fig. 2f). We now show expression of *Itga6* mRNA at the single cell level (Fig. 2f), which confirms that *Gata6lin+* cells derived from wounds express greater levels of *Itga6* than those in control skin.*

9. Additionally, in Fig2b, the authors show each timepoint as a single violinplot, although each timepoint likely includes multiple cell types/states (see e.g. Gonzales et al 2021; Joost et al 2018; Haensel et al 2020). It would be better to already use a state-separation here (as shown in 2d) to make sure that the changes happen within the same cells.

*As suggested by the reviewer we now use state-separation (new Fig. 2e, f and Extended Data Fig. 2a). This helps clarify the concerns about control cells expressing *Itga6*. We have kept the violin*

plots as additional evidence and performed statistical analysis including the new single-cell dataset (Extended Data Fig. 2a, b).

10. Looking at the correlation between dedifferentiation and differentiation (Fig 2c), it's very difficult to understand what was done. Which cells exactly were chosen for the DEG analysis? What exactly is shown on the axes, is it a median expression of the gene in different cell populations? Are the axis values based on another measure, rather than the actual median expression of the gene? Could the authors also provide a volcano plot showing the identified DEGs? As the panel is right now, it's not possible to understand what the strong anti-correlation actually means.

To investigate whether there was a reverse commitment phase that resembled the normal differentiation process we analysed the correlation between differentially expressed genes in differentiating and dedifferentiating cells (545 genes were found). As dedifferentiating cells, we selected wound Gata6lin+ cells that expressed Lrig1, whereas differentiating cells were selected as unwounded lin- cells expressing Lrig1. The axis values are based on aggregated median expression values of the gene across all the cells in each group (differentiating and dedifferentiating). We have clarified this in the text and have provided volcano plots and a table for the identified DEGs (Fig. 2h, Table 1).

11. For the pseudotime analysis in Fig2d, can the authors also include the wound time point annotation? Moreover, it's curious how the described pseudotime seems to progress through the OB cell state. Which genes were used for the annotation of cell types/states and how robust is it against potential changes occurring during wounding i.e., higher Krt14 in certain wound-cell states?

The pseudotime analysis has become much simpler now that we have included more cells. As suggested by the reviewer, we now visualise the time points (Fig. 2b). We had generated heatmaps based on the Joost dataset (old Fig. 2e) to map our cells but in the new revised manuscript we have placed the annotated cell types along the pseudotime trajectory. For cell type prediction/annotation we used the Joost dataset applying the standard Seurat workflow of TransferLabel method. This dataset comprises a large set of genes for each predicted epidermal compartment (Joost et al., 2016), so even though cells can lose their identity during wound healing the results are consistent with our observations in vivo showing a trajectory from the uHF to IFE and bulge. The results showing progress through the OB cell state are consistent with previous findings from the lab showing that Gata6lin+ cells of HF proximal to wounds can migrate into the lower HF (Donati et al., 2017), a point we have now clarified in the text. We have also provided more details of the pseudotime analysis in the Methods section.

12. Please include the annotation of Gata6lin identity on the pseudotime, i.e., which cells are from Gata6lin+ and which from Gata6lin- populations?

Only Gata6lin+ cells were selected for pseudotime analysis (Fig. 2b). We have clarified this in the text.

13. The authors claim that Lrig1 expression preceded Ki67 in Gata6lin+ cells after wounding (FigS2b,c). From the provided images it is difficult to say where exactly the Lrig1 and Gata6lin+ cells are and how much they overlap. The authors should provide higher quality images and zoom-ins detailing the exact identities of the cells of interest. Minor: in FigS2b wounds, are there indeed all cells Ki67+ or are the images overexposed?

We now provide new images and quantification (Extended Data Fig.2d, e)

Figure 3 & Extended Data 3:

14. Minor: In FigS3, please improve quality, as it's impossible to read the gene names from the figure.

We have replaced Fig S3 and have included a table containing all the genes present in each of the gene modules (Table 2).

15. In Fig 3a, it's difficult to understand what the rows are. In figure legend it says that cell states are shown on the vertical axes, but how does it match with the annotation on the horizontal axis and the gene expression in rows?

We show single cells on the vertical axis, and the expression of different transcription factors on the horizontal axis. We have rearranged Fig. 3a and now show expression of the enriched transcription factors in the three clusters found in the pseudotime analysis.

16. To identify co-expression of Myc and other genes, it would be easier to look at scatterplots of respective gene expressions. Especially since Myc expression can be found in many cells, it would be interesting to see how its expression level correlates with the other genes. Fig3e is an interesting depiction of the co-expression levels, but it's unclear what was considered a positive Myc expression cutoff? Relating to Fig3d, which is the actual endpoint the authors consider for pseudotime in Fig3e? A more detailed description would be important.

As suggested by the reviewer, we have provided scatterplots to show how Myc expression correlates with that of other genes (Lrig1, Ki67 and Gata6) (Fig. 3c). The endpoint is state 5 shown in Fig. 2 (old version). We considered a cell to be positive for Myc if it had a value >0 for Myc. Using this simple criterion we labelled cells expressing Myc and then repeated the same step for the other genes. Then we labelled the cells as positive for both (or triple-positive if three genes) genes and present the density plot using ggplot2 with respect to pseudotime.

Figure 4 & Extended data 5:

17. Please provide overviews of the depicted areas to see the wound site and overall expression patterns together with the detailed zoom-ins. Higher-quality images are desirable.

We now provide overviews of the depicted areas and higher magnifications of areas of interest, as well as higher quality images (Fig. 4 and Extended Data Fig. 4).

18. For the intravital imaging shown in FigS5a, please include a larger area to be able to see/identify what is shown and where in the tissue this occurs. It would be best to have this timelapse as a supplementary video, showing the bigger context and changes over time.

We now provide a top view of the in vivo imaging to make it easier for readers to understand the physiological context (Extended Data Fig. 4a).

19. In quantifications of Myc expression in different populations, could the authors explain better how they decide where the IFE starts and which cells still belong to the uHF? There is an example in Fig4a, but only in the wound proximal HF, so it's difficult to extrapolate what the authors would consider IFE in the distal HF context.

As suggested by the reviewer, we have provided new images showing how cells were designated as uHF or IFE cells (Fig. 4a).

20. In Fig4g, the authors show that Myc^{+/+} mice have more Gata6^{lin+} cells in the IFE. Could they also provide an image supporting this finding, since the existing image in 4d shows only a red hair

shaft in the IFE area and the rest of the cells seem to be confined to the infundibulum.

We agree with the reviewer that panel 4d was confusing. We have replaced this image for clarification purposes (Fig. 4a).

21. (!) In Fig4m, the authors say that “this resulted in dedifferentiation of Gata6lin+ cells, as indicated by Lrig1 expression and migration into the IFE”. However, the provided image shows highly likely a touch dome next to a guard hair, not a normal IFE as the authors claim. Please include an example of normal IFE to substantiate the claim. Moreover, since the authors see Gata6lin+ cells in the touch dome and guard hair, could they describe in more detail how the baseline Gata6 and Lrig1 expression levels are in those compartments? Alternatively, if the IFE with Gata6lin+ traced cells becomes morphologically touch-dome like, could the authors provide stainings showing the lack of other touch-dome related cell types (e.g.: Krt8 or Gli1 stainings)?

We now include a more comprehensive analysis of the hydrogel experiments. We agree that the Fig. 4m was confusing and that was not representative of a normal IFE. We do not observe Gata6lin+ cells contributing to touch domes. The revised manuscript provides more exhaustive characterisation of the effect of hydrogels in stimulating Gata6lin+ dedifferentiation (Fig. 5).

Figure 5:

22. In Fig5d, the authors show Gata6lin+ cells migrating from the HFs (even from second row HFs) to the wound site, covering the IFE areas. With the presented experimental images, this cartoon is not supported and the most generous interpretation would be that the Gata6lin+ cells stay within the immediate HF opening area. Could the authors provide whole mount (top-down) images of the re-epithelialized wound, showing the existence of Gata6lin+ cells between 1st and 2nd row HFs and in the wound site?

We have now provided a top view of an in-vivo time-lapse experiment showing a re-epithelialising wound, in which we can observe migration of the Gata6lin+ cells in the first few rows of hair follicles that surround the wound (Extended Data Fig. 4a). We have also revised the cartoon as we agree the old version was potentially misleading (Fig. 6c).

Data deposition:

Data: It would be good to also deposit the fastq files from sequencing experiments, together with the already deposited count matrices and metadata.

As suggested by the reviewer we have already uploaded the data on NCBI-GEO, which can be accessed using the private token provided in the manuscript.

Minor comments:

- Barplots and violinplots are labeled in figure legends as histograms, while there’s only a single actual histogram panel in Fig 3e. *We have changed these.*

- How strict was sorting in making sure that no duplicates and contaminant populations entered the wells? Is there any additional step to computationally remove duplicates? *Even though we used strict sorting criteria to isolate keratinocytes (Extended Data Fig. 1c) we did obtain a small number of contaminating fibroblasts. The fibroblasts identified based on known marker genes and removed from the RNA-seq analysis. We followed the standard procedure of filtering and finding duplicates and did not find any amongst the keratinocytes. In total 706 cells were sent for sequencing. 684 cells were identified as keratinocytes.*

•As understood, there are control cells coming from 2 different mice (9d and 11d post-wounding). Was there any batch effect with these different controls and how was it identified and corrected for? *Control cells were taken from different mice and batch-corrected as shown in Extended Data Fig. 1h. Once we were sure that technical batch effects had been eliminated we used the ComBat method from the sva package with a formula string consisting of both time and run information. We have updated the methods section with the relevant details.*

•What clustering method was used in the scRNA-seq analysis? *We used shared nearest neighbour (SNN) modularity optimisation-based clustering algorithm.*

Reviewer #3 (Remarks to the Author):

Bernabe-Rubio et al. investigate the process of dedifferentiation of Gata6+ epidermal cells upon skin wounding. The manuscript is a follow-up study from Donati et al. (2017) that first demonstrated a contribution of Gata6lin+ cells to skin wound repair. By combining lineage tracing and single-cell transcriptomics, the authors first show that the progeny of Gata6+ cells (Gata6lin+ cells) is transcriptionally indistinguishable from non-progeny epidermal cells in the wound site. Single-cell trajectory analysis then suggests that the dedifferentiation of Gata6+ cells occurs via a reversal of normal differentiation from Gata6+ differentiated cells to Lrig1+ stem cells. Pseudotime analysis further revealed varying expression of different transcription factors over the course of the regenerative response. In line with ligand-receptor analysis, the authors identify c-MYC as a potential mediator of the dedifferentiation of Gata6+ cells. Cell culture experiments subsequently demonstrate that overexpression of c-MYC and GATA6 promotes colony formation and reduces terminal differentiation. In contrast, depletion of c-MYC within Gata6+ cells in vivo results in a reduction of the number of Gata6+ cells and Lrig1+Gata6lin+ cells in the wound site. In addition to the injury model, the authors use treatment with retinoic acid (RA) as well as a hydrogel-based mechanical expansion model to show that MYC is required for dedifferentiation Gata6+ cells. Lastly, the authors assess the influence of mechanical factors in the process. Atomic force microscopy (AFM) analysis showed increased epidermal cell stiffness in hair follicles close to the skin wound in c-MYC proficient skin, while no change in epidermal cell stiffness was observed between hair follicles proximal and distal to the wound in mice with c-MYC depletion targeted to Gata6+ cells. Based on these data, the authors propose that dedifferentiation and epidermal cell stretching have similar transcriptional profiles.

Overall, the manuscript provides some new insights into the molecular process of epidermal cell dedifferentiation. It demonstrates that perturbations of epidermal homeostasis through physical stress (injury, mechanical expansion) or RA treatment induce c-MYC-dependent dedifferentiation of epidermal cells. However, the link between Gata6 and c-MYC has already been made in Donati et al. (2017), raising questions about the novelty of the story. Furthermore, the manuscript requires significant revisions to improve its flow as well as data presentation. In addition, some controls are missing for the c-MYC depletion experiments.

We disagree with the reviewer's conclusion that "the link between Gata6 and c-MYC has already been made in Donati et al. (2017), raising questions about the novelty of the story". We believe that the present manuscript is a major conceptual advance because it shows that the transcriptional trajectory by which a differentiated cell undergoes dedifferentiation is via direct reversal of the normal differentiation trajectory, a finding that is relevant to any experimental model in which dedifferentiation occurs. Nevertheless we accept the criticisms regarding data flow, presentation and controls and have addressed these in the revised manuscript.

Comments:

1) Statistical power is critical to draw robust conclusions from single-cell RNA-seq data. However, the datasets in the study, in particular, those from the time course experiment (Fig 2), appear to contain only relatively small cell numbers. Did the authors use any analytical test to ascertain statistical power? Further details on the number of mice used for the single-cell RNA-seq analysis

are missing. In addition, several statements and presented data require further statistical analyses: e.g., how was defined that the transcriptomes of Gata6lin⁺ and Gata6lin⁻ cells are indistinguishable? Some violin plots are missing details on the statistical significance of differences described (see Figs 2b, 3b, 5h, 5i).

As outlined in response to Reviewer 2 we have increased the number of cells, thereby improving the statistical power of the study. We now provide statistical analyses in the violin plots shown in Extended Data Fig. 2b, c, Fig. 7h, i, and Extended Data Fig. 6i and provide more information about the number of mice used for the scRNAseq experiments. We conclude that wound Gata6lin⁺ and Gata6lin⁻ cells are transcriptionally indistinguishable because they are present in all clusters obtained using unsupervised clustering (Fig. 1d). We have fully revised the manuscript to ensure that statements are supported by appropriate statistical analysis.

2) Can the authors provide some information on the quality control for the single-cell dataset (i.e., number of detected genes and transcripts/UMIs, percentage of mitochondrial genes).

This is now provided in the Methods section. Although it has been suggested in other studies that <5% mitochondrial genes has no major effect on the downstream analysis, we still curated the top 100 mitochondrial genes from Mouse Mitocarta 2.0 database and regressed them out during scaling.

3) Figure 1: Two recent studies (Cockburn et al., bioRxiv, 2021; Kretzschmar et al., Stem Cell Reports, 2021) suggest that the population "IFE-DI" with low (dim) expression of Krt10 and Ptgs1 are actually basal keratinocytes. Did the authors make similar findings? Also, can the authors display the transcript counts for Gata6 in a tSNE plot? How exactly did the authors annotate the cell types based on Joost et al. (2016)? Was this done based on marker gene expression or using a specific tool such as Seurat's projection of reference data? Can the authors visualise the enrichment of Gata6lin⁺ cells per annotated cell type?

We have previously reported the patterned distribution of Krt10⁺ cells in human IFE (Jensen et al., 1999). We did not observe Krt10⁺ cells in the basal layers of undamaged mouse wt HFs but they were present in the HFs of mice in which Myc had been deleted in Gata6⁺ cells (Extended Data Fig. 3e). In our scRNAseq dataset we only have 5 predicted IFE-DI cells, which is too few for a thorough analysis.

We now display transcript counts for Gata6 (Fig. 2e). Yes, we performed Seurat projection to show categories based on the Joost dataset. We have provided more details in the Methods section.

4) Figure 2: The trajectory/pseudotime analysis is missing some critical data: Please visualise the collection time points in the trajectory analysis in Fig 2d and in the pseudotime data shown in the extended data figure 3. Also, state 2 is missing in the extended data figure 3. Is it an error? Otherwise, please explain. Did the authors find Gata6lin⁺ cells in the outer bulge of wounded/unwounded skin? If not, could the authors explain the enrichment for genes associated with the outer bulge in state 2? The manuscript would greatly improve if the authors performed RNA velocity to show the change of direction in differentiation process.

We now visualise collection time points in the trajectory and pseudotime analysis. We have replaced Extended Data Fig. 3, which now includes the additional scRNAseq data. Yes, in HFs close to wounds we find Gata6lin⁺ cells in the outer bulge. This is not the case for unwounded skin. The presence of Gata6lin⁺ cells in lower HFs proximal to wounds was described by Donati et al., 2017, which is consistent with our results. Following the reviewer's suggestion, we have performed RNA velocity and show the trajectory of Gata6lin⁺ cells in Fig. 2c of the revised manuscript.

5) Figure 3: For their ligand-receptor analysis, the authors assume that Gata6lin⁻ cells are sender cells and Gata6lin⁺ cells are receiving cells. Could the authors clarify the rationale for this? As single cell suspensions do not allow to formally assess directly interacting cells, can the authors apply sequencing of physically interacting cells (e.g., Boisset et al., Nature Methods, 2018; Giladi

et al., Nature Biotechnology, 2020) to provide evidence for this rationale? For Fig. 3d, can the authors provide correlation data on the expression of Lrig1 and Myc? For instance, using Seurat's biaxial plots? Likewise, the correlation between Lrig1 and Mki67 expression in Myc+ and Myc- cells should be tested statistically. Panels f and g show increased colony forming efficiency upon overexpression of GATA6 and c-MYC. Do the authors also find evidence for increased colony size? This would fit well with the observed increase in proliferation.

We wanted to understand how the cell microenvironment influenced dedifferentiation of Gata6lin+ cells, and thus we assumed that Gata6lin- cells are sender cells and Gata6lin+ cells are target cells.

We thank the reviewer for suggesting that we apply sequencing of physically interacting cells. However, this is mainly used to characterise interactions between different cell types, rather than subpopulations of one cell type. The technology is new and we feel it falls out of the scope of the present paper.

We have tested the correlation between Lrig1 and Myc and that of Lrig1 and Mki67 statistically (Fig. 3c), as suggested by the reviewer, and have quantified colony size in the colony formation assays (Extended Data Fig. 3e).

6) Figure 4: How were the quantifications shown here performed? The method part is missing a detailed description on this. In addition, the skin in some of the images provided appears crosscut. This could influence the values obtained for some of the measurements, in particular, the measurement of epidermal thickness. Could the authors please comment? Since the thickness of the interfollicular epidermis is not affected by depletion of c-MYC in Gata6lin+ cells, did the authors observe any delay in wound closure or increased proliferation of Gata6lin- cells (e.g., tdTomato- Lrig1+ stem cells)? Is there some other type of redundancy, for instance, by increased proliferation and migration of hair follicle bulge stem cells and their progeny? Considering that Gata6lin- Lrig1+ stem cells in the upper hair follicle should still be able to proliferate and contribute to skin regeneration, the reduction of epidermal thickness of the upper hair follicle upon depletion of c-MYC in Gata6lin+ cells is quite surprising. Can the authors confirm that Myc expression is fully intact in Gata6lin- Lrig1+ stem cells in the upper hair follicle? Is the number of Lrig1+ stem cells or their proliferative activity changed upon depletion of c-MYC in Gata6lin+ cells? What is the percentage of Lrig1+ stem cells expressing tdTomato with and without c-MYC depletion in the Gata6lin?

We have now clarified how quantifications were performed. We have now used DAPI as a mask and quantified the nuclear intensity of Myc in Gata6lin+ cells and lin- cells. Instead of showing the number of Lrig1+ Gata6lin+ cells and the number of Gata6lin+ cells found in IFE, we now show the corresponding percentages. Regarding the calculation of the epidermal thickness, we took measurements in the IFE and in the upper HF using Krt14 and Iv1 as basal and suprabasal layer markers, respectively or ensured that the basement membrane did not vary in thickness in H&E stained sections. We have clarified this in the Methods section.

We did not observe any delay in wound closure in Myc knockout mice. We have analysed proliferation in Gata6lin- Lrig1+ cells of wt and Myc knockout mice and found no difference, based on Ki67 labelling (Extended Data Fig. 4d, e). In addition, we performed RNAscope for Lrig1 (upper HF), Lgr6 (upper HF and IFE), and Lgr5 (bulge). We found that following wounding there is a reduction in Lrig1 in the upper HF (Fig. 4i, j), whereas levels of Lgr5 and Lgr6 remain unchanged (Extended Data Fig. 4h, i). This is consistent with the idea that the Gata6 lineage provides the upper HF with Lrig1+ cells and explains the reduction in the epidermal thickness of the upper HF on Myc deletion. Our results showing a reduction in epidermal thickening are consistent with previously reported data showing that Myc deletion via the Krt5 promoter results in epidermal thinning (Zanet et al., 2005). In Myc KO mice Gata6lin- Lrig1+ cells in the upper HF express Myc and proliferate similarly to the control cells (Fig. 4h, Extended Data Fig. 4e). We now show the percentage of Lrig1+ stem cells expressing tdTomato in wt and Myc knockout mice (Fig. 4e).

7) Figure 5: In line with my comments above, the strong change in epidermal cell stiffness upon depletion c-MYC in Gata6lin+ cells appears somewhat surprising. Could the authors please explain this robust phenotype given the well documented contribution of multiple stem cell compartments

to wound repair?

We have extensively revised the section of the manuscript describing cell stiffness, as described in our response to Reviewer 1.

As pointed out by the reviewer, there is a contribution of multiple stem cell compartments to wound repair. The increase in cell stiffness caused by Myc deletion in HFs distal to wounds was not observed in HFs proximal to wounds (Fig. 6d, proximal HFs), probably due to additional mechanisms involved in wound healing that can be both extrinsic (ECM, inflammation) and intrinsic (different epidermal lineages).

8) The staining showing that MYC is expressed in Gata6lin+ cells (4a and 4d) is questionable. On the one hand, MYC staining is more intense in the cytosol than in the nucleus and the overall intensity is higher in the -/- cells. It is also difficult to interpret because it is not convincing that GATA6-positive cells are also MYC-positive.

We have replaced the images shown in panels 4a and 4d and now show the Myc channel separately to avoid confusion (Fig. 4a, d). We have included new quantifications in which we have analysed the intensity fluorescence of the nuclear Myc instead of the number of cells using DAPI as a mask.

9) The last author's laboratory has by now exhaustively documented that deletion or overexpression of MYC has dramatic effects on all aspects of MYC function in skin homeostasis, and that an extensive literature covers the links between MYC and stem cell function: that some level of MYC function is required for dedifferentiation and subsequent expansion of cells after wounding seems to me to be a very, very expected result. In detail, some of the results seem to differ from previous claims: for example, the authors have shown that MYC induces differentiation and suppresses colony formation of keratinocytes, but now show a moderate increase (Figure 3f,g).

Consistent with previous reports from the lab, we show that overexpression of Myc or Gata6 alone induces differentiation (previous Fig. 3f, g). We only observe an increase in the number of colonies in the condition where Gata6 is induced first and Myc is overexpressed 1 day later. Therefore, Myc prevents Gata6-mediated differentiation, which is consistent with Myc being involved in dedifferentiation. As stated in the discussion, we hypothesise that Myc has a dual role, with low levels of Myc inducing dedifferentiation, and high levels of Myc inducing differentiation. Our lab has previously presented evidence that different levels of Myc elicit different epidermal responses (Berta et al., 2009).

Decision Letter, first revision:

Dear Dr. Watt,

I am writing on behalf of my colleague Dr. Stylianos Lefkopoulos, who is currently out of the office. Please accept our apology for the delay getting back to you with a decision due to difficulties in retrieving reviewers' comments.

Your revised manuscript "Mechanical factors induce Myc-dependent dedifferentiation of adult epidermal cells via direct reversal of the terminal differentiation trajectory", has now been seen by the 3 original referees. In light of their advice, we regret that we cannot offer to publish the study in Nature Cell Biology.

As you will see, although the reviewers find this work interesting, they raise serious and persisting concerns that question the strength of the data and of the novel conclusions that can be drawn at this stage.

We are very sorry that we could not be more positive on this occasion, but we thank you for the opportunity to consider this work.

With kind regards,
Zhe Wang

Zhe Wang, PhD
Senior Editor
Nature Cell Biology

Tel: +44 (0) 207 843 4924
email: zhe.wang@nature.com

Reviewers' comments:

Reviewer #1 (Remarks to the Author):

The revised manuscript has addressed many of the points of this and the other reviewers. In particular, the single cell RNAseq experiments have been improved. However, despite providing additional data on the hydrogel implantation experiments, the evidence that this experiment would exert mechanical forces on the epidermis remains rather weak and overall the conclusion that the effects of MYC on cell fate as implied by the title remain largely unsubstantiated. My recommendation is to substantially tone down the conclusions for this part (also strongly recommend revising the title) as well as addressing the points below.

Specific points:

1. The fact that hydrogel implantation results in mild inflammation which “peaks” a few days after the hyperproliferative response is not strong evidence that damage within the epidermis and the resulting signals coming from the epidermis would not trigger the hyperproliferative response. These types of signals do not necessarily require immune cell infiltration, eg TNF secretion from the keratinocytes themselves could cause this. The authors should specifically demonstrate activation of mechanosignaling in the epidermis: YAP/MRTF activation, pMLC2 activation, remodeling of the actin cytoskeleton, cell shape changes to warrant the conclusion that the hydrogels in fact relay mechanical stress to the epidermis.

2. This reviewer had misunderstood what has been measured by AFM, apologies for that. But the fact that the authors claim to be measuring cell stiffness from cross sections in cryo-embedded tissues is even more confusing. The cytoplasm of epidermal cells is quite small and most of the intracellular space is occupied by the nucleus. By measuring dead cells from a cryomatrix-embedded cross section, it is not clear how cell stiffness can be inferred or what it actually means. Given that the AFM measures a material composite, it is most likely that most of the force curves are derived from a nuclear cross section. How is nuclear stiffness related to the functions of Myc and the “mechanical effects” that are studied here is unclear. I would strongly recommend removing the in vivo AFM measurements as they are inconclusive at best and potentially artefactual.

3. The in vitro experiments using AFM force indentation for stiffness measurements are more logical. Given the indentation depth the authors presumably measure the stiffness of the apical cell cortex. Thus, it is surprising that increased stiffness is accompanied by reduced cortical actin in the Myc-deficient keratinocytes. What is the mechanism of increased stiffness? The authors should analyze the keratin network to reconcile this potential contradiction, for example increased keratin 10 could explain this (see eg work from Thomas Magin laboratory). But again, the altered mechanical properties might very well be a cause, rather than a consequence (as the authors propose) of the observed other changes.

Reviewer #2 (Remarks to the Author):

The authors have made an enormous effort to address the reviewers’ questions by performing most of the suggested experiments and analyses, which certainly is positively noted. Despite this effort, as seen by my comments below, one of the central original concerns remains, namely are the authors indeed looking at dedifferentiation in this manuscript?

Overall, none of the points that the authors bring up prove that the Gata6+Tom+ labelled cells in unwounded skin are exclusively suprabasal (majority of cells being suprabasal is not sufficient). The authors most often refer to their previous study (Donati et al NCB 2017), however in the present study it clearly can be seen that basal cells are labelled even before wounding (clearly seen in Fig S1b; counting at least 3 basal cells, or Fig S5a control; counting several basal cells in the HF including infundibulum and HF opening). In the FACS plots, we do indeed see a “shift” towards higher Itga6 intensities, which is consistent with wound response-inducing expression of basal genes (Joost et al Cell Rep 2018), but again this is not proving dedifferentiation. When comparing the gene expression tSNEs in Fig 2, one can also see several 0d cells with both Gata6 expression and Itga6 expression

within several distinct cell populations.

In sum, how can the authors be sure that what they observe in this study are not basal cells that expand upon injury and persist within the basal layer?

Comparing this study to their previous Donati 2017 study, do Gata6+ cells of dorsal skin (this study) and tail skin (Donati 2017) behave differently? In Donati 2017, the Gata6 GL cells seem to remain preferentially suprabasal at eg. day 3 post wounding (Fig 4a, left in Donati 2017) whereas in dorsal skin Gata6 GL cells directly occupy a large basal area clearly seen at day 3 (Fig S2d in this study). This observation is later in the manuscript highlighted where the authors say (p. 8): "The location of these cells led us to speculate that dedifferentiation of Gata6lin+ cells occurs within the upper HF. Consistent with this, live cell imaging of anaesthetised mice showed that prior to Gata6lin+ cells migrating into the IFE there was an expansion of the Gata6lin+ compartment within the HFs at the wound margin (Extended Data Fig. 4a, b)." Firstly, how did the authors determine that this was indeed a suprabasal cell and not a dividing basal cell? Wound edges are challenging to image and the images only show nuclei and Tomato labelled cells but no basement membrane. Secondly, this in vivo wound healing process was imaged 3 days after wounding, a time point that was also included in Park et al NCB 2017, where the authors used a photo-activatable fluorescent reporter to label suprabasal cells and follow their destiny. Park 2017 conclude that 'Notably, cell displacement between layers was observed to occur exclusively in an upward direction at both time points examined, despite previous models proposing downward cell contribution from differentiated layers to the basal one (Supplementary Fig. 3).' It is noted that Park 2017 imaged ear skin, however, a second study using lineage tracing in tail also argues against dedifferentiation (Aragona et al Nat Commun 2017). It is still debated whether this may be a timing issue (i.e. suprabasal-to-basal cell conversion takes longer time to manifest) or specifically Gata6+ cells are more responsive to conversion (Dekoninck & Blanpain 2018). As the original observation of dedifferentiation is obviously of high interest and often debated, it is especially important to exclude that pre-existing Gata6+ labelled basal cells are the main wound-response basal cell contributors instead of dedifferentiated suprabasal Gata6+ progeny.

Even the inverse correlation of Gata6lin+ cells that dedifferentiated to Lrig1+ stem cells and Lrig1+ lin- cells that differentiated to Gata6+ cells could be explained if Gata6lin+ cells are already basal. It has been shown that epidermal HF and IFE cells in proximity to a wound increase basal signature gene expression (Joost et al Cell Rep 2018; Haensel et al Cell Rep 2020), which could indeed lead to a seeming "dedifferentiation", where the wound Gata6lin+ cells acquire higher expression of basal markers than their unwounded counterparts.

Finally, following up on the Myc co-operation in the potential 'dedifferentiation' process: it is convincing that Myc-/- dramatically reduces the contribution/efflux of cells to the wound IFE, but this again just could reflect the inhibition of proliferation/expansion of pre-existing basal Gata6+ labelled cells as those are also targeted.

All in all, to my very large regret – given the interesting topic and the major effort of the authors – it is not convincing that we are indeed looking at dedifferentiating cells (and not pre-existing expanding basal cells), which is the basis for a manuscript entitled 'Mechanical factors induce Myc-dependent dedifferentiation of adult epidermal cells via direct reversal of the terminal differentiation trajectory'.

Minor comment: The authors emphasize to have performed RNA-velocity analysis, which is not the case. The "Cell fate transition analysis" method section describes their use of CellRank, which is a

collection of methods to estimate cell trajectories based on several inputs (it can include RNA-velocity, but not by default). Here, they used the pseudotime kernel of CellRank with Monocle 3 pseudotime input to calculate cell fate transitions. This has nothing to do with RNA-velocity (i.e., estimating cell fate trajectories by their spliced-unsliced mRNA expression).

Reviewer #3 (Remarks to the Author):

Bernabe-Rubio et al. have comprehensively revised their manuscript and convincingly addressed most of my comments. However, some critical concerns remain, as outlined below:

- 1) For most analysis time points, the number of biological replicates and number of analysed single cells remains low, which weakens the overall robustness of the data presented in the study.
- 2) Comment 2 from the initial revision has not been addressed, as desired. Could the authors please provide data on the number of detected genes and transcripts/UMIs and percentage of mitochondrial genes per replicate (e.g., using box plots)?
- 3) Based on the new data provided in Fig. 2a & b, it seems a significant number of Gata6lin+ cells has the transcriptional profile of IFE basal layer cells on day 0 of wounded skin. In addition, Fig. 2e shows a moderate number of Gata6 read counts in these basal cells. In line with the concerns of reviewer #2, this leaves the possibility that a large fraction of Gata6+ cells may indeed be localised to the IFE basal layer at the beginning of the wounding experiment, which strongly weakens the authors' conclusions on the involvement of predominantly dedifferentiating Gata6lin+ cells in IFE regeneration.
- 4) The authors have added biaxial plots in Fig. 3c, in response to comment 5 from the initial revision. In the accompanying manuscript text, the authors state the plots "show a high correlation between Myc and Lrig1 expression in wound cells". However, an r value of 0.24 is a weak correlation at most (typically, r values below 0.25 indicate no relationship). The authors should therefore tone down their statement. Of note, the label on the x-axis, presumably displaying Myc expression, is missing.
- 5) Fig. 7f shows significant autofluorescence of the hair shaft in some images. Could the authors please clarify whether/how they have considered this in their quantifications shown in Fig. 7g?

Minor comment:

- 6) I would recommend the use of two diverging colours for the colour coding in Fig. 2d to increase readability of the data.

**Although we cannot publish your paper, it may be appropriate for another journal in the Nature Portfolio. If you wish to explore the journals and transfer your manuscript please use our manuscript transfer portal. You will not have to re-supply manuscript metadata and files, but please note that this

link can only be used once and remains active until used. For more information, please see our manuscript transfer FAQ page.

Note that any decision to opt in to In Review at the original journal is not sent to the receiving journal on transfer. You can opt in to In Review at receiving journals that support this service by choosing to modify your manuscript on transfer. In Review is available for primary research manuscript types only.

**For Nature Portfolio general information and news for authors, see <http://npg.nature.com/authors>.

Author Rebuttal, first revision:

Reviewer #1 (Remarks to the Author):

The revised manuscript has addressed many of the points of this and the other reviewers. In particular, the single cell RNAseq experiments have been improved. However, despite providing additional data on the hydrogel implantation experiments, the evidence that this experiment would exert mechanical forces on the epidermis remains rather weak and overall the conclusion that the effects of MYC on cell fate as implied by the title remain largely unsubstantiated. My recommendation is to substantially tone down the conclusions for this part (also strongly recommend revising the title) as well as addressing the points below.

We present evidence that hydrogel implantation causes cell contractility in Gata6lin+ epidermal cells in a Myc-dependent manner, as indicated by p-MLC2 activation (Fig. 7f, g). In addition, we show that Myc modulates actin-cytoskeleton remodelling associated with dedifferentiation in hydrogel-treated skin (Fig. S6f-h). Nevertheless, we are happy to tone down the conclusions from the hydrogel experiments and to change the title of the manuscript.

Specific points:

1. The fact that hydrogel implantation results in mild inflammation which “peaks” a few days after the hyperproliferative response is not strong evidence that damage within the epidermis and the resulting signals coming from the epidermis would not trigger the hyperproliferative response. These types of signals do not necessarily require immune cell infiltration, eg TNF secretion from the keratinocytes themselves could cause this.

We agree with the reviewer that inflammatory signalling does not require immune cell infiltration. In fact, in the revised manuscript we showed that inflammation caused by bleomycin does not induce dedifferentiation (Fig. 5g-j). Upregulation of Tnfa and Ccl2, two markers of inflammation caused by bleomycin treatment, did not have an effect on dedifferentiation (Fig. 5g-j, FigS5g). Our results are consistent with a report showing that after hydrogel implantation epidermal inflammation is not linked to cell proliferation (Aragona et al., Nature 2020). This was discussed in the revised text.

The authors should specifically demonstrate activation of mechanosignaling in the epidermis: YAP/MRTF activation, pMLC2 activation, remodeling of the actin cytoskeleton, cell shape changes to warrant the conclusion that the hydrogels in fact relay mechanical stress to the epidermis.

In the revised version, as a measure of cell contractility, we showed that pMLC2 is upregulated upon hydrogel injection (Fig. 7f, g). Remodelling of the actin cytoskeleton after hydrogel implantation in wt and Myc knockout mice is also shown (Fig. S6f, g). We are happy to examine YAP/MRTF activation, as suggested by the reviewer.

2. This reviewer had misunderstood what has been measured by AFM, apologies for that. But the fact that the authors claim to be measuring cell stiffness from cross sections in cryo-embedded tissues is even more confusing. The cytoplasm of epidermal cells is quite small and most of the intracellular space is occupied by the nucleus. By measuring dead cells from a cryomatrix-embedded cross section, it is not clear how cell stiffness can be inferred or what it actually means. Given that the AFM measures a material composite, it is most likely that most of the force curves are derived from a nuclear cross section. How is nuclear stiffness related to the functions of Myc and the “mechanical effects” that are studied here is unclear. I would strongly recommend

removing the in vivo AFM measurements as they are inconclusive at best and potentially artefactual.

We disagree that the in vivo AFM measurements should be removed. There are several publications that have examined stiffness on histological sections, and we would be happy to cite them as justification of the validity of the approach. While it is probably true that in tissue most of the mechanical response arises from the cell nuclei, it is surprising that we see a clear difference when comparing distal HF's with proximal HF's and when comparing wt with Myc knockout mice (Fig. 6a-c). This indicates that there is a mechanical response to wounding, which is regulated by Myc. The in vivo AFM measurements are in strong agreement with the single cell observations showing that Myc depletion leads to stiffer cells (Fig. 6d). In fact, single cell stiffness measurements were performed on the entire cellular area, and hence, the mechanical response is most likely dominated by the nuclear rigidity. In light of the reviewer's comment, we are happy to discuss in the manuscript that the mechanical response is likely dominated by the stiffness of the nucleus, suggesting the interesting hypothesis that Myc might play a role in the regulation of nuclear mechanics.

3. The in vitro experiments using AFM force indentation for stiffness measurements are more logical. Given the indentation depth the authors presumably measure the stiffness of the apical cell cortex. Thus, it is surprising that increased stiffness is accompanied by reduced cortical actin in the Myc-deficient keratinocytes. What is the mechanism of increased stiffness? The authors should analyze the keratin network to reconcile this potential contradiction, for example increased keratin 10 could explain this (see eg work from Thomas Magin laboratory). But again, the altered mechanical properties might very well be a cause, rather than a consequence (as the authors propose) of the observed other changes.

We have already presented evidence that Myc-depleted cells express higher levels of the differentiation markers Cst6 (Fig 6h, i) and Krt10 (Fig. S6a, b), and discussed how this could account for the effect of Myc depletion on cell stiffness both in the revised text and in our previous point-by-point response (Reviewer 1, point 3). We can, nevertheless, tone down the conclusions, as the reviewer suggests.

In summary, we feel that Reviewer 1 did not assess the revised manuscript thoroughly, because most of the points he/she makes have already been addressed in the revised version.

Reviewer #2 (Remarks to the Author):

The authors have made an enormous effort to address the reviewers' questions by performing most of the suggested experiments and analyses, which certainly is positively noted. Despite this effort, as seen by my comments below, one of the central original concerns remains, namely are the authors indeed looking at dedifferentiation in this manuscript?

We present strong evidence throughout the paper that we are looking at dedifferentiation, such as the transition from suprabasal-like to basal-like signatures (Fig. 2d), the FACS data (Fig. 1a, FigS1c-e), and the location of Gata6lin+ cells exclusively in the suprabasal layers of unwounded/ctrl epidermis (Fig. 1b, Fig. 4a, Fig. 5a, Fig. 5i, Fig. 7a, Fig. S2d). Moreover, the switch from Gata6lin+ to Lrig1+ stem cells is irrefutable evidence of dedifferentiation, as Lrig1+ cells give rise to the Gata6 lineage (Fig. 2f, Fig. S2c, Fig. 4a, b). Additional evidence of dedifferentiation is demonstrated by the initial enrichment of Lrig1+ stem cells, which is due to the switch from Gata6lin+ to Lrig1+ lineage, and not due to proliferation of Lrig1+ cells (Fig. S2d,

e). Finally, *Myc*-knockout in three different contexts (wounding, retinoic acid, and hydrogels) prevents the transition from suprabasal to basal positions in *Gata6lin*⁺ cells as well as the conversion from *Gata6* lineage into *Lrig1* lineage (Fig. 4d-f, Fig. 5k-m, Fig. S5a-d).

Overall, none of the points that the authors bring up prove that the *Gata6*⁺*Tom*⁺ labelled cells in unwounded skin are exclusively suprabasal (majority of cells being suprabasal is not sufficient). The authors most often refer to their previous study (Donati et al NCB 2017), however in the present study it clearly can be seen that basal cells are labelled even before wounding (clearly seen in Fig S1b; counting at least 3 basal cells, or Fig S5a control; counting several basal cells in the HF including infundibulum and HF opening). In the FACS plots, we do indeed see a “shift” towards higher *Itga6* intensities, which is consistent with wound response-inducing expression of basal genes (Joost et al Cell Rep 2018), but again this is not proving dedifferentiation. When comparing the gene expression tSNEs in Fig 2, one can also see several 0d cells with both *Gata6* expression and *Itga6* expression within several distinct cell populations.

Fig. 1b, Fig. 4a, Fig. 5a, Fig. 5i, Fig. 7a, Fig. S2d show representative examples of intact HFs where virtually all cells are suprabasal and there is no overlap between *Gata6*⁺ and *Itga6*⁺ cells (Fig. 1b). Only in wound-proximal HFs and healing IFE, *Gata6lin*⁺ cells express *Itga6* in the basal layer, consistent with the transition from suprabasal to basal positions (Fig. 1b). Please see Fig. 5f for a quantification of the ratio between suprabasal and basal cells. As shown in this figure, 100% of the cells are suprabasal in the control situation. *Lin*⁺ cells are present in the basal layer only after hydrogel injection. Similar observations were made in wounded and retinoic acid-treated skin.

Fig. 2 and Fig. S2 clearly show that there is a downregulation of differentiation markers and upregulation of basal markers. We observe this transition not only at the single cell level but also when we compare gene expression at different time points after wounding (Fig. 2 and Fig. S2). We agree that Figs S1b and S5a may be confusing due to oversaturation and would be happy to replace these images.

Regarding the tSNEs in Fig. 2, we do not observe ‘several 0d cells with *Gata6* and *Itga6* expression’, as the reviewer states. We only observe 3 cells with high *Itga6* expression and low *Gata6* expression at 0d, which can be explained as follows: control cells for single-cell sequencing were isolated from healthy skin adjacent to the wounds (see Methods section). Even though we were very careful in the wound isolation, some signalling related to dedifferentiation may affect these areas. Besides, it is technically difficult to get 100% purity when doing cell sorting (95-99% purity is the normal range). Thus, we cannot rule out the possibility that these 3 cells are contaminants (*lin*⁻ cells) from the single-cell isolation, as they do not express *Gata6*. We would be happy to revisit our single-cell analysis and to discuss this in the text.

In sum, how can the authors be sure that what they observe in this study are not basal cells that expand upon injury and persist within the basal layer?

We are confident that we are observing a transition from suprabasal to basal (Fig. 2d-f and Fig S2a-c), as explained above. Our single-cell data and our lineage tracing strongly indicate downregulation of suprabasal markers and upregulation of basal markers. Please see Fig. 1b for an overview of the entire process. Note that distal HFs show *Gata6lin*⁺ cells only in suprabasal layers whereas proximal HFs show *lin*⁺ cells in suprabasal and basal positions. The presence of

Gata6lin⁺ cells expressing Itga6 in the basal layer indicates that only basal cells that have migrated downwards express Itga6.

Fig. S2d-e shows that dedifferentiation does not require proliferation. 1 day after wounding there is a significant enrichment of Gata6lin⁺ Lrig1⁺ cells that occurs irrespective of proliferation (Fig. S2d, e). Given that 1 day after wounding proliferation remains unchanged in Gata6lin⁺ cells the most plausible explanation is that this enrichment in Lrig1⁺ cells is due to dedifferentiation of suprabasal Gata6lin⁺ cells to Lrig1⁺ stem cells. Hypothetical basal cells would not contribute to this initial enrichment as they already express Lrig1. This is further evidence that we are looking at dedifferentiation.

Moreover, hydrogel injection caused a substantial expansion of the Gata6lin⁺ compartment from the uHF into the IFE 1 day after implantation (Fig. 5g-i). Our data shows that prior to hydrogel implantation 100% of lin⁺ cells are suprabasal (Fig. 5f). Thus, even if there were a few Gata6lin⁺ basal cells in healthy epidermis that we did not detect in our lineage tracing, it is extremely unlikely that they were responsible for such dramatic effect.

Comparing this study to their previous Donati 2017 study, do Gata6⁺ cells of dorsal skin (this study) and tail skin (Donati 2017) behave differently? In Donati 2017, the Gata6 GL cells seem to remain preferentially suprabasal at eg. day 3 post wounding (Fig 4a, left in Donati 2017) whereas in dorsal skin Gata6 GL cells directly occupy a large basal area clearly seen at day 3 (Fig S2d in this study). This observation is later in the manuscript highlighted where the authors say (p. 8): “The location of these cells led us to speculate that dedifferentiation of Gata6lin⁺ cells occurs within the upper HF. Consistent with this, live cell imaging of anaesthetised mice showed that prior to Gata6lin⁺ cells migrating into the IFE there was an expansion of the Gata6lin⁺ compartment within the HFs at the wound margin (Extended Data Fig. 4a, b).” Firstly, how did the authors determine that this was indeed a suprabasal cell and not a dividing basal cell? Wound edges are challenging to image and the images only show nuclei and Tomato labelled cells but no basement membrane. Secondly, this in vivo wound healing process was imaged 3 days after wounding, a time point that was also included in Park et al NCB 2017, where the authors used a photo-activatable fluorescent reporter to label suprabasal cells and follow their destiny. Park 2017 conclude that ‘Notably, cell displacement between layers was observed to occur exclusively in an upward direction at both time points examined, despite previous models proposing downward cell contribution from differentiated layers to the basal one (Supplementary Fig. 3).’ It is noted that Park 2017 imaged ear skin, however, a second study using lineage tracing in tail also argues against dedifferentiation (Aragona et al Nat Commun 2017). It is still debated whether this may be a timing issue (i.e. suprabasal-to-basal cell conversion takes longer time to manifest) or specifically Gata6⁺ cells are more responsive to conversion (Dekoninck & Blanpain 2018). As the original observation of dedifferentiation is obviously of high interest and often debated, it is especially important to exclude that pre-existing Gata6⁺ labelled basal cells are the main wound-response basal cell contributors instead of dedifferentiated suprabasal Gata6⁺ progeny.

Some of the papers that the reviewer cites have been discussed in the revised manuscript. We will discuss these aspects in more detail in the text. In vivo time-lapse experiments shown in the present manuscript were performed in the ear, as in Donati et al., 2017. Collagen and nuclei are shown in green so we can distinguish the basement membrane (see Methods section). In Donati et al., it was already described that there was migration of Gata6⁺ cells from suprabasal to basal compartments, so this is not the focus of the present manuscript. When characterising changes in cell state, we focused on the Gata6 population which exclusively resides in the sebaceous duct, whereas other researchers have focused on IFE cell populations as discussed in our present manuscript and in numerous citations by others of our 2017 publication. While the reviewer may disagree with the Donati et al paper others find it compelling, as judged by over 100 citations.

Even the inverse correlation of Gata6lin⁺ cells that dedifferentiated to Lrig1⁺ stem cells and Lrig1⁺ lin⁻ cells that differentiated to Gata6⁺ cells could be explained if Gata6lin⁺ cells are already basal. It has been shown that epidermal HF and IFE cells in proximity to a wound increase basal signature gene expression (Joost et al Cell Rep 2018; Haensel et al Cell Rep 2020), which could indeed lead to a seeming “dedifferentiation”, where the wound Gata6lin⁺ cells acquire higher expression of basal markers than their unwounded counterparts.

We beg to differ with the reviewer’s conclusion. If the lin⁺ cells were already basal the correlation would be positive, not inverse. Please revisit Fig. 1b and note that Gata6lin⁺ cells in distal/ctrl HFs are exclusively suprabasal whereas in proximal HFs they are in contact with the basement membrane and express Itga6 (Fig. 1b). This indicates that only Gata6lin⁺ cells that have migrated downwards express Itga6, supporting our conclusion that Gata6lin⁺ cells dedifferentiate and acquire stem cell-like properties.

Finally, following up on the Myc co-operation in the potential ‘dedifferentiation’ process: it is convincing that Myc^{-/-} dramatically reduces the contribution/efflux of cells to the wound IFE, but this again just could reflect the inhibition of proliferation/expansion of pre-existing basal Gata6⁺ labelled cells as those are also targeted.

Myc-depletion not only prevents migration from the HF to the IFE but also the conversion of Gata6lin⁺ cells into Lrig1⁺ cells in three different contexts (Fig. 4d-f, Fig. 5k-m, Fig. S5a-d). Given that Gata6⁺ cells arise from Lrig1⁺ cells, this strongly indicates there is a transition into a less differentiated state that is modulated by Myc. Notably, the switch from Gata6lin⁺ cells to Lrig1⁺ stem cells did not require proliferation (Fig. S2d, e), indicating that initiation of dedifferentiation occurs irrespective of cell proliferation. Thus, the increase in Gata6lin⁺ Lrig1⁺ cells is not only due to “proliferation/expansion”, as the reviewer states, but also due to conversion of suprabasal Gata6lin⁺ cells to Lrig1⁺ stem cells.

All in all, to my very large regret – given the interesting topic and the major effort of the authors – it is not convincing that we are indeed looking at dedifferentiating cells (and not pre-existing expanding basal cells), which is the basis for a manuscript entitled ‘Mechanical factors induce Myc-dependent dedifferentiation of adult epidermal cells via direct reversal of the terminal differentiation trajectory’.

In summary, reviewer 2 did not focus on the scope of our study, which is to characterise the molecular mechanisms underlying dedifferentiation, but instead criticised our earlier publication (Donati et al., 2017).

Minor comment: The authors emphasize to have performed RNA-velocity analysis, which is not the case. The “Cell fate transition analysis” method section describes their use of CellRank, which is a collection of methods to estimate cell trajectories based on several inputs (it can include RNA-velocity, but not by default). Here, they used the pseudotime kernel of CellRank with Monocle 3 pseudotime input to calculate cell fate transitions. This has nothing to do with RNA-velocity (i.e., estimating cell fate trajectories by their spliced-unspliced mRNA expression).

We will change this following the reviewer’s suggestion.

Reviewer #3 (Remarks to the Author):

Bernabe-Rubio et al. have comprehensively revised their manuscript and convincingly addressed most of my comments. However, some critical concerns remain, as outlined below:

1) For most analysis time points, the number of biological replicates and number of analysed single cells remains low, which weakens the overall robustness of the data presented in the study.

We disagree. In the revised manuscript, we have increased the number of single cells sequenced three-fold, included a biological replicate and an earlier time point.

2) Comment 2 from the initial revision has not been addressed, as desired. Could the authors please provide data on the number of detected genes and transcripts/UMIs and percentage of mitochondrial genes per replicate (e.g., using box plots)?

We will include this as the reviewer requests.

3) Based on the new data provided in Fig. 2a & b, it seems a significant number of Gata6lin⁺ cells has the transcriptional profile of IFE basal layer cells on day 0 of wounded skin. In addition, Fig. 2e shows a moderate number of Gata6 read counts in these basal cells. In line with the concerns of reviewer #2, this leaves the possibility that a large fraction of Gata6⁺ cells may indeed be localised to the IFE basal layer at the beginning of the wounding experiment, which strongly weakens the authors' conclusions on the involvement of predominantly dedifferentiating Gata6lin⁺ cells in IFE regeneration.

We disagree with the reviewer's conclusion. There is no significant fraction of unwounded Gata6lin⁺ cells in IFE; indeed we did not see any in our lineage tracing. This is supported by our single-cell dataset and by that of Joost et al., 2016. The Joost dataset shows that Gata6 is primarily expressed in the uHF compartment, not in the IFE. Our Fig. 1c shows that the transcriptomes of Gata6lin⁺ cells map primarily to the uHF compartment.

Cells identified as IFE in Fig 2b are from wounded skin, which is consistent with a transition from HF to IFE. Our earlier study (Donati et al., 2017) predicted that HF and IFE Gata6lin⁺ cells would share transcriptional profiles after wounding, based on immunofluorescence labelling for a range of difference markers and so it is not surprising that Gata6lin⁺ cells fall into the IFE category.

Please see heatmaps in Fig. 2e (first version of the manuscript) showing the correlation between cell states and cell compartments. States 1 and 3, which is where control cells fall, show no expression in IFE-B. We are happy to include heatmaps showing the correlation between cell clusters and cell compartments considering the single cell data generated during the revision.

4) The authors have added biaxial plots in Fig. 3c, in response to comment 5 from the initial revision. In the accompanying manuscript text, the authors state the plots “show a high correlation between Myc and Lrig1 expression in wound cells”. However, an r value of 0.24 is a weak correlation at most (typically, r values below 0.25 indicate no relationship). The authors should therefore tone down their statement. Of note, the label on the x-axis, presumably displaying Myc expression, is missing.

We will tone down our conclusion and will correct the labels.

5) Fig. 7f shows significant autofluorescence of the hair shaft in some images. Could the authors please clarify whether/how they have considered this in their quantifications shown in Fig. 7g?

We excluded the hair shaft in our analysis and only measured areas containing Tomato+ cells. We will clarify this in the figure legend.

Minor comment:

6) I would recommend the use of two diverging colours for the colour coding in Fig. 2d to increase readability of the data.

We will change this, as suggested by the reviewer.

Decision Letter, second revision:

Dear Fiona,

Thank you for your email asking us to reconsider our decision on your manuscript, "Mechanical factors induce Myc-dependent dedifferentiation of adult epidermal cells via direct reversal of the terminal differentiation trajectory". We are always willing to hear the authors' perspective, but we must first prioritize decisions on new submissions. We appreciate your patience while we considered this appeal.

I have discussed again your manuscript, the referees' comments and your rebuttal, in detail with my colleagues. While we would be open to further considering the points you raised, we cannot commit to a specific course of action without further referee input. We would therefore need to send your manuscript back to the original referees before we decide whether to proceed with the manuscript or not. In case you are interested in this option, you could use the link below to resubmit with a revised manuscript and a revised rebuttal, if you will.

In addition, we would ask you again to ensure you have paid close attention to our guidelines on statistical and methodological reporting (listed below) as failure to do so may delay the reconsideration of the revised manuscript. In particular please provide:

- a Supplementary Figure including unprocessed images of all gels/blots in the form of a multi-page pdf file. Please ensure that blots/gels are labeled and the sections presented in the figures are clearly indicated.
- a Supplementary Table including all numerical source data in Excel format, with data for different figures provided as different sheets within a single Excel file. The file should include source data giving rise to graphical representations and statistical descriptions in the paper and for all instances where the figures present representative experiments of multiple independent repeats, the source data of all repeats should be provided.

On resubmission please provide the completed Editorial Policy Checklist (found here <https://www.nature.com/documents/nr-editorial-policy-checklist.pdf>), and Reporting Summary (found here <https://www.nature.com/documents/nr-reporting-summary.pdf>). This is essential for reconsideration of the manuscript and these documents will be available to editors and referees in the event of peer review. For more information see below. Please also ensure that the presentation of

statistical information in the revised submission complies with Nature Cell Biology's statistical guidelines (see below).

Please use the link below to submit the complete manuscript files, should you decide to proceed.

[Redacted]

Please let us know if you wish to proceed and when we can expect your revised manuscript.

With kind regards,

Stelios

Stylios Lefkopoulos, PhD
He/him/his
Associate Editor
Nature Cell Biology
Springer Nature
Heidelberger Platz 3, 14197 Berlin, Germany

E-mail: stylios.lefkopoulos@springernature.com

Twitter: @s_lefkopoulos

GUIDELINES FOR EXPERIMENTAL AND STATISTICAL REPORTING

REPORTING REQUIREMENTS – To improve the quality of methods and statistics reporting in our papers we have recently revised the reporting checklist we introduced in 2013. We are now asking all life sciences authors to complete two items: an Editorial Policy Checklist (found here <https://www.nature.com/documents/nr-editorial-policy-checklist.pdf>) that verifies compliance with all required editorial policies and a reporting summary (found here <https://www.nature.com/documents/nr-reporting-summary.pdf>) that collects information on experimental design and reagents. These documents are available to referees to aid the evaluation of the manuscript. Please note that these forms are dynamic 'smart pdfs' and must therefore be downloaded and completed in Adobe Reader. We will then flatten them for ease of use by the reviewers. If you would like to reference the guidance text as you complete the template, please access these flattened versions at <http://www.nature.com/authors/policies/availability.html>.

We strongly recommend the presentation of source data for graphical and statistical analyses as a separate Supplementary Table, and request that source data for all independent repeats are provided when representative experiments of multiple independent repeats, or averages of two independent experiments are presented. This supplementary table should be in Excel format, with data for different figures provided as different sheets within a single Excel file. It should be labelled and numbered as one of the supplementary tables, titled “Statistics Source Data”, and mentioned in all relevant figure legends.

Author Rebuttal, second revision:
--

Reviewer #1 (Remarks to the Author):

The revised manuscript has addressed many of the points of this and the other reviewers. In particular, the single cell RNAseq experiments have been improved. However, despite providing additional data on the hydrogel implantation experiments, the evidence that this experiment would exert mechanical forces on the epidermis remains rather weak and overall the conclusion that the effects of MYC on cell fate as implied by the title remain largely unsubstantiated. My recommendation is to substantially tone down the conclusions for this part (also strongly recommend revising the title) as well as addressing the points below.

In the previous revised version, we presented evidence that hydrogel implantation causes cell contractility in Gata6⁺ epidermal cells in a Myc-dependent manner, as indicated by p-MLC2 activation (Fig. 7f, g). In addition, we showed that Myc modulates actin-cytoskeleton remodelling associated with dedifferentiation in hydrogel-treated skin (Fig. S6f-h). In the present revised version, we have addressed the points raised by the reviewer as set out below, we have toned down the conclusions from the hydrogel experiments, and have changed the title of the manuscript as follows: ‘Myc-dependent dedifferentiation of Gata6⁺ epidermal cells resembles reversal of terminal differentiation’

Specific points:

1. The fact that hydrogel implantation results in mild inflammation which “peaks” a few days after the hyperproliferative response is not strong evidence that damage within the epidermis and the resulting signals coming from the epidermis would not trigger the hyperproliferative response. These types of signals do not necessarily require immune cell infiltration, eg TNF secretion from the keratinocytes themselves could cause this.

We agree with the reviewer that inflammatory signalling does not require immune cell infiltration. Aragona et al., found that epidermal inflammation after hydrogel implantation was not linked to cell proliferation (Aragona et al., Nature 2020). Since we showed that dedifferentiation did not require proliferation (Fig. S2d, e), whether hydrogel-induced proliferation was caused by inflammation or by mechanical factors was not the purpose of our experiment. Our aim was to examine whether inflammation had an effect on dedifferentiation.

In the previously revised manuscript we showed that inflammation caused by bleomycin does not induce dedifferentiation (Fig. 5g-j), even though Tnfa and Ccl2, two markers of inflammation, were upregulated by bleomycin treatment (FigS5i). However, as the reviewer states, we cannot rule out the possibility that other signalling related to inflammation is involved in dedifferentiation and we have modified the text accordingly.

The authors should specifically demonstrate activation of mechanosignaling in the epidermis: YAP/MRTF activation, pMLC2 activation, remodeling of the actin cytoskeleton, cell shape changes to warrant the conclusion that the hydrogels in fact relay mechanical stress to the epidermis.

In the previous revised version, we showed that pMLC2, a measure of cell contractility, is upregulated upon hydrogel injection (Fig. 7f, g). Remodelling of the actin cytoskeleton after hydrogel implantation in wt and Myc knockout mice was also shown (Fig. S6f-h). As suggested by the reviewer, we have now examined YAP/MRTF activation after hydrogel implantation (Fig. 5a, b). YAP and MAL translocated to the nucleus following hydrogel implantation and this

translocation was abolished in Gata6lin+ cells of Myc-depleted mice (Fig. 5n, o, Fig. S5l, m). These results indicate that hydrogel implantation causes mechanotransduction, which is Myc-dependent in Gata6lin+ cells.

2. This reviewer had misunderstood what has been measured by AFM, apologies for that. But the fact that the authors claim to be measuring cell stiffness from cross sections in cryo-embedded tissues is even more confusing. The cytoplasm of epidermal cells is quite small and most of the intracellular space is occupied by the nucleus. By measuring dead cells from a cryomatrix-embedded cross section, it is not clear how cell stiffness can be inferred or what it actually means. Given that the AFM measures a material composite, it is most likely that most of the force curves are derived from a nuclear cross section. How is nuclear stiffness related to the functions of Myc and the “mechanical effects” that are studied here is unclear. I would strongly recommend removing the in vivo AFM measurements as they are inconclusive at best and potentially artefactual.

The reviewer raises an interesting point; however, we disagree that the in vivo AFM measurements should be removed. There are several publications that have examined stiffness on histological sections, which support the validity of the approach (Fiore et al., Nature 2020, Bildstein et al., PNAS 2016). In these studies, the authors reported changes in stiffness along the epidermal layers (Fiore et al., 2020) and the hair follicle (Bildstein et al., 2016). While it is probably true that in tissue most of the mechanical response arises from the cell nuclei, we nevertheless see a clear difference when comparing distal HFs with proximal HFs and when comparing wt with Myc knockout mice (Fig. 6a-c). This indicates that there is a mechanical response to wounding, which is regulated by Myc. The in vivo AFM measurements are in strong agreement with the single cell observations that show that Myc depletion leads to stiffer cells (Fig. 6d). Since the single cell stiffness measurements were performed on the entire cellular area, the mechanical response is most likely dominated by the nuclear rigidity. In light of the reviewer’s comment, we now discuss the possibility that the mechanical response is dominated by the stiffness of the nucleus and that Myc may play a role in the regulation of nuclear mechanics.

3. The in vitro experiments using AFM force indentation for stiffness measurements are more logical. Given the indentation depth the authors presumably measure the stiffness of the apical cell cortex. Thus, it is surprising that increased stiffness is accompanied by reduced cortical actin in the Myc-deficient keratinocytes. What is the mechanism of increased stiffness? The authors should analyze the keratin network to reconcile this potential contradiction, for example increased keratin 10 could explain this (see eg work from Thomas Magin laboratory). But again, the altered mechanical properties might very well be a cause, rather than a consequence (as the authors propose) of the observed other changes.

We have already presented evidence that Myc-depleted cells express higher levels of Krt10 (Fig. S6a, b) and the differentiation marker Cst6 (Fig 6h, i), and discussed how this could account for the effect of Myc depletion on cell stiffness both in the revised text and in our previous point-by-point response (Reviewer 1, point 3). In the latest version of the manuscript, we now discuss the work from Thomas Magin’s lab.

Reviewer #2 (Remarks to the Author):

The authors have made an enormous effort to address the reviewers’ questions by performing most of the suggested experiments and analyses, which certainly is positively noted. Despite this

effort, as seen by my comments below, one of the central original concerns remains, namely are the authors indeed looking at dedifferentiation in this manuscript?

We thank the reviewer for appreciating the effort we put into addressing all the reviewers' comments. We present strong evidence throughout the paper that we are looking at dedifferentiation, such as the transition from suprabasal-like to basal-like signatures (Fig. 2d), the FACS data (Fig. 1a, FigS1d-f), and the location of Gata6lin+ cells exclusively in the suprabasal layers of unwounded/ctrl epidermis (Fig. 1b, Fig. 4a, Fig. 5i, Fig. 7a, Fig. S2d, Fig. S5f, Fig. S6a). Moreover, the switch from Gata6lin+ to Lrig1+ stem cells is irrefutable evidence of dedifferentiation, as Lrig1+ cells give rise to the Gata6 lineage (Fig. 2f, Fig. S2c, Fig. 4a, b). Additional evidence of dedifferentiation is provided by the initial enrichment of Lrig1+ stem cells, which is due to the switch from the Gata6lin+ to the Lrig1+ lineage, and not due to proliferation of Lrig1+ cells (Fig. S2d, e). Finally, Myc-knockout in three different contexts (wounding, retinoic acid, and hydrogels) prevents the transition from the suprabasal to the basal layer in Gata6lin+ cells as well as the conversion from Gata6 lineage into Lrig1 lineage (Fig. 4d-g, Fig. 5k-m, Fig. S5a-d).

Overall, none of the points that the authors bring up prove that the Gata6+Tom+ labelled cells in unwounded skin are exclusively suprabasal (majority of cells being suprabasal is not sufficient). The authors most often refer to their previous study (Donati et al NCB 2017), however in the present study it clearly can be seen that basal cells are labelled even before wounding (clearly seen in Fig S1b; counting at least 3 basal cells, or Fig S5a control; counting several basal cells in the HF including infundibulum and HF opening). In the FACS plots, we do indeed see a “shift” towards higher Itga6 intensities, which is consistent with wound response-inducing expression of basal genes (Joost et al Cell Rep 2018), but again this is not proving dedifferentiation. When comparing the gene expression tSNEs in Fig 2, one can also see several 0d cells with both Gata6 expression and Itga6 expression within several distinct cell populations.

The referee's comment that the “majority of suprabasal cells is not sufficient” dismisses the multiple lines of quantitative data we have provided. Fig. 1b, Fig. 4a, Fig. 5i, Fig. 7a, Fig. S2d, Fig. S5f, FigS6a show representative examples of intact HFs where virtually all the Gata6lin+ cells are suprabasal and there is no overlap between Gata6+ and Itga6+ cells (Fig. 1b, Fig. S1c). It is only in wound-proximal HFs and healing IFE that Gata6lin+ cells express Itga6 in the basal layer, consistent with the transition from the suprabasal to basal position (Fig. 1b). Please see Fig. 5f for a quantification of the ratio between suprabasal and basal cells. As shown in this figure, 100% of the cells are suprabasal in the control situation. Lin+ cells are present in the basal layer only after hydrogel injection. Furthermore, in the new revised version of the manuscript we have included additional quantification showing the ratio between suprabasal and basal cells upon wounding (Fig. S1c). Virtually all the lin+ cells are suprabasal in unwounded mice. We agree that Figs S1b and S5a may be confusing due to oversaturation and we have replaced these images accordingly.

Additional evidence of dedifferentiation is shown in Fig. 2 and Fig. S2, which clearly show that there is a downregulation of differentiation markers and upregulation of basal markers. We observe this transition not only at the single cell level but also when we compare gene expression at different time points after wounding (Fig. 2 and Fig. S2).

While single-cell RNA-seq analysis is helpful to understand cell heterogeneity, it does not reflect cell identity and cell fate with 100% accuracy. Rather, it predicts cell identity and cell fate based on pre-established datasets/sets of genes. In light of the referee's concern about Gata6lin+ cells being suprabasal, we have revisited our single-cell data and re-analysed our cell identity prediction for the lin+ population using the second level of clustering from the Joost dataset (Fig. S1h). Lin+ cells strongly correlated with uHF-V, which has been predicted to be a suprabasal

compartment. This was confirmed by high expression of *Krt79* and *Krt17* in control cells (Fig. S2a).

Regarding the tSNEs in Fig. 2, we do not observe 'several 0d cells with *Gata6* and *Itga6* expression', as the reviewer states. We acknowledge that there are a few cells with high *Itga6* expression at 0d (4.3% of the entire population). However, among these cells, only 1 cell shows high *Gata6* expression. It is technically difficult to achieve 100% purity when doing cell sorting (95-99% purity is the normal range). Thus, we cannot rule out the possibility that the *Itga6*⁺ cells are contaminants (*lin*⁻ cells) from the single-cell isolation, as they fall within that range and do not express *Gata6*. We now state this in the text.

In sum, how can the authors be sure that what they observe in this study are not basal cells that expand upon injury and persist within the basal layer?

We are confident that we are observing a transition from suprabasal to basal (Fig. 2d, Fig. 2f and Fig S2a-c), as explained above. Our single-cell data and our lineage tracing strongly indicate downregulation of suprabasal markers and upregulation of basal markers. Please see Fig. 1b for an overview of the entire process. Note that distal HFs show Gata6lin⁺ cells only in the suprabasal layers whereas proximal HFs show lin⁺ cells in suprabasal and basal positions.

Fig. S2d-e shows that dedifferentiation does not require proliferation. 1 day after wounding there is a significant enrichment of Gata6lin⁺ Lrig1⁺ cells that occurs irrespective of proliferation (Fig. S2d, e). Given that 1 day after wounding proliferation remains unchanged in Gata6lin⁺ cells the most plausible explanation is that this enrichment in Lrig1⁺ cells is due to dedifferentiation of suprabasal Gata6lin⁺ cells to Lrig1⁺ stem cells. Hypothetical basal cells would not contribute to this initial enrichment as they already express Lrig1. This is further evidence that we are looking at dedifferentiation.

Moreover, hydrogel injection caused a substantial increase (~20%) in the number of basal Gata6lin⁺ cells only 1 day after implantation (Fig. 5f-h). Our data shows that prior to hydrogel implantation 100% of lin⁺ cells are suprabasal (Fig. 5f). Thus, even if there were a few Gata6lin⁺ basal cells in healthy epidermis that we did not detect in our lineage tracing, it is extremely unlikely that they were responsible for such a dramatic effect.

Comparing this study to their previous Donati 2017 study, do *Gata6*⁺ cells of dorsal skin (this study) and tail skin (Donati 2017) behave differently? In Donati 2017, the *Gata6* GL cells seem to remain preferentially suprabasal at eg. day 3 post wounding (Fig 4a, left in Donati 2017) whereas in dorsal skin *Gata6* GL cells directly occupy a large basal area clearly seen at day 3 (Fig S2d in this study). This observation is later in the manuscript highlighted where the authors say (p. 8): "The location of these cells led us to speculate that dedifferentiation of *Gata6lin*⁺ cells occurs within the upper HF. Consistent with this, live cell imaging of anaesthetised mice showed that prior to *Gata6lin*⁺ cells migrating into the IFE there was an expansion of the *Gata6lin*⁺ compartment within the HFs at the wound margin (Extended Data Fig. 4a, b)." Firstly, how did the authors determine that this was indeed a suprabasal cell and not a dividing basal cell? Wound edges are challenging to image and the images only show nuclei and Tomato labelled cells but no basement membrane. Secondly, this in vivo wound healing process was imaged 3 days after wounding, a time point that was also included in Park et al NCB 2017, where the authors used a photo-activatable fluorescent reporter to label suprabasal cells and follow their destiny. Park 2017 conclude that 'Notably, cell displacement between layers was observed to occur exclusively in an upward direction at both time points examined, despite previous models proposing downward cell contribution from differentiated layers to the basal one (Supplementary Fig. 3).' It is noted that Park 2017 imaged ear skin, however, a second study using lineage tracing in tail also argues against dedifferentiation (Aragona et al Nat Commun 2017). It is still debated whether this may

be a timing issue (i.e. suprabasal-to-basal cell conversion takes longer time to manifest) or specifically Gata6⁺ cells are more responsive to conversion (Dekoninck & Blanpain 2018). As the original observation of dedifferentiation is obviously of high interest and often debated, it is especially important to exclude that pre-existing Gata6⁺ labelled basal cells are the main wound-response basal cell contributors instead of dedifferentiated suprabasal Gata6⁺ progeny.

We are confident that there are no such “pre-existing Gata⁺ labelled basal cells”, as outlined above. The papers that the reviewer cites have been discussed in the revised manuscript. In vivo time-lapse experiments shown in the present manuscript were performed in the ear, as in Donati et al., 2017. Collagen and nuclei are shown in green so we can distinguish the basement membrane (see Methods section). For clarity we have now added a middle and a bottom plane. In the present manuscript we focused on the hair follicle and show that there is an expansion of the Gata6⁺ compartment prior to migration into the IFE. We did not intend to discriminate between suprabasal and basal cells as the resolution is not high enough to do so in the HFs. Nevertheless, Donati et al. have already described migration of Gata6⁺ cells from suprabasal to basal compartments, so this is not the focus of the present manuscript. When characterising changes in cell state, we focused on the Gata6 population, which exclusively resides in the sebaceous duct, whereas other researchers have focused on IFE cell populations as discussed in our present manuscript and in numerous citations by others of our 2017 publication.

Even the inverse correlation of Gata6^{lin}⁺ cells that dedifferentiated to Lrig1⁺ stem cells and Lrig1⁺ lin⁻ cells that differentiated to Gata6⁺ cells could be explained if Gata6^{lin}⁺ cells are already basal. It has been shown that epidermal HF and IFE cells in proximity to a wound increase basal signature gene expression (Joost et al Cell Rep 2018; Haensel et al Cell Rep 2020), which could indeed lead to a seeming “dedifferentiation”, where the wound Gata6^{lin}⁺ cells acquire higher expression of basal markers than their unwounded counterparts.

We beg to differ with the reviewer’s conclusion. If the lin⁺ cells were already basal the correlation would be positive, not inverse. Please revisit Fig. 1b and note that Gata6^{lin}⁺ cells in distal/ctrl HFs are exclusively suprabasal whereas in proximal HFs they are in contact with the basement membrane and express Itga6 (Fig. 1b). Virtually all lin⁺ cells remain suprabasal 6 days after wounding (Fig. S1c) even though they may express basal markers (see day 6 cells in Fig. 2), “seeming dedifferentiation”, as indicated by the reviewer. Transcriptionally, most day 6 cells and control cells are indistinguishable. However, 25% of day 9 cells are basal (Fig. S1c) and cluster separately from day 0 and day 6 cells (Fig. 2a). This indicates that there is a transition into a less differentiated state and that cells that have dedifferentiated are different from those that may “seem dedifferentiating”. In hydrogel-injected skin, this transition is dramatically more pronounced since 20% of the cells are basal only 1 day after implantation (Fig. 5f). As 100% of the control lin⁺ cells are suprabasal and proliferation starts 2 days after injection (Fig. S5f, g), this indicates that the basal cells observed at day 1 arise from migration of suprabasal cells. Altogether, this supports our conclusion that Gata6^{lin}⁺ cells dedifferentiate and acquire stem cell-like properties.

Finally, following up on the Myc co-operation in the potential ‘dedifferentiation’ process: it is convincing that Myc^{-/-} dramatically reduces the contribution/efflux of cells to the wound IFE, but this again just could reflect the inhibition of proliferation/expansion of pre-existing basal Gata6⁺ labelled cells as those are also targeted.

Myc-depletion not only prevents migration from the HF to the IFE but also the conversion of Gata6^{lin}⁺ cells into Lrig1⁺ cells in three different contexts (Fig. 4d-f, Fig. 5k-m, Fig. S5a-d).

Given that Gata6+ cells arise from Lrig1+ cells, this strongly indicates there is a transition into a less differentiated state that is modulated by Myc. Notably, the switch from Gata6lin+ cells to Lrig1+ stem cells did not require proliferation (Fig. S2d, e), indicating that initiation of dedifferentiation occurs irrespective of cell proliferation. Thus, the increase in Gata6lin+ Lrig1+ cells is not due to “proliferation/expansion”, as the reviewer states, but also due to conversion of suprabasal Gata6lin+ cells to Lrig1+ stem cells.

All in all, to my very large regret – given the interesting topic and the major effort of the authors – it is not convincing that we are indeed looking at dedifferentiating cells (and not pre-existing expanding basal cells), which is the basis for a manuscript entitled ‘Mechanical factors induce Myc-dependent dedifferentiation of adult epidermal cells via direct reversal of the terminal differentiation trajectory’.

Minor comment: The authors emphasize to have performed RNA-velocity analysis, which is not the case. The “Cell fate transition analysis” method section describes their use of CellRank, which is a collection of methods to estimate cell trajectories based on several inputs (it can include RNA-velocity, but not by default). Here, they used the pseudotime kernel of CellRank with Monocle 3 pseudotime input to calculate cell fate transitions. This has nothing to do with RNA-velocity (i.e., estimating cell fate trajectories by their spliced-unspliced mRNA expression).

We apologise for this mistake. We have changed the text accordingly.

Reviewer #3 (Remarks to the Author):

Bernabe-Rubio et al. have comprehensively revised their manuscript and convincingly addressed most of my comments. However, some critical concerns remain, as outlined below:

1) For most analysis time points, the number of biological replicates and number of analysed single cells remains low, which weakens the overall robustness of the data presented in the study.

In the revised manuscript, we increased the number of single cells sequenced three-fold, included a biological replicate and added an earlier time point.

2) Comment 2 from the initial revision has not been addressed, as desired. Could the authors please provide data on the number of detected genes and transcripts/UMIs and percentage of mitochondrial genes per replicate (e.g., using box plots)?

We apologise for this mistake. We have now included the data as the reviewer requested (Fig. S1g).

3) Based on the new data provided in Fig. 2a & b, it seems a significant number of Gata6lin+ cells has the transcriptional profile of IFE basal layer cells on day 0 of wounded skin. In addition, Fig. 2e shows a moderate number of Gata6 read counts in these basal cells. In line with the concerns of reviewer #2, this leaves the possibility that a large fraction of Gata6+ cells may indeed be localised to the IFE basal layer at the beginning of the wounding experiment, which strongly weakens the authors’ conclusions on the involvement of predominantly dedifferentiating Gata6lin+ cells in IFE regeneration.

We acknowledge that the label transfer method used in Fig. 2b may not be accurate, as there is no significant fraction of unwounded Gata6lin+ cells in the IFE; indeed we did not see any in our

lineage tracing. This is supported by our single-cell dataset (Fig. 1c and Fig S1h) and by that of Joost et al., 2016. The Joost dataset shows that *Gata6* is primarily expressed in the uHF compartment, not in the IFE. Our Fig. 1c shows that the transcriptomes of *Gata6*⁺ cells map primarily to the uHF compartment. In addition, new Fig. S1h shows that day 0 *lin*⁺ cells strongly correlate with the suprabasal compartment uHF-V, and not with the IFE compartment.

For clarity, we have now included heatmaps showing the correlation between cell clusters and epidermal compartments (Fig. 2b). The label transfer method used in the previous version predicts cell identity based on highly variable genes, whereas the enrichment-based method uses well-characterised set of genes. We used appropriate statistical methods (one-tailed tests) and multiple testing corrections (BH-adjusted p-value).

The results in Fig. 2b show a transition from HF to IFE. Our earlier study (Donati et al., 2017) predicted that HF and IFE *Gata6*⁺ cells would share transcriptional profiles after wounding, based on immunofluorescence labelling for a range of different markers and so it is not surprising that *Gata6*⁺ cells fall into the IFE category.

4) The authors have added biaxial plots in Fig. 3c, in response to comment 5 from the initial revision. In the accompanying manuscript text, the authors state the plots “show a high correlation between *Myc* and *Lrig1* expression in wound cells”. However, an *r* value of 0.24 is a weak correlation at most (typically, *r* values below 0.25 indicate no relationship). The authors should therefore tone down their statement. Of note, the label on the x-axis, presumably displaying *Myc* expression, is missing.

We have toned down our conclusions and corrected the labels.

5) Fig. 7f shows significant autofluorescence of the hair shaft in some images. Could the authors please clarify whether/how they have considered this in their quantifications shown in Fig. 7g?

We excluded the hair shaft in our analysis and only measured areas containing Tomato⁺ cells. We have clarified this in the figure legend.

Minor comment:

6) I would recommend the use of two diverging colours for the colour coding in Fig. 2d to increase readability of the data.

We have changed this, as suggested by the reviewer.

Decision Letter, third revision:

Our ref: NCB-A47462C

6th July 2023

Dear Fiona,

Thank you for submitting your revised manuscript "Myc-dependent dedifferentiation of Gata6+ epidermal cells resembles reversal of terminal differentiation" (NCB-A47462C). Unfortunately, the original reviewer #3 was unable to re-review and we therefore had to ask reviewer #1 to cross-comment on your responses to the comments by original reviewer #3. Therefore, your manuscript has now been seen by the original referees #1 and #2 and their comments are below. The reviewers find that the paper has improved in revision, and therefore we'll be happy in principle to publish it in Nature Cell Biology, pending minor revisions to satisfy the referees' final requests (textually address the limitations noted by reviewer #1) and to comply with our editorial and formatting guidelines.

If the current version of your manuscript is in a PDF format, please email us a copy of the file in an editable format (Microsoft Word or LaTeX)-- we cannot proceed with PDFs at this stage.

Thank you again for your interest in Nature Cell Biology Please do not hesitate to contact me if you have any questions.

Best wishes,
Stelios

Stylios Lefkopoulos, PhD
He/him/his
Associate Editor
Nature Cell Biology
Springer Nature
Heidelberger Platz 3, 14197 Berlin, Germany

E-mail: stylios.lefkopoulos@springernature.com

Twitter: @s_lefkopoulos

Reviewer #1 (Remarks to the Author):

The authors have further made efforts to improve the manuscript and the additional data has strengthened the conclusions. Also it is reassuring to see that the manuscript has been edited to indicate caveats and to exercise caution on interpretations.

Specifically considering the points raised by this reviewer and Rev 3 (as requested by the editor); the data remains somewhat noisy. In particular looking at Figs 1e and 2a it still cannot be fully excluded that some of the de-differentiation signal comes from leakiness of the reporter allele and steady state detection of the lin⁺ cells in the IFE (why is the TdTomato signal so weak in the post-wounding colonies compared to everywhere else?; why are day 0 cells spread so evenly through all clusters?) Also the hydrogel experiments remain difficult to interpret due to the fact that it is not clear how mechanical forces from the expanding hydrogel are transmitted to the GATA⁺ cells and myc, as well as the precise contribution of mechanics vs other damage signaling in this process. These aspects will remain open for future studies.

Reviewer #2 (Remarks to the Author):

Given the high interest and ongoing vibrant debate about (Gata6⁺ cell) de-differentiation, the in-depth revisit of the Tom⁺ initial labeling (unwounded) images and their replacement to truly representative ones was critically important. Thank you for clarifying this major concern and related questions. The additional RNA-seq analysis and textual clarifications also strengthen this work and I have no more requests.

Author Rebuttal, third revision:

Reviewer #1 (Remarks to the Author):

The authors have further made efforts to improve the manuscript and the additional data has strengthened the conclusions. Also it is reassuring to see that the manuscript has been edited to indicate caveats and to exercise caution on interpretations.

Specifically considering the points raised by this reviewer and Rev 3 (as requested by the editor); the data remains somewhat noisy. In particular looking at Figs 1e and 2a it still cannot be fully excluded that some of the de-differentiation signal comes from leakiness of the reporter allele and steady state detection of the lin⁺ cells in the IFE (why is the TdTomato signal so weak in the post-wounding colonies compared to everywhere else?; why are day 0 cells spread so evenly through all clusters?) Also the hydrogel experiments remain difficult to interpret due to the fact that it is not clear how mechanical forces from the expanding hydrogel are transmitted to the GATA⁺ cells and myc, as well as the precise contribution of mechanics vs other damage signaling in this process. These aspects will remain open for future studies.

We thank the reviewer for the positive feedback.

Fig. 1e shows hair follicles proximal to the wound (left panels) and wounded IFE (right panels) so it is expected to find Gata6lin⁺ cells in the IFE.

In the absence of Tamoxifen, we do not detect any tdTomato signal. Moreover, under Tamoxifen treatment, control Gata6lin⁺ cells are exclusively detected in the uHF. Fig. 5g and Fig S5d (n=3 mice and n=4 mice, respectively) show a quantification of the percentage of Gata6lin⁺ cells in the IFE. In steady state conditions (Ctrl), no Gata6lin⁺ cells were detected in the IFE. In Fig. S1h we only analysed day 0 Gata6lin⁺ cells observing that they strongly correlate with the uHF-V compartment (second level of clustering from the Joost dataset). Most day 0 Gata6lin⁺ cells in Fig. 2a fall on clusters 3 and 2, which correlate with the uHF-II compartment (first level of clustering from the Joost dataset). As mentioned in the response to reviewer 2, we acknowledge that a few day 0 lin⁺ cells (4.3% of the entire population) express high levels of Itga6 and might be contaminants (lin⁻ cells) from the single-cell isolation, as they fall within the normal range of single-cell sorting purity (95-99%). These contaminants might have an IFE-like signature, as shown in Fig. 1c. We have modified the text accordingly.

We fully agree with the reviewer that the precise mechanisms by which mechanical forces are transmitted to the Gata6 population upon hydrogel injection are not clear. Different aspects, such as hair follicle orientation, rearrangement of cell junctions or changes in stiffness caused by hydrogels, could be considered in future studies. The precise degree to which mechano-signalling pathways contribute to dedifferentiation compared to damage signalling pathways will also require further investigation. We now state this in the text.

Reviewer #2 (Remarks to the Author):

Given the high interest and ongoing vibrant debate about (Gata6+ cell) de-differentiation, the in-depth revisit of the Tom+ initial labeling (unwounded) images and their replacement to truly representative ones was critically important. Thank you for clarifying this major concern and related questions. The additional RNA-seq analysis and textual clarifications also strengthen this work and I have no more requests.

We thank the reviewer for the positive feedback.

Decision Letter, final checks:

Our ref: NCB-A47462C

17th July 2023

Dear Dr. Watt,

Thank you for your patience as we've prepared the guidelines for final submission of your Nature Cell Biology manuscript, "Myc-dependent dedifferentiation of Gata6+ epidermal cells resembles reversal of terminal differentiation" (NCB-A47462C). Please carefully follow the step-by-step instructions provided in the attached file, and add a response in each row of the table to indicate the changes that you have made. Please also check and comment on any additional marked-up edits we have proposed within the text. Ensuring that each point is addressed will help to ensure that your revised manuscript can be swiftly handed over to our production team.

In recognition of the time and expertise our reviewers provide to Nature Cell Biology's editorial process, we would like to formally acknowledge their contribution to the external peer review of your manuscript entitled "Myc-dependent dedifferentiation of Gata6+ epidermal cells resembles reversal of terminal differentiation". For those reviewers who give their assent, we will be publishing their names alongside the published article.

Nature Cell Biology offers a Transparent Peer Review option for new original research manuscripts submitted after December 1st, 2019. As part of this initiative, we encourage our authors to support increased transparency into the peer review process by agreeing to have the reviewer comments, author rebuttal letters, and editorial decision letters published as a Supplementary item. When you submit your final files please clearly state in your cover letter whether or not you would like to participate in this initiative. Please note that failure to state your preference will result in delays in accepting your manuscript for publication.

Cover suggestions

As you prepare your final files we encourage you to consider whether you have any images or illustrations that may be appropriate for use on the cover of Nature Cell Biology.

Nature Cell Biology has now transitioned to a unified Rights Collection system which will allow our Author Services team to quickly and easily collect the rights and permissions required to publish your work. Approximately 10 days after your paper is formally accepted, you will receive an email in providing you with a link to complete the grant of rights. If your paper is eligible for Open Access, our Author Services team will also be in touch regarding any additional information that may be required to arrange payment for your article.

Please note that *Nature Cell Biology* is a Transformative Journal (TJ). Authors may publish their research with us through the traditional subscription access route or make their paper immediately open access through payment of an article-processing charge (APC). Authors will not be required to make a final decision about access to their article until it has been accepted. Find out more about Transformative Journals

Please use the following link for uploading these materials:
[Redacted]

Best regards,

Kendra Donahue
Staff
Nature Cell Biology

On behalf of

Stylios Lefkopoulos, PhD
He/him/his
Associate Editor
Nature Cell Biology
Springer Nature
Heidelberger Platz 3, 14197 Berlin, Germany

E-mail: stylios.lefkopoulos@springernature.com
Twitter: @s_lefkopoulos

Reviewer #1:

Remarks to the Author:

The authors have further made efforts to improve the manuscript and the additional data has strengthened the conclusions. Also it is reassuring to see that the manuscript has been edited to indicate caveats and to exercise caution on interpretations.

Specifically considering the points raised by this reviewer and Rev 3 (as requested by the editor); the data remains somewhat noisy. In particular looking at Figs 1e and 2a it still cannot be fully excluded that some of the de-differentiation signal comes from leakiness of the reporter allele and steady state detection of the lin⁺ cells in the IFE (why is the TdTomato signal so weak in the post-wounding colonies compared to everywhere else?; why are day 0 cells spread so evenly through all clusters?) Also the hydrogel experiments remain difficult to interpret due to the fact that it is not clear how mechanical forces from the expanding hydrogel are transmitted to the GATA⁺ cells and myc, as well as the precise contribution of mechanics vs other damage signaling in this process. These aspects will remain open for future studies.

Reviewer #2:

Remarks to the Author:

Given the high interest and ongoing vibrant debate about (Gata6⁺ cell) de-differentiation, the in-depth revisit of the Tom⁺ initial labeling (unwounded) images and their replacement to truly representative ones was critically important. Thank you for clarifying this major concern and related questions. The additional RNA-seq analysis and textual clarifications also strengthen this work and I have no more requests.

Final Decision Letter:

Dear Fiona,

I am pleased to inform you that your manuscript, "Myc-dependent dedifferentiation of Gata6+ epidermal cells resembles reversal of terminal differentiation", has now been accepted for publication in Nature Cell Biology. Congratulations to you and the whole team!

Please note that *Nature Cell Biology* is a Transformative Journal (TJ). Authors may publish their research with us through the traditional subscription access route or make their paper immediately open access through payment of an article-processing charge (APC). Authors will not be required to make a final decision about access to their article until it has been accepted. Find out more about Transformative Journals

Authors may need to take specific actions to achieve compliance with funder and institutional open access mandates. If your research is supported by a funder that requires immediate open access (e.g. according to Plan S principles) then you should select the gold OA route, and we will direct you to the compliant route where possible. For authors selecting the subscription publication route, the journal's standard licensing terms will need to be accepted, including self-archiving policies. Those licensing terms will supersede any other terms that the author or any third

party may assert apply to any version of the manuscript.

If you have not already done so, we strongly recommend that you upload the step-by-step protocols used in this manuscript to the Protocol Exchange (www.nature.com/protocolexchange), an open online resource established by Nature Protocols that allows researchers to share their detailed experimental know-how. All uploaded protocols are made freely available, assigned DOIs for ease of citation and are fully searchable through nature.com. Protocols and Nature Portfolio journal papers in which they are used can be linked to one another, and this link is clearly and prominently visible in the online versions of both papers. Authors who performed the specific experiments can act as primary authors for the Protocol as they will be best placed to share the methodology details, but the Corresponding Author of the present research paper should be included as one of the authors. By uploading your Protocols to Protocol Exchange, you are enabling researchers to more readily reproduce or adapt the methodology you use, as well as increasing the visibility of your protocols and papers. You can also establish a dedicated page to collect your lab Protocols. Further information can be found at www.nature.com/protocolexchange/about

With kind regards,
Stelios

Stylianos Lefkopoulos, PhD
He/him/his
Associate Editor
Nature Cell Biology
Springer Nature
Heidelberger Platz 3, 14197 Berlin, Germany

E-mail: stylianos.lefkopoulos@springernature.com
Twitter: @s_lefkopoulos